# Incentivizing Consistent, Effective and Scalable Reasoning Capability in Audio LLMs via Reasoning Process Rewards

**Jiajun Fan**[◇,*] **Roger Ren**[△]**, Jingyuan Li**[△]**, Rahul Pandey**[△]**, Prashanth Gurunath Shivakumar**[△]

**Ivan Bulyko**[△]**, Ankur Gandhe**[△]**, Ge Liu**[◇]**, Yile Gu**[△]

◇ Siebel School of Computing and Data Science, University of Illinois Urbana-Champaign
△ Amazon
◇ {jiajunf3, geliu}@illinois.edu
△ {rogerren, jylii, rpandyn, psshvak, ibbulyko, aggandhe, yilegu}@amazon.com

## Abstract

The role of reasoning in Audio Large Language Models remains widely under-explored, as introducing a reasoning process often degrades rather than improves performance during inference, a phenomenon we term test-time inverse scaling, where longer reasoning chains yield progressively worse results. We demonstrate that this stems not from fundamental limitations of reasoning itself, but from inadequate training: models without proper guidance for the reasoning process produce hallucinatory, inconsistent reasoning that accumulates errors over longer chains. To address these challenges, we introduce CESAR (Consistent, Effective, and Scalable Audio Reasoners), shifting from outcome verification to rewarding the reasoning process. Our online reinforcement learning framework employs Group Relative Policy Optimization with a multi-faceted reward suite that incentivizes not only correctness and format but also consistency, structured analytical patterns, causal reasoning, domain-knowledge integration, and calibrated reasoning depth. CESAR resolves test-time inverse scaling, transforming reasoning from detriments into gains while revealing model-specific "reasoning sweet spots", where performance peaks during test-time scaling. We achieve state-of-the-art results on MMAU Test-mini, substantially outperforming Gemini 2.5 Pro and GPT-4o Audio, and near-human-level performance on MMSU reasoning tasks. Through AI-as-judge evaluations and qualitative comparisons, we provide both quantitative and qualitative validation of our improved reasoning quality. Importantly, enhanced reasoning creates synergistic effects, simultaneously improving multimodal reasoning and perception capabilities. Overall, CESAR establishes a principled method for developing robust and scalable reasoning in Audio LLMs.

## 1 Introduction

The advent of Audio Large Language Models (Audio LLMs) has opened a new frontier in multi-modal AI, promising sophisticated understanding of complex acoustic environments (Gong et al., 2024; Tang et al., 2024; Xu et al., 2025). Yet, a critical paradox emerges when these models are asked to reason: while chain-of-thought (CoT) prompting is a proven catalyst for reasoning in text-based domains (Wei et al., 2022; Jaech et al., 2024; DeepSeek-AI et al., 2025), in audio it often backfires, underperforming non-reasoning versions. We are the first to systematically identify and diagnose this phenomenon as a **test-time inverse scaling problem** in Audio LLMs, where reasoning processes not only fail to improve performance but actively degrade it during inference, with longer reasoning chains yielding progressively worse results—often underperforming their direct answering versions that bypass reasoning entirely (Fig. 1). This test-time inverse scaling might lead to the

---
*Work done as intern at Amazon

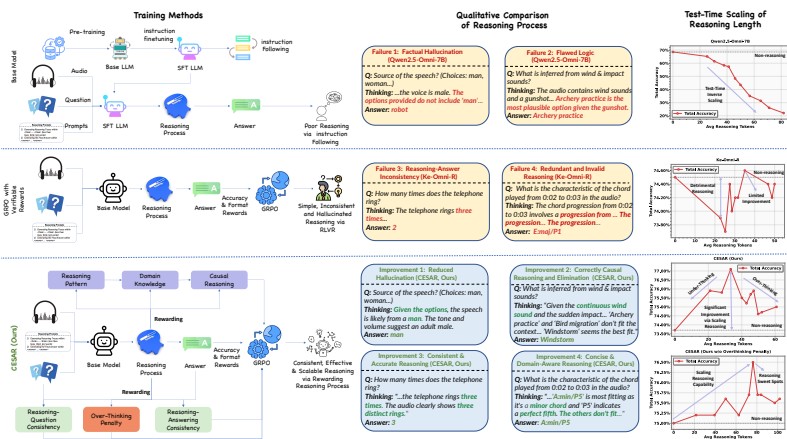

Figure 1: General Framework of Different Training Methods for Audio Reasoning Models.

premature conclusion that reasoning is inherently harmful for Audio LLMs (Li et al., 2025), but our investigation reveals the true culprit: models produce hallucinatory, inconsistent, and logically unsound reasoning processes when forced to "think" without proper training on *how* to reason.

Current methodologies are fundamentally ill-equipped to solve this problem. The dominant approach—supervised fine-tuning (SFT) on CoT datasets (Ma et al., 2025a; Xie et al., 2025)—teaches models to merely memorize and mimic reasoning templates rather than developing genuine analytical capabilities. While recent reinforcement learning with verifiable rewards (RLVR) methods (Li et al., 2025; Zhao et al., 2025) represent progress, they remain constrained by outcome-only reward structures that exclusively value final answer correctness and format compliance. This shallow supervision fails to address the root cause: poor reasoning processes that accumulate errors over longer chains, allowing models to generate final answers through flawed or irrelevant logic while perpetuating the very issues of inconsistency and hallucination that lead to test-time inverse scaling.

We address these limitations by introducing **CESAR** (Consistent, Effective and Scalable Audio Reasoners), representing a fundamental **paradigm shift from outcome verification to rewarding the reasoning process**. Our framework leverages Group Relative Policy Optimization (GRPO) (Shao et al., 2024) not merely as a verifier, but as a mechanism to explicitly cultivate reasoning as a controllable, trainable skill. At its core lies a multi-faceted reward suite that provides granular feedback on the reasoning process itself, systematically incentivizing answer **correctness**, **format** compliance, **consistency** between thoughts and answers, **structured** analytical patterns, **causal** reasoning, **domain** knowledge integration, and **calibrated** reasoning depth that avoids catastrophic overthinking. Through this process-centric approach, we transform reasoning from an unpredictable liability into a reliable, scalable asset that enables effective test-time scaling through discovered "reasoning sweet spots." In summary, our approach makes several key contributions:

1. We identify and diagnose the **test-time inverse scaling phenomenon** in Audio LLMs, where reasoning processes degrade performance during inference due to hallucination, inconsistency, and unstructured thought patterns. We demonstrate this stems from inadequate training of reasoning processes rather than unsolvable limitations of reasoning itself.

2. We propose **CESAR**, a framework employing reasoning process rewards that incentivize consistency, structured analytical patterns, domain knowledge integration, and calibrated reasoning depth to extend current outcome-only RLVR methods. With GRPO, CESAR explicitly cultivates robust reasoning, resolving the test-time inverse scaling problem.

3. We demonstrate that cultivating robust reasoning capability by **CESAR unlocks effective test-time scaling**: while poorly trained models suffer catastrophic degradation with longer reasoning chains, our models discover optimal "reasoning sweet spots" for substantial training-free performance gains, validating our scalable reasoning capability.

4. Our method achieves **SOTA** on MMAU Test-mini, surpassing GPT-4o Audio and Gemini 2.5 Pro, and **near human-level performance** on MMSU reasoning tasks. Most impor-

tantly, we observe that enhanced reasoning creates synergistic effects that simultaneously improve multimodal reasoning and perception capabilities.

5. We introduce a novel **AI-as-judge** evaluation framework, **human evaluations** and comprehensive **qualitative analysis** that rigorously validate our enhanced reasoning quality, demonstrating commanding win rates against strong baselines and concrete reductions in hallucination while improving logical coherence and reasoning-answer consistency.

## 2 RELATED WORK

**Audio Large Language Models.** The development of Audio Large Language Models (Audio LLMs) has rapidly progressed from foundational audio-to-text tasks to sophisticated multimodal systems. Early work (Elizalde et al., 2023; Wu et al., 2022) established cross-modal understanding through contrastive learning. This paved the way for decoder-based models capable of open-ended generation (Deshmukh et al., 2023; Gong et al., 2024). The current generation of models, such as SALMONN (Tang et al., 2024), the Qwen-Audio series (Chu et al., 2023; 2024), and Audio Flamingo (Kong et al., 2024; Goel et al., 2025a), have demonstrated increasingly comprehensive capabilities through large-scale pre-training and instruction tuning. As state-of-the-art models like Qwen2.5-Omni (Xu et al., 2025), GPT-4o Audio (Hurst et al., 2024) and Gemini 2.5 (Comanici & et al., 2025) achieve near-human audio understanding, the research frontier has shifted towards a more profound challenge: enabling these models to genuinely *reason* about the acoustic world.

**The Limits of Supervised Reasoning.** Chain-of-thought (CoT) prompting has been transformative for eliciting reasoning in text-based LLMs (Wei et al., 2022). Naturally, this paradigm was extended to the audio domain through supervised fine-tuning (SFT) on CoT datasets (Ma et al., 2025a; Xie et al., 2025; Goel et al., 2025a). However, these SFT-based approaches share a fundamental limitation: they teach models to **imitate reasoning templates, not to develop genuine analytical skill**. This results in models that can produce syntactically plausible reasoning traces but which are often brittle, fail to generalize to complex, unseen problems, and do not address the underlying causes of reasoning failure we identify, such as hallucination and inconsistency.

**The Untapped Potential of Reinforcement Learning.** Reinforcement learning (RL) offers a promising alternative to supervised imitation, as demonstrated by the success of models like OpenAI's o1 (Jaech et al., 2024) and DeepSeek-R1 (DeepSeek-AI et al., 2025) in the text domain. These works show that sophisticated reasoning can be cultivated directly through reward optimization, with methods like Group Relative Policy Optimization (GRPO) (Shao et al., 2024) proving particularly effective. Early attempts to apply these techniques to audio, such as R1-AQA (Li et al., 2025) and Ke-Omni-R (Zhao et al., 2025), have shown initial success. However, they are fundamentally constrained by an **outcome-oriented reward paradigm**, optimizing solely for the correctness of the final answer. This shallow supervision signal is insufficient; it fails to penalize logical fallacies or reward coherent analytical processes, thereby directly contributing to the poor reasoning capability that causes the test-time inverse scaling problem. Our work addresses this gap by moving from outcome verification to a granular, process-oriented reward system.

## 3 METHODOLOGY

Existing audio reasoning methods like Ke-Omni-R (Zhao et al., 2025) suffer from reasoning-answer inconsistency, unstructured reasoning, and test-time inverse scaling (Fig. 1) caused by outcome-only verifiable rewards and uncontrolled reasoning emergence. In this paper, we propose CESAR to transform reasoning into a controllable skill through comprehensive reasoning process rewards that incentivize consistency, structured reasoning, and optimal reasoning depth—while discovering model-specific "reasoning sweet spots" where performance peaks during test-time scaling.

### 3.1 PROBLEM FORMULATION

Let $\mathcal{D} = (a_i, q_i, \mathcal{C}_i, y_i)_{i=1}^{N}$ denote the audio question-answering dataset, where $a_i$ represents the audio input, $q_i$ is the question, $\mathcal{C}_i = c_1, c_2, c_3, c_4$ is the set of multiple-choice options, and $y_i \in \mathcal{C}_i$ is the ground truth answer. Our goal is to train an Audio LLM $\pi_\theta$ that can generate both a reasoning

process $t_i$ and a final answer $\hat{y}_i$ given the input $(a_i, q_i, C_i)$. The model output follows a structured format where we have:

$$\pi_\theta(a_i, q_i, C_i) = \langle\text{think}\rangle t_i \langle/\text{think}\rangle \langle\text{answer}\rangle \hat{y}_i \langle/\text{answer}\rangle \tag{1}$$

Here, $t_i$ represents the CoT reasoning process and $\hat{y}_i$ is the predicted answer. This structured output allows us to separately evaluate both the reasoning quality and the final answer correctness.

Reinforcement learning fine-tuning seeks to optimize the audio LLMs to maximize rewards:

$$\pi^* = \arg\max_\pi \mathbb{E}[R(s_i)], \tag{2}$$

where $s_i = (t_i, \hat{y}_i)$ represents the complete model output. Current approaches like R1-AQA and Ke-Omni-R employ outcome-only rewards based solely on answer correctness and format compliance: $R_{\text{RLVR}}(s_i) = \mathbb{I}[\hat{y}_i = y_i] + \mathbb{I}[\text{ValidFormat}(s_i)]$. This impoverished signal leads to three critical failure modes: (1) **Random Emergence** of reasoning patterns without effective control; (2) **Reasoning-Answer Inconsistency** where models generate answers inconsistent with their reasoning logic; (3) **Lack of Structured Reasoning** strategies like elimination or multi-step deduction.

Our fundamental insight is that genuine reasoning capability requires explicit process-oriented incentivization rather than spontaneous emergence. We achieve this through a multi-faceted reward suite that provides granular feedback on reasoning quality, consistency, and structure during training, transforming reasoning from an unpredictable phenomenon into a controllable, trainable skill.

## 3.2 FROM OUTCOME-BASED TO PROCESS-ORIENTED REASONING CONTROL

Current RLVR approaches fundamentally fail to distinguish between genuine reasoning and fortunate guessing, leading to random emergence of reasoning behaviors that cannot be systematically controlled or guaranteed. Our framework introduces a novel paradigm that transforms reasoning from an unpredictable emergent phenomenon into a controllable, trainable capability through comprehensive process supervision.

Our total reward $R_{\text{total}}(s_i)$ decomposes into two complementary components that address distinct aspects of reasoning quality:

$$\underbrace{\alpha_1 R_{\text{acc}}(s_i) + \alpha_2 R_{\text{format}}(s_i)}_{\text{Verifiable Rewards}} + \underbrace{\alpha_3 R_{\text{consistency}}(s_i) + \alpha_4 R_{\text{keywords}}(s_i) + \alpha_5 R_{\text{overthinking penalty}}(s_i)}_{\text{Reasoning Process Rewards}} \tag{3}$$

The verifiable rewards maintain essential correctness constraints and structural integrity, while our **Reasoning Process Rewards** explicitly shape reasoning quality, consistency, and conciseness. Here $s_i = (t_i, \hat{y}_i)$ represents the complete model output encompassing both reasoning trace and final answer, and $\{\alpha_j\}_{j=1}^5$ are weight coefficients that balance answer correctness with reasoning refinement. In practice, we set $\alpha_1 = 5.0$ for accuracy and $\alpha_{2-5} = 1.0$ for other components.

### 3.2.1 FOUNDATION: VERIFIABLE CORRECTNESS AND STRUCTURAL INTEGRITY

While transcending traditional RLVR limitations, our framework maintains rigorous grounding in verifiable outcomes. The **Accuracy Reward** $R_{\text{acc}}(s_i) = \mathbb{I}[\hat{y}_i = y_i]$ establishes the fundamental correctness constraint that ensures reasoning improvements do not come at the expense of answer accuracy. This binary signal prevents the optimization process from learning elaborate but incorrect reasoning patterns, anchoring all process improvements in empirical validity.

The **Format Reward** enforces structural compliance and prevents the model from bypassing the reasoning framework:

$$R_{\text{format}}(s_i) = \mathbb{I}[\text{ValidFormat}(s_i)] \tag{4}$$

This reward ensures the model produces outputs with proper XML tag structure, specifically requiring both $\langle\text{think}\rangle t_i \langle/\text{think}\rangle$ and $\langle\text{answer}\rangle \hat{y}_i \langle/\text{answer}\rangle$ components. This creates a disciplined reasoning environment where models must engage with the reasoning process rather than circumventing it through format violations.

### 3.2.2 Semantic Coherence and Reasoning-Answer Alignment

A critical challenge in current reasoning approaches is the pervasive problem of reasoning-answer inconsistency, where models generate correct answers despite fundamentally flawed or irrelevant reasoning processes. Additionally, when reasoning traces are unrelated to the question and choices, models are prone to hallucination. The **Reasoning Consistency Reward** introduces explicit semantic supervision that ensures reasoning traces genuinely support their corresponding conclusions:

$$R_{\text{consistency}}(s_i) = \text{Sim}_{\text{semantic}}(t_i, \hat{y}_i) + \text{Sim}_{\text{semantic}}(t_i, Q_i), \tag{5}$$

where $Q_i = (q_i, \mathcal{C}_i)$ represents the complete question context including both the question text and available choices. This dual-alignment formulation addresses two critical failure modes. The answer-alignment component $\text{Sim}_{\text{semantic}}(t_i, \hat{y}_i)$ prevents reasoning processes from becoming disconnected from their conclusions, which would render the reasoning ineffective. The question-alignment component $\text{Sim}_{\text{semantic}}(t_i, Q_i)$ ensures the reasoning remains focused on the posed question and available choices, preventing hallucination and off-topic elaboration.

We implement semantic similarity using concept overlap (e.g., via overlapped words):

$$\text{Sim}_{\text{semantic}}(x, y) = \frac{\text{ConceptOverlap}(x, y)}{\max(|\text{Concepts}(x)|, |\text{Concepts}(y)|)} \tag{6}$$

where the normalization ensures bounded similarity scores in $[0, 1]$. This approach represents a departure from outcome-only optimization, introducing explicit supervision signals that distinguish between reasoning processes that accidentally arrive at correct answers and those that systematically derive conclusions through valid analytical pathways.

### 3.2.3 Incentivizing Structured Reasoning and Penalizing Overthinking

To explicitly shape reasoning quality, our framework employs a two-pronged strategy: we positively **incentivize structured reasoning** while simultaneously **penalizing inefficient overthinking**. The primary mechanism for structured reasoning is the **Keywords Reward**, which acts as a cognitive scaffold to transform random emergent thoughts into controlled, sophisticated analytical behaviors:

$$R_{\text{keywords}}(s_i) = R_{\text{pattern}}(s_i) + R_{\text{logic}}(s_i) + R_{\text{domain}}(s_i) \tag{7}$$

This tri-component design addresses three fundamental aspects of structured reasoning: structured analytical patterns, logical rigor, and domain expertise integration.

**Structured Analytical Patterns.** The pattern recognition component systematically rewards models for developing structured reasoning architectures rather than relying on intuitive leaps: $R_{\text{pattern}}(s_i) = \sum_{p \in \mathcal{P}} \cdot \mathbb{I}[\text{Pattern}_p \text{ detected in } t_i]$. The pattern set $\mathcal{P}$ captures sophisticated reasoning architectures through key categories such as sequential organization, comparative analysis, systematic evaluation, and explicit justification. Complete pattern specifications are detailed in App. B.6.

**Logical Rigor and Causal Reasoning.** The reasoning indicators component cultivates sophisticated logical thinking by rewarding linguistic markers that indicate deep analytical processes: $R_{\text{logic}}(s_i) = \sum_{l \in \mathcal{L}} \cdot \mathbb{I}[\text{Keyword}_l \text{ detected in } t_i]$. The reasoning logic taxonomy $\mathcal{L}$ strategically targets distinct logical functions including formal deduction markers, premise establishment, hypothetical reasoning, and evidential conclusions. These linguistic signatures promote sophisticated logical progression from premises to conclusions (complete taxonomy in App. B.6).

**Domain Knowledge Integration.** We also incentivize the use of domain knowledge, where the domain component rewards models for incorporating audio-specific expertise rather than generic reasoning patterns: $R_{\text{domain}}(s_i) = \sum_{d \in \mathcal{D}} w_d \cdot \mathbb{I}[\text{Term}_d \text{ detected in } t_i]$. The domain vocabulary $\mathcal{D}$ encompasses specialized terminology across acoustic properties, musical concepts, speech analysis, and environmental audio understanding. This encourages models to ground their reasoning in signal-specific expertise rather than superficial pattern matching (complete vocabulary in App. B.6).

**Overthinking Penalty.** The necessary counterpart to rewarding structured thought is penalizing its inefficient opposite. The **Overthinking Penalty** addresses a critical failure mode: the tendency for models to engage in redundant, verbose reasoning that accumulates errors rather than improving

analysis quality. This component actively discourages overthinking by penalizing excessively long reasoning traces:

$$R_{\text{overthinking penalty}}(s_i) = f_{\text{length}}(|t_i|) = 1 - \frac{|t_i|}{L_{\text{max\_output}}} \tag{8}$$

where $f_{\text{length}}(l)$ is a linear penalty function that decreases as reasoning length $|t_i|$ increases, normalized by the maximum output length $L_{\text{max\_output}}$ (we set as 256 in practice). This design specifically targets common failure modes including circular reasoning, repetitive analysis, and tangential elaboration. By learning to terminate reasoning at an appropriate depth, models develop a meta-cognitive awareness that prevents hallucination accumulation while maintaining analytical rigor.

## 3.3 Cultivating Reasoning Capability via Online RL

Our framework operationalizes process-oriented reasoning control through Group Relative Policy Optimization (GRPO) (Shao et al., 2024), systematically cultivating reasoning capabilities rather than relying on random emergence. For each training sample $(a_i, q_i, \mathcal{C}_i, y_i)$, we sample $K$ responses $\{s_i^{(k)}\}_{k=1}^{K} \sim \pi_\theta(\cdot | a_i, q_i, \mathcal{C}_i)$ and optimize the objective:

$$\mathcal{L}_{\text{GRPO}} = \mathcal{L}_{PG}^{\text{multi-faceted}} + \beta \cdot \mathcal{L}_{KL}, \tag{9}$$

where the policy gradient loss $\mathcal{L}_{PG}^{\text{multi-faceted}} = -\mathbb{E}\left[\sum_{k=1}^{K} A(s^{(k)}) \cdot \log \pi_\theta(s^{(k)} | a, q, \mathcal{C})\right]$ provides granular feedback on both analytical processes and final outcomes. The advantage function $A(s_i^{(k)}) = R_{\text{total}}(s_i^{(k)}) - \frac{1}{K}\sum_{j=1}^{K} R_{\text{total}}(s_i^{(j)})$ enables models to distinguish between high-quality and low-quality reasoning processes, while KL regularization $\mathcal{L}_{KL} = \mathbb{E}\left[\text{KL}(\pi_\theta || \pi_{\text{ref}})\right]$ maintains training stability.

To enhance model robustness against linguistic variance, we employ systematic **data augmentation** that expands our training corpus $\mathcal{D}$ into an augmented version $\mathcal{D}'$ by generating multiple linguistic variations for each question while preserving ground-truth answers. For each instance $(a_i, q_i, \mathcal{C}_i, y_i) \in \mathcal{D}$, we apply answer-invariant transformation templates $\mathcal{T} = \{T_1, \ldots, T_M\}$, where each transformation $T_k$ generates $q'_{i,k} = T_k(q_i, \mathcal{C}_i)$, creating training samples $(a_i, q'_{i,k}, \mathcal{C}_i, y_i)$ with unchanged audio and answers. This forces the model to learn underlying reasoning patterns rather than superficial textual correlations. Complete template specifications are provided in App. B.

## 3.4 Unlocking Reasoning Capability via Test-Time Scaling

To understand the test-time inverse scaling phenomenon and validate our proposed methods, we introduce **Test-Time Scaling** to systematically analyze reasoning dynamics by evaluating performance across varying maximum thinking lengths $L_{\text{max\_think}}$. We define performance as $P(L_{\text{max\_think}}) = \mathbb{E}\left[\mathbb{I}[\hat{y} = y] \mid |t| \le L_{\text{max\_think}}\right]$ and identify the **"reasoning sweet spot"** where performance peaks: $L_{\text{sweet}} = \arg\max_L P(L)$. Through this simple scaling of reasoning length, CESAR achieves substantial improvements, with particularly dramatic gains at its reasoning sweet spot, while baseline models show limited improvement or continued degradation (See Fig. 3). This method effectively unlocks reasoning capability at test-time by revealing that our process-oriented training enables models to discover and utilize their optimal reasoning depth for maximum performance.

# 4 Experiments

## 4.1 Experimental Setup

We evaluate our framework on challenging out-of-distribution (OOD) audio reasoning benchmarks: MMAU Test-mini (Sakshi et al., 2025) with 1k expertly annotated questions spanning speech, sounds, and music requiring 27 distinct reasoning skills, and MMSU (Wang et al., 2025) with 5k audio-question pairs and granular perception-reasoning task separation. Training uses the AVQA dataset (Yang et al., 2022) enhanced through systematic data augmentation that generates diverse

Table 1: MMAU Test-Mini benchmark results. **Blue** indicates best performance, **green** indicates second-best. Accuracy (%) is reported across audio modalities. OP means overthinking penalty. See App. D.4 for details.

| Method | Reasoning | Sound | Music | Speech | Total Accuracy |
|---|---|---|---|---|---|
| **Our Proposed Methods** | | | | | |
| CESAR | ✓ | **83.48** | **73.05** | 74.77 | **77.10** |
| CESAR | ✗ | 79.88 | 67.96 | 73.27 | 73.70 |
| CESAR w/o OP | ✓ | **81.98** | 70.06 | **77.48** | **76.50** |
| CESAR w/o OP | ✗ | 80.48 | 70.06 | 74.47 | 75.00 |
| **RL Baseline Methods** | | | | | |
| Ke-Omni-R | ✓ | 79.28 | 70.06 | 74.47 | 74.60 |
| Ke-Omni-R | ✗ | 78.38 | **70.96** | 74.17 | 74.50 |
| **Proprietary Models** | | | | | |
| Gemini 2.5 Pro | - | 75.08 | 68.26 | 71.47 | 71.60 |
| Gemini 2.5 Flash | - | 73.27 | 65.57 | **76.58** | 71.80 |
| GPT-4o Audio | - | 64.56 | 56.29 | 66.67 | 62.50 |
| **Base Models** | | | | | |
| Qwen2.5-Omni-7B | ✓ | 69.07 | 59.58 | 66.97 | 65.20 |
| Qwen2.5-Omni-7B | ✗ | 72.37 | 64.37 | 69.07 | 68.60 |

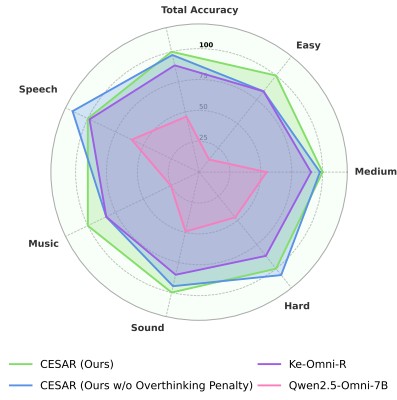

CESAR (Ours) — Ke-Omni-R
CESAR (Ours w/o Overthinking Penalty) — Qwen2.5-Omni-7B

Figure 2: Task-wise comparison on the MMAU Test-mini Benchmark (Scores are normalized by CESAR). See App. D.5 for more results.

Table 2: MMSU Results (Wang et al., 2025). Best scores are in blue, second-best in green. Results show accuracy (%) across perception and reasoning tasks. See App. D.6 for more results.

| Models | Perception Tasks | | | | Reasoning Tasks | | | | Overall |
|---|---|---|---|---|---|---|---|---|---|
| | Semantics | Phonology | Paralinguistics | Avg | Semantics | Phonology | Paralinguistics | Avg | |
| CESAR (Ours) | **60.16** | 50.16 | 39.50 | **48.45** | **88.72** | 80.66 | 57.01 | **81.07** | **64.24** |
| Ke-Omni-R | 58.74 | 46.31 | **40.50** | 47.09 | 86.82 | 74.31 | **60.00** | 78.06 | 62.08 |
| Gemini 1.5 Pro | 57.06 | **53.60** | 31.23 | 46.10 | 79.47 | **83.46** | 46.33 | 76.16 | 60.68 |
| Qwen2.5-Omni-7B | 55.12 | 37.33 | 39.35 | 42.50 | **88.00** | 81.37 | 48.36 | 79.83 | 60.57 |
| GPT-4o Audio | 59.70 | 41.56 | 21.44 | 39.67 | 80.83 | 78.74 | 26.25 | 71.96 | 56.38 |
| Human | **87.10** | **94.32** | **92.88** | **91.24** | 82.16 | **87.60** | **89.12** | **86.77** | **89.72** |

question phrasings while preserving answer labels. Our experiments employ Qwen2.5-Omni-7B with GRPO, sampling $K = 8$ responses per training example. Reward coefficients balance correctness ($\alpha_1 = 5.0$) with other rewards ($\alpha_{2-5} = 1.0$). We compare against base model variants, Ke-Omni-R baseline, proprietary models, and open-source audio models. Unless otherwise specified, all reported scores of our methods are achieved with reasoning. Complete details are in App. B.

## 4.2 MAIN RESULTS: STATE-OF-THE-ART PERFORMANCE ACROSS BENCHMARKS

**MMAU: Significant Performance Gains via Rewarding the Reasoning Process**   As shown in Tab. 1, our method establishes new SOTA performance on MMAU Test-mini, decisively surpassing leading proprietary models including GPT-4o Audio and Gemini 2.5 Pro. Most importantly, we demonstrate that process-oriented training delivers synergistic improvements across both reasoning modes: compared to the base model, CESAR achieves substantial gains both with reasoning and without reasoning, proving that cultivating reasoning processes fundamentally enhances the model's cognitive capabilities. Our framework also significantly outperforms outcome-only RL methods, with reasoning mode delivering larger benefits than the Ke-Omni-R baseline. The radar analysis (Fig. 2) reveals controllable reasoning architectures: our two variants exhibit engineered cognitive profiles—CESAR w/o OP excelling on hard tasks through deeper analysis, while full CESAR maintains balanced efficiency across difficulty levels—establishing reasoning as a systematically controllable capability rather than random emergence. See App. D.3, D.4 and D.5 for more results.

**MMSU: Perceptual Improvements with Near-Human Reasoning**   On the MMSU benchmark (Tab. 2), our CESAR achieves dual advances: reasoning capabilities that approach human levels (including super-human performance in semantic reasoning), while simultaneously outperforming larger competitors on perception tasks. This reveals an interesting synergistic effect where cultivating advanced reasoning through our reasoning process rewards also refines foundational auditory

Table 3: Performance on the MMAU-Pro Benchmark (Kumar et al., 2025). We compare CESAR against key baselines and SOTA models. Best scores are highlighted in `blue`, second-best scores in `green`. All values are accuracy (%) and rounded to one decimal place (same as MMAU Pro paper). See App. D.1 for more results.

| Model | Sound | Music | Speech | Sound–Music | Speech–Music | Speech–Sound | S–M–Speech | Spatial | Voice | Multi-Audio | Open-ended | IF | Average |
|---|---|---|---|---|---|---|---|---|---|---|---|---|---|
| **CESAR (Ours)** | 54.1 | 63.5 | 64.0 | 48.0 | 43.5 | 53.4 | 71.4 | 40.6 | 54.5 | 34.2 | 62.4 | 35.6 | 56.4 |
| Ke-Omni-R | 46.9 | 64.3 | 61.8 | 48.0 | 47.8 | 51.1 | 57.1 | 49.2 | 47.2 | 35.6 | 59.2 | 24.1 | 54.5 |
| Qwen2.5-Omni-7B (Base) | 43.1 | 55.6 | 54.2 | 32.0 | 45.7 | 46.6 | 28.6 | 37.2 | 51.0 | 33.3 | 58.4 | 31.0 | 49.1 |
| Gemini-2.5 Flash | 51.9 | 64.9 | 73.4 | 42.8 | 58.7 | 61.3 | 42.8 | 36.3 | 71.7 | 21.2 | 67.5 | 95.1 | 59.2 |
| Gemini-2.0 Flash | 48.4 | 56.9 | 69.5 | 39.6 | 57.6 | 55.9 | 42.8 | 34.6 | 68.6 | 26.5 | 66.8 | 94.2 | 55.7 |
| GPT-4o Audio | 44.7 | 63.1 | 68.2 | 40.4 | 43.5 | 62.5 | 57.1 | 21.4 | 57.5 | 32.6 | 43.2 | 82.5 | 52.5 |
| Audio Flamingo 3 | 55.9 | 61.7 | 58.8 | 40.0 | 41.3 | 47.7 | 57.1 | 26.8 | 58.6 | 26.0 | 44.2 | 33.3 | 51.7 |
| R1-AQA | 47.9 | 31.9 | 33.7 | 32.0 | 36.9 | 20.4 | 28.5 | 23.6 | 32.7 | 11.4 | 38.5 | 44.2 | 34.1 |
| Human | 78.2 | 70.5 | 82.3 | 79.3 | 78.5 | 82.4 | 85.7 | 88.2 | 68.4 | 79.8 | 77.3 | 100.0 | 77.9 |

perception capabilities. However, while both capabilities advance substantially, the results illuminate a critical asymmetry: reasoning improvements have reached near-human parity, whereas perception performance, despite leading existing models, still exhibits a considerable gap relative to human baselines. This disparity identifies the "perceptual bottleneck" as a key area for future work in achieving comprehensive human-level audio understanding. See App. D.6 for more results.

**MMAU-Pro: Robust Reasoning on a Challenging "In-the-Wild" Benchmark.** To further test the limits of our framework, we evaluate CESAR on the highly challenging MMAU-Pro benchmark (Kumar et al., 2025). This benchmark is specifically designed to defeat simple heuristics by using "in-the-wild" audio and complex, multi-hop reasoning tasks, including multi-audio analysis and open-ended questions. As summarized in Table 3, the results confirm CESAR's superiority. With an overall score of 56.4%, CESAR establishes itself as the **top-performing 7B-parameter model**, significantly surpassing other powerful 7B models and even larger audio LLMs like GPT-4o Audio and Audio Flamingo 3. This performance is highly competitive even with massive-scale proprietary models, trailing only Gemini 2.5 Flash among AI models. Crucially, CESAR substantially outperforms the outcome-only RL baseline, Ke-Omni-R. This advantage is particularly pronounced in reasoning-heavy categories such as 'Instruction Following' and 'Open-ended QA', providing direct evidence that our process-oriented rewards cultivate a more robust and generalizable reasoning capability. In many core audio-related reasoning tasks, CESAR's performance begins to close the gap on human-level capabilities, validating the effectiveness of our approach.

### 4.3 Curing Test-Time Inverse Scaling and Unlocking Scalable Reasoning

In Fig. 3, our Test-Time Scaling analysis reveals the **test-time inverse scaling** problem, where baseline models exhibit either a catastrophic performance collapse (under-optimized model) or volatile performance with no clear benefit from longer reasoning (standard RL baseline). In contrast, our methods resolve this issue, transforming reasoning from detriments into gains. As shown in Fig. 3 (Left), even without the overthinking penalty, our model's performance steadily climbs to a peak of 76.50%. Moreover, our full method demonstrates superior calibration; by explicitly penalizing inefficient thought, it discovers a more optimal **"reasoning sweet spot,"** achieving a higher peak accuracy of 77.1% with a much shorter reasoning chain of approximately 35-40 tokens. This proves our methods enable consistent, effective reasoning that unlocks scalable capability during inference to achieve performance gains through scaling reasoning lengths. See App. D.7 for more results.

### 4.4 AI-as-Judge Evaluation: Quantifying Reasoning Quality Beyond Accuracy

To move beyond accuracy and verify our improved reasoning, we introduce an AI-as-Judge for head-to-head comparisons via GPT-4o Audio. As shown in Fig. 3 (Right), our method's reasoning process achieves commanding win rates against both baselines. Notably, even without the Overthinking Penalty, our core rewards still yield a dominant performance, while its inclusion further elevates the win rate. This corroborates the superior performance of our full method in the MMAU (Tab. 1). These results provide direct evidence that our framework generates verifiably superior reasoning, a qualitative leap not captured by accuracy alone. See App. D.8 for more details and prompts used.

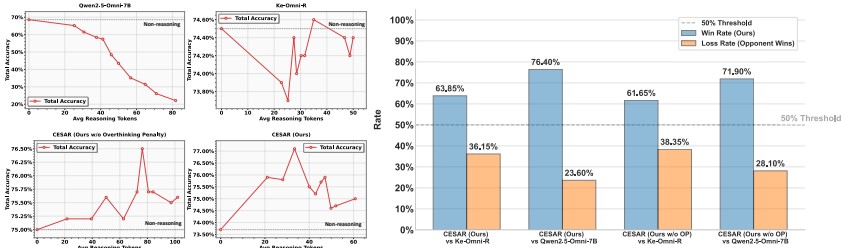

Figure 3: **Remediating Test-Time Inverse Scaling and Quantifying Reasoning Quality on MMAU Test-mini. (Left)** Test-time scaling analysis shows performance when increasing the reasoning tokens by sweeping different maximum thinking lengths from 0 to maximum output length (i.e., 250) in intervals of 25. **(Right)** An AI-as-judge evaluation with GPT-4o Audio (Hurst et al., 2024) provides quantitative proof of our superior reasoning quality, showing our models achieve commanding win rates against strong baselines. Throughout, OP denotes the overthinking penalty.

Table 4: Human Evaluation Win Rates (Majority-Vote Protocol). Results for the 1,000-sample MMAU Test-mini benchmark, based on over **3,000 individual human judgments** (3 annotators per sample). 'W', 'L', 'T' represent the win, lose (baseline win), and tie rates for CESAR. Each of the 1,000 final results shown was decided by the majority vote of the three expert annotators. See App. D.2 for more results and details.

| Model Comparison | Overall | | | Music | | | Sound | | | Speech | | |
|---|---|---|---|---|---|---|---|---|---|---|---|---|
| | **W** | **L** | **T** | **W** | **L** | **T** | **W** | **L** | **T** | **W** | **L** | **T** |
| **CESAR** vs. Qwen2.5-Omni (Base) | **88.60** | 6.60 | 4.80 | **88.62** | 6.29 | 5.09 | **87.69** | 7.21 | 5.11 | **89.49** | 6.31 | 4.20 |
| **CESAR** vs. Ke-Omni-R (RL Baseline) | **63.10** | 14.80 | 22.10 | **64.37** | 14.37 | 21.26 | **66.07** | 14.11 | 19.82 | **58.86** | 15.92 | 25.23 |

## 4.5 HUMAN EVALUATION: VALIDATING REASONING QUALITY WITH HUMAN JUDGEMENT

To obtain a definitive, human-level assessment of reasoning quality beyond automated metrics or AI judges, we conducted a large-scale human evaluation. This study was performed on the **entire 1k-sample MMAU Test-mini benchmark**. Each question was evaluated by three independent expert annotators, resulting in **over 3,000 individual human judgments**. Annotators were presented with reasoning traces from two anonymous models in a head-to-head comparison and asked to select the superior reasoning process based on logic, faithfulness to the audio, and consistency—crucially, evaluators remained blind to correct answers and model identities to eliminate bias.

The results, summarized in Table 4, are decisive. The data shows our method is **overwhelmingly preferred** by human experts. Against the base Qwen2.5-Omni-7B model, CESAR's reasoning process **completely dominates**, winning in 88.60% of overall cases. This domination is consistent across all audio modalities, demonstrating the profound impact of our RL framework. More importantly, CESAR also **decisively outperforms** the strong outcome-only RL baseline, Ke-Omni-R, with a 63.10% overall win rate. This latter finding is critical, providing clear, human-backed evidence that our **process-oriented rewards** (CESAR) cultivate a genuinely superior and more trustworthy reasoning capability than simply rewarding the final outcome (Ke-Omni-R).

## 4.6 QUALITATIVE ANALYSIS: CONCRETE EVIDENCE OF REASONING IMPROVEMENTS

Beyond quantitative improvements, our methods also produce superior reasoning processes. Tab. 5 illustrates two critical failure modes that our process-oriented rewards address. In the military parade example, our model systematically analyzes acoustic cues ("marching, music, and shouting") to reach the correct conclusion, while Qwen2.5-Omni-7B misinterprets organized sounds as "chaotic noise" and makes unfounded inferences about violence. The telephone counting example reveals an even more fundamental issue: reasoning-answer inconsistency, where Ke-Omni-R correctly identifies "three rings" in its reasoning trace but inexplicably outputs "2" as the final answer. Our consistency reward explicitly prevents such disconnects between reasoning processes and conclusions, ensuring that correct reasoning translates to correct answers. See App. D.9 for more results.

Table 5: Qualitative Comparison of Reasoning Processes in MMAU Test-Mini. Our method demonstrates consistent and well-structured reasoning processes. **Green text** highlights cultivated structured reasoning, while *red italics* mark reasoning errors. Complete qualitative analysis in App. D.9.

| | |
|---|---|
| **Example: Logical Deduction from Acoustic Cues** 
 **Question:** What kind of event could be taking place in the audio? 
 **Correct Answer:** A military parade with a marching band | |
| **CESAR (Correct)** | **Qwen2.5-Omni-7B (Wrong)** |
| **Reasoning:** *The presence of marching,* **music**, *and* **shouting suggests** *a large-scale event. The options are a quiet library, a heated argument, a military parade, and a gentle breeze. A military parade* **fits the description best**, *as it involves a marching band and organized* **shouting**. 

 **Answer:** A military parade with a marching band | **Reasoning:** *The audio contains the sound of a man speaking, followed by a loud,* **chaotic noise.** *The chaotic noise could be interpreted as a violent escalation, so a heated argument is the most likely scenario.* 

 **Answer:** A heated argument escalating to violence |
| **Example: Superior Reasoning-Answer Consistency** 
 **Question:** How many times does the telephone ring in the audio? 
 **Correct Answer:** 3 | |
| **CESAR (Correct)** | **Ke-Omni-R (Wrong)** |
| **Reasoning:** *Considering the options*, *the telephone* **rings** *three times. The* **audio** *clearly shows three distinct* **rings**. 
 **Answer:** 3 | **Reasoning:** *The telephone rings* **three times**, *as indicated by the three distinct ring sounds in the audio.* 
 **Answer:** 2 |

## 4.7 ABLATION STUDY: QUANTIFYING COMPONENT CONTRIBUTIONS

Table 6: Progressive ablation study on MMAU Test-mini. We start from the full CESAR method and progressively remove components to isolate their individual contributions. All scores are obtained using reasoning at their reasoning sweet spots. See App. D.10 for more results.

| Method | RL Training | Consistency | Keywords | Data Augmentation | Overthinking Penalty | Overall Accuracy (%) |
|---|---|---|---|---|---|---|
| **Full Method (CESAR)** | ✓ | ✓ | ✓ | ✓ | ✓ | **77.10** |
| Ablating Overthinking Penalty | ✓ | ✓ | ✓ | ✓ | ✗ | 76.50 |
| Ablating Data Augmentation | ✓ | ✓ | ✓ | ✗ | ✗ | 76.20 |
| Ablating Keywords | ✓ | ✓ | ✗ | ✗ | ✗ | 75.20 |
| Ablating Consistency (Ke-Omni-R) | ✓ | ✗ | ✗ | ✗ | ✗ | 74.60 |
| Ablating RL Training (Base Model) | ✗ | ✗ | ✗ | ✗ | ✗ | 65.20 |

Our progressive ablation study (Tab. 6) systematically deconstructs the components of our method. The results confirm the necessity of RL, as its removal triggers a catastrophic performance collapse. Building upon this RL foundation, our process-oriented rewards demonstrate strong synergy. The *Keywords* reward yields the largest single gain over the outcome-only RL baseline (Ke-Omni-R) by sculpting higher-quality, structured reasoning processes. The *Consistency* reward also provides a crucial boost by bridging the critical gap between a model's reasoning and its final output. The final components, *Data Augmentation* and the *Overthinking Penalty*, provide the necessary robustness and calibration to achieve peak performance. Ultimately, the ablation study demonstrates a clear synergistic effect: while each component provides a quantifiable and crucial performance gain, it is their holistic integration within the CESAR framework that unlocks state-of-the-art performance.

## 5 CONCLUSION

In this paper, we introduce CESAR to address the test-time inverse scaling problems in Audio LLMs, where CoT reasoning degrades performance due to inadequate optimization of reasoning processes in existing SFT and RLVR methods. Our methods shift from outcome verification to rewarding the reasoning process transforms reasoning from detriments into significant performance gains through GRPO with multi-faceted process rewards. We achieve SOTA results on massive benchmarks, surpassing GPT-4o Audio and Gemini 2.5 Pro, while demonstrating that test-time scaling is a double-edged sword—catastrophic for poorly trained models but enabling substantial gains through discovered "reasoning sweet spots" for models with strong reasoning capabilities like CESAR. Our comprehensive evaluation across multiple OOD benchmarks reveals synergistic effects where enhanced reasoning improves both multimodal reasoning and perception capabilities, while our AI-as-judge evaluations and qualitative comparisons provide both quantitative and qualitative validation of our improved reasoning quality beyond accuracy. CESAR establishes a principled methodology for developing robust, scalable reasoning in Audio LLMs. See App. A.3 for details on our LLM usage.

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

## A    DISCUSSION

In this paper, we explore how to cultivate robust, scalable, and effective reasoning in Audio Large Language Models. Despite the widespread success of chain-of-thought (CoT) reasoning in domains such as mathematics and coding (Comanici & et al., 2025; Jaech et al., 2024; DeepSeek-AI et al., 2025; Shao et al., 2024), efforts to introduce reasoning capabilities into Audio Large Language Models (Balaji et al., 2023; Ghosh et al., 2025; KimiTeam et al., 2025) have encountered a central paradox: CoT, a reliable catalyst for reasoning in text, consistently fails in the audio domain. While numerous works have attempted to leverage CoT prompting to enhance audio LLM reasoning and understanding capabilities (Ma et al., 2025a; Xie et al., 2025), several studies including R1-AQA (Li et al., 2025) have discovered that incorporating reasoning mechanisms not only fails to improve performance but may actually harm it.

Our systematic investigation reveals a more profound issue that we term *test-time inverse scaling* in Audio LLMs—a phenomenon we are the first to systematically diagnose as a test-time problem where prompting a model to "think" during inference yields worse results than instinctual, direct answering. When we scale state-of-the-art open-source models such as Qwen2.5-Omni-7B during test-time, their performance counterintuitively degrades as reasoning length increases, often falling below their non-reasoning baselines (Fig. 3). Similar patterns emerge in Ke-Omni-R (Zhao et al., 2025), where reasoning during inference frequently underperforms direct answering approaches. **This test-time inverse scaling manifests in two critical failure modes: (1) any Audio LLM exhibiting worse performance when reasoning is enabled compared to direct answering (as observed in Qwen2.5-Omni-7B and most cases of Ke-Omni-R in Tab. 1), and (2) progressive performance degradation as reasoning chain length increases during test-time (as demonstrated in Fig. 6).**

Our investigation reveals that this test-time inverse scaling is not a fundamental limitation of reasoning itself, but a symptom of inadequate training: the models possess poor reasoning capability because they have never been properly *taught how* to reason. Our research fundamentally reframes this challenge, moving it from a problem of pattern memorization to one of controllable skill development. We demonstrate that effective reasoning is not an unpredictable emergent phenomenon, but a trainable capability that can be systematically cultivated by directly rewarding the reasoning *process*, thereby transforming reasoning from a liability into a systematic advantage for audio understanding.

> **Key Insight:** The failure of reasoning in Audio LLMs stems not from a fundamental limitation of the models, but from a flawed training paradigm. True reasoning capability is unlocked by shifting focus from supervising outcomes to directly rewarding the intrinsic quality of the reasoning *process*.

Existing methods are hamstrung by this flawed paradigm. Supervised fine-tuning produces brittle mimics, while contemporary reinforcement learning approaches, with their myopic focus on final-answer correctness, inadvertently reinforce the very flaws—inconsistency, hallucination, and unstructured thought—that cause reasoning to fail. Our work pioneers an approach centered on reasoning process rewards, using a multi-faceted reward suite to transform reasoning from a random liability into a reliable asset. The following findings chart a new course for the field.

> **Key Finding 1:** Test-Time inverse scaling should be reframed not as a fundamental law, but as a diagnostic signal for flawed reasoning processes. This issue is fully solvable with process-oriented supervision.

Our analysis provides a definitive diagnosis for why unguided reasoning is so detrimental. As vividly demonstrated by the base Qwen2.5-Omni-7B model's catastrophic performance collapse (from 68.60% down to 65.20% in Tab. 18), allowing an under-optimized model to "think" longer provides more opportunities for logical errors and hallucinations to compound. Our framework proves this is not an immutable property. By explicitly rewarding internal consistency, CESAR directly targets the root cause of this degradation, resulting in a complete reversal of the phenomenon. This finding suggests that readers encountering test-time inverse scaling should treat it as a clear

signal that a model's reasoning process requires direct, granular intervention, shifting focus from outcome-only rewards to the quality of the cognitive process itself.

> **Key Finding 2:** Reasoning can be transformed from an unpredictable emergent property into a controllable and engineerable skill, whose quality can be quantitatively measured beyond simple task accuracy.

A critical question for any RL method is whether the agent is truly learning a skill or simply exploiting the reward. Our work offers two contributions here. Our multi-faceted rewards, particularly those incentivizing structured and logical patterns (App. D.9), act as a cognitive scaffold to guide the model toward desired analytical behaviors. To validate that this guidance cultivates a genuine skill, our AI-as-judge evaluation provides quantitative proof of superior reasoning quality. The commanding win rates of CESAR introduce a valuable and scalable methodology for the field, enabling researchers to move beyond accuracy to rigorously evaluate the thought process itself. Reasoning, therefore, no longer needs to be a matter of chance; it can become a matter of design.

> **Key Finding 3:** The optimal reasoning budget is not universal but model-specific. This "reasoning sweet spot" can be unlocked at inference time, but only after a robust reasoning process has been cultivated during training.

Our introduction of test-time scaling reveals that the value of increased computation is entirely conditional on the quality of the learned policy. For the base model, more computation is actively harmful; for the outcome-only RL model, it yields volatile gains. In stark contrast, because CESAR has learned a coherent reasoning process—calibrated in part by the 'Overthinking Penalty'—test-time scaling becomes a powerful, practical optimization lever. It allows us to identify a distinct performance peak—a "reasoning sweet spot"—that other models cannot reach. This establishes a critical principle: a model must first learn to *think well* before *thinking more* becomes beneficial. This insight naturally leads to a two-stage best practice for practitioners: first, cultivate robust reasoning through process-oriented training, and then employ test-time scaling as an efficient, training-free strategy to identify the model's optimal computational budget at inference.

> **Key Finding 4:** Cultivating deliberate, step-by-step reasoning creates a powerful synergistic uplift, enhancing both a model's intuitive answering and its foundational perception.

This finding reveals a deep connection between different modes of cognition. The rigorous process of learning to reason forces the model to organize its understanding of the world more effectively. This enhanced internal representation sharpens its "fast," intuitive thinking, evidenced by a massive 5.1% improvement in its direct-answering capability over the base model (73.70% vs. 68.60%). The benefits even cascade to the sensory level, improving foundational perception scores on the MMSU benchmark. Better thinking, it turns out, leads to better hearing.

> **Key Finding 5:** By elevating reasoning to near-human levels, our work acts as a powerful diagnostic for the field, revealing that the primary barrier to progress is a foundational perceptual bottleneck.

Perhaps our most significant contribution is diagnostic: by successfully addressing high-level reasoning, our work brings the next major barrier into sharp focus. On the MMSU benchmark (Tab. 2), CESAR achieves near-human and even super-human reasoning capabilities (e.g., 88.72% vs. human 82.16% in Semantic Reasoning). This very success allows us to clearly identify the next great challenge. The remaining performance gap to humans can be confidently attributed to a different layer of the system: foundational perception, where our model (48.45%) still lags far behind human acuity (91.24%). Our work thus transforms the research landscape, providing a clear, data-driven direction to solve this perceptual bottleneck.

## A.1 Limitations

Our investigation also sheds light on several limitations, including a fundamental challenge for the field and method-specific considerations for future work.

**The Perceptual Bottleneck.** The primary limitation we identify is a foundational **perceptual bottleneck** affecting all current models. This issue is paradoxically highlighted by our own model's success; our results on the MMSU benchmark reveal a stark asymmetry where CESAR achieves super-human reasoning capabilities (e.g., 88.72% in semantic reasoning) while its foundational perception still significantly lags behind human acuity (48.45% vs. 91.24%). This demonstrates that even with near-perfect reasoning, a model's performance is ultimately capped by its ability to perceive a high-fidelity representation of the acoustic world. Resolving this is a critical next step for the entire field.

**Computational Requirements.** The GRPO-based training regimen, which requires sampling multiple responses for each input during online optimization, is computationally intensive. One standard training run of ours requires significant GPU resources, and this computational overhead, while justified by the substantial performance gains, may present a barrier to adoption for research groups with limited hardware resources.

**Hyperparameter Tuning.** Introducing a multi-faceted reward suite inevitably brings the challenge of hyperparameter optimization, specifically in balancing the weights of each reward component. We took steps to mitigate this complexity, for instance by normalizing each reward signal to a consistent [0, 1] range. Furthermore, through empirical investigation, we discovered that giving a higher weight to the accuracy reward while keeping other process-oriented rewards equally weighted yielded the best results. This suggests a potential curriculum learning effect: the model first prioritizes optimizing for accuracy—the most direct path to significant reward gains—and then, upon reaching a performance plateau, begins to refine its policy based on the more nuanced signals from the reasoning process rewards. We believe this is a valuable practical insight and encourage readers applying similar multi-reward frameworks to experiment with prioritizing the primary accuracy reward to guide the initial stages of policy optimization.

## A.2 FUTURE WORKS

Our work establishes a principled approach to building robust, controllable reasoning in Audio LLMs, addressing the test-time inverse scaling problem that has plagued the field. Having demonstrated that process-oriented training can reliably improve reasoning capabilities, several promising research directions emerge.

**The Perceptual Bottleneck Problem.** With reasoning capabilities now approaching human levels, our results reveal that perceptual limitations have become the primary constraint on overall performance. The audio encoders used in current systems appear to be the main bottleneck preventing further progress. This suggests that developing more sophisticated audio representations—perhaps through self-supervised learning or novel architectural innovations—should be a priority for the community. Our improved reasoning capabilities provide a clear benchmark for evaluating whether perceptual improvements translate to better end-to-end performance.

**Cross-Modal Applications.** The success of process-oriented training in audio raises questions about its broader applicability. Testing whether similar principles work for vision, robotics, or other modalities would help determine if we've uncovered domain-specific insights or more general principles of machine reasoning. Early experiments applying our framework to visual question answering or robotic planning could provide valuable insights into the universality of process-oriented approaches.

### A.3 THE USE OF LARGE LANGUAGE MODELS (LLMS)

In accordance with the conference guidelines, we acknowledge the use of Large Language Models (LLMs) during the preparation of this manuscript. We utilized LLMs for paper writing assistance, specifically for language polishing and improving the clarity and readability of our work. The LLMs assisted in refining linguistic expression, ensuring proper grammar and academic writing style, and enhancing the overall flow of technical content.

All core research contributions, including the novel methodology, experimental design, theoretical analysis, and scientific insights presented in this work, were developed independently by the authors. The LLMs were used solely as writing assistance tools (i.e., for polishing writing) and did not contribute to the conceptual development, experimental validation, or interpretation of results.

## A.4 ETHICS

Our work aims to enhance multimodal reasoning capabilities in audio LLMs without introducing any additional ethical concerns or resolving existing ones.

# B    EXPERIMENTAL DETAILS

## B.1    BASELINE METHODS

To validate the superiority of our approach, we compare it against a comprehensive set of baselines that represent different training paradigms and model classes.

**Base Model**    Our foundational model is **Qwen2.5-Omni-7B**, a powerful, unified end-to-end multimodal model capable of perceiving diverse inputs including audio, video, and images, and generating both text and speech responses (Xu et al., 2025). We evaluate it in two distinct modes: direct-answering (zero-shot) and CoT-prompted. This crucial comparison allows us to empirically diagnose the test-time inverse scaling problem: by contrasting the performance of a powerful but under-optimized reasoner with and without a reasoning process, we can isolate the performance degradation caused by unguided "thinking" and establish a clear baseline from which to measure the absolute gains provided by our RL framework.

**RL Baseline**    Our most direct competitor is **Ke-Omni-R** (Zhao et al., 2025), the current state-of-the-art audio reasoning model that shares the same **Qwen2.5-Omni-7B** base architecture and is also trained using the GRPO algorithm. This makes it the perfect control group for our study. However, Ke-Omni-R relies on a simpler Reinforcement Learning from Verifiable Rewards (RLVR) setup, where rewards are based solely on the correctness of the final answer within a concise reasoning trace of fewer than 50 words (Zhao et al., 2025). This comparison therefore serves as a direct ablation of our novel, multi-faceted reward suite. By contrasting our process-oriented approach with Ke-Omni-R's outcome-only paradigm, we can effectively measure the performance ceiling of existing RL methods and demonstrate the significant improvements unlocked by rewarding the reasoning process itself.

**Other Models**    To situate our work in the broader landscape, we also report scores from other leading models. This includes top-performing proprietary systems such as the Gemini series (Comanici & et al., 2025) and GPT-4o Audio (Hurst et al., 2024), which represent the state-of-the-art in closed-source multimodal AI. Furthermore, we compare against a wide range of open-source audio LLMs that are primarily trained using supervised fine-tuning (SFT) on CoT datasets, such as Audio-Reasoner (Xie et al., 2025). This comprehensive comparison ensures that our results are contextualized against the full spectrum of current approaches, from powerful proprietary APIs to various SFT-based methods. For comprehensive details on these models, we refer the reader to their original papers and the benchmark papers (Wang et al., 2025; Ma et al., 2025b; Sakshi et al., 2025).

## B.2 EVALUATION BENCHMARKS

To rigorously validate the generalization capabilities of our framework, we conduct a comprehensive evaluation on several distinct, challenging, and entirely **out-of-distribution (OOD)** audio understanding benchmarks. None of the audio clips, questions, or underlying tasks in these benchmarks overlap with our training corpus (AVQA). This strict separation ensures that our evaluation measures genuine, transferable reasoning skill, rather than task-specific memorization or reward hacking, thereby providing a true test of our model's ability to reason in novel acoustic scenarios.

**MMAU Test Mini** We selected the **MMAU (Massive Multi-Task Audio Understanding and Reasoning Benchmark)** test-mini split (Sakshi et al., 2025) as our principal testbed due to its unparalleled breadth and focus on expert-level cognition. Comprising approximately 1000 expertly annotated questions, the benchmark is systematically distributed across the three core audio domains: speech, environmental sounds, and music (Sakshi et al., 2025). Its design explicitly targets 27 distinct cognitive skills, which are divided into information extraction and complex reasoning categories (Sakshi et al., 2025). The significant challenge of MMAU stems from its demand for expert-level, domain-specific knowledge—such as identifying musical chord progressions or decoding phonological sequences—combined with sophisticated reasoning that moves far beyond simple perception. This comprehensive and demanding nature makes it the ideal environment to validate the general and versatile reasoning capabilities cultivated by CESAR, and explains why our framework achieves state-of-the-art performance on this benchmark.

**MMSU** For a granular, diagnostic analysis of spoken language understanding, we utilize the **MMSU (Massive Multi-task Spoken Language Understanding and Reasoning Benchmark)** (Wang et al., 2025), which serves as a surgical tool for dissecting the relationship between high-level cognition and low-level perception. Containing 5,000 audio-question pairs across 47 distinct tasks grounded in established linguistic theory (Wang et al., 2025), its unique value lies in the formal bifurcation of all tasks into foundational *Perception* (e.g., identifying falling tones) and higher-level *Reasoning* (e.g., interpreting sarcasm from prosodic cues) (Wang et al., 2025). This explicit separation is strategically vital, as it allows us to provide clear, quantitative evidence for our key discovery: CESAR's ability to achieve near human-level performance on the *Reasoning* tasks validates the effectiveness of our training paradigm. Simultaneously, the significant gap that remains on *Perception* tasks, despite some synergistic improvement, provides definitive proof of the "perceptual bottleneck," clarifying a critical direction for future research.

**MMAR** To stress-test our model's reasoning capabilities under the most demanding conditions, we include an evaluation on **MMAR (A Challenging Benchmark for Deep Reasoning)** (Ma et al., 2025b), a benchmark specifically designed to probe the limits of deep, multi-step, and compositional reasoning. Its 1,000 tasks are uniquely characterized by longer audio clips (averaging 20 seconds (Ma et al., 2025b)) and complex, real-world *mixed-modality* audio, where overlapping sources like speech, background music, and sound effects must be disentangled (Ma et al., 2025b). The primary difficulty of MMAR lies in its demand for sustained temporal reasoning and the ability to perform multi-hop inferences on composite acoustic scenes, a task that often requires graduate-level domain knowledge (Ma et al., 2025b). We chose MMAR to prove that the reasoning skills cultivated by CESAR are not brittle but robust and scalable. By succeeding here, we demonstrate that our framework builds a durable cognitive capability that holds up under extreme complexity, providing powerful, supplementary evidence of our model's advanced reasoning prowess, with detailed results presented in App. D.11.

**MMAU Full Test Set** To validate scalability, we extend our evaluation to the complete **MMAU Full Test Set**, comprising approximately 9,000 audio question-answering samples across speech, sound, and music modalities (Sakshi et al., 2025). This substantially larger evaluation corpus provides comprehensive assessment of our framework's performance and demonstrates that our advantages hold at scale. Detailed results are presented in App. D.3.

**MMAU-Pro** For an even more demanding test, we evaluate on **MMAU-Pro** (Kumar et al., 2025), a challenging benchmark with 5,305 expert-annotated instances designed to probe the limits of audio reasoning. MMAU-Pro features "in-the-wild" audio, longer clips (averaging 20 seconds), multi-

audio reasoning tasks, spatial audio understanding, and open-ended question answering. Its design explicitly minimizes language priors and requires genuine audio-grounded reasoning, making it an ideal testbed for validating that our process-oriented training cultivates robust reasoning capabilities that generalize beyond the training distribution. Detailed results are presented in App. D.1.

### B.3 TRAINING DATA AND AUGMENTATION STRATEGY

**Training Data.** The primary training corpus for CESAR is the **AVQA** dataset. To ensure a fair comparison, our main RL baseline, Ke-Omni-R, also uses AVQA as its foundation. However, it is crucial to note that Ke-Omni-R supplements its training with the specialist **MusicBench** dataset. Despite not using this in-domain music data, CESAR still outperforms Ke-Omni-R on the music tasks of the MMAU benchmark (73.05% vs. 70.06%). This provides strong evidence that cultivating a general, robust reasoning process enhances multimodal generalization, allowing the model to effectively transfer its learned analytical skills to specialized domains even without explicit in-domain training data.

**Systematic Data Augmentation via Question Rephrasing.** To enhance model robustness and prevent the learning of superficial textual correlations, we employ a systematic data augmentation scheme. This method expands our training corpus by generating multiple linguistic variations for each question while preserving the ground-truth answer, thereby compelling the model to learn the underlying reasoning task rather than shallow text patterns. Formally, for each instance $(a_i, q_i, \mathcal{C}_i, y_i) \in \mathcal{D}$, we apply a set of answer-invariant transformation templates $\mathcal{T} = \{T_1, \ldots, T_M\}$. Each transformation $T_k$ generates a new question $q'_{i,k} = T_k(q_i, \mathcal{C}_i)$, creating a new training sample $(a_i, q'_{i,k}, \mathcal{C}_i, y_i)$.

Our approach uses simple but effective template-based transformations that reframe questions to target specific reasoning capabilities. For instance, an original question like "What are the main sources of sound in this video?" with choices [motorboat, bus, train, truck] is transformed using capability-specific templates:

- **Temporal Reasoning:** "Which sound source appears most prominently in the temporal sequence: {choices}?"
- **Counting Tasks:** "Which option has the highest occurrence frequency among: {choices}?"
- **Comparative Analysis:** "Which sound demonstrates the strongest relationship with other audio elements: {choices}?"

This strategy systematically expands training diversity, forcing the model to develop generalizable reasoning skills that contribute directly to its robust performance.

### B.4 TRAINING HYPERPARAMETERS AND PROMPTING CONFIGURATION

**Training Pipeline and Hyperparameters.** Our training pipeline is built upon the Qwen2.5-Omni-7B model, which we fine-tune using **Group Relative Policy Optimization (GRPO)** (Shao et al., 2024). To ensure a fair comparison and isolate the impact of our proposed reasoning process rewards, our core GRPO hyperparameters (e.g., KL coefficient $\beta$, batch size, learning rate) are kept consistent with those of the RLVR baseline, Ke-Omni-R (Zhao et al., 2025). This approach prioritizes methodological clarity and reproducibility. For a detailed breakdown of these specific hyperparameter values, we refer the reader to the original Ke-Omni-R work (Zhao et al., 2025). The optimization process uses the **AdamW** optimizer with a learning rate of **1e-5** and a global batch size of **32**, sampling $K = 8$ responses per input for each GRPO step.

**Inference Prompts** We adopt the prompt template directly from our primary baseline, Ke-Omni-R, to ensure a fair and direct comparison. This shared template instructs the model to follow a strict, two-part output format, namely: (1) *Generating Reasoning Traces within* `<think>...</think>` *(less than {max_think_len} words)*, and subsequently (2) *Generating the Final Answer within* `<answer>...</answer>`. While the prompt structure is shared, the crucial distinction—and a core component of our methodology—lies in its application during evaluation. Whereas Ke-Omni-R reports performance at a fixed, static reasoning length, we leverage the `{max_think_len}` parameter to perform our test-time scaling analysis (see Sec. D.7). By systematically evaluating the model across the full spectrum of values, we are able to not only demonstrate robustness against the test-time inverse scaling problem but also to identify the optimal, model-specific "reasoning sweet spots" that unlock peak performance, providing a much richer understanding of a model's true capabilities.

**AI-as-Judge Evaluation Prompts.** To quantitatively assess reasoning quality, we employed an AI-as-judge framework using a SOTA multimodal LLM (GPT-4o Audio). The evaluation prompt instructed the judge to perform a head-to-head comparison between the reasoning traces of two models. The judge's decision was guided by specific criteria, including logical coherence, faithfulness to acoustic evidence, and the overall soundness of the analytical path, with a focus on the process rather than just the final answer's correctness. The complete prompt and detailed methodology are provided in App. D.8.

**Reward Configuration** To ensure a fair and direct comparison with the RL baseline, we align our core GRPO training parameters with those of Ke-Omni-R. The critical distinction lies in our reward configuration. After exploring various hyperparameter settings, we identified a simple yet remarkably effective weighting scheme for the components in equation 3: the accuracy reward weight ($\alpha_1$) is set to **5.0**, while the weights for all other process-oriented rewards (consistency, keywords, overthinking penalty) are set to **1.0**. This configuration maintains a strong optimization pressure towards generating correct final answers, while the process rewards act as crucial regularizers and fine-grained guides. They shape the reasoning trajectories without overpowering the primary objective of correctness. As substantiated by our ablation study (App. D.10), the thoughtful *design* of these reward functions, rather than their specific weightings, is the primary driver of performance, demonstrating the robustness of our overall framework.

### B.5 COMPUTATIONAL RESOURCES

All reinforcement learning experiments were performed on a high-performance computing cluster equipped with 8 NVIDIA H200 GPUs, each providing 141GB of HBM3e memory. A standard training run for our final model on the augmented AVQA dataset concluded in approximately 61.44 hours on this infrastructure.

### B.6 Detailed Keywords

The **Keywords Reward** ($R_{\text{keywords}}$) is a central component of our process-oriented supervision framework, engineered to guide the model toward generating reasoning traces that are structured, logical, and domain-aware. This reward is calculated as a composite score that aggregates signals from three distinct categories: structured analytical patterns, logical rigor indicators, and domain-specific terminology. To implement this, we programmatically scan each generated reasoning trace for the presence of specific keywords and patterns. The detection mechanism employs a combination of **simple string matching** for exact phrases (e.g., `considering the options`, `is consistent with`) and **regular expressions** for more flexible patterns (e.g., numbered lists like `1.`, `2.`). Each detected term or pattern from our predefined taxonomies contributes positively to the final reward score, thereby explicitly incentivizing the model to construct more sophisticated and coherent reasoning processes. The comprehensive taxonomies of these keywords and phrases, broken down by their function, are detailed in Tables 7, 8, and 9.

Table 7: Keywords for Structured Analytical Patterns ($R_{\text{pattern}}$).

| Category | Description | Example Keywords / Phrases |
|---|---|---|
| Sequential Organization | Indicates a step-by-step analytical process or temporal ordering. | `first`, `second`, `then`, `next`, `finally`, `step 1`, `1.`, `2.` |
| Comparative Analysis | Phrases used for comparing and contrasting different options or ideas. | `rather than`, `compared to`, `in contrast to`, `on the other hand` |
| Systematic Evaluation | Suggests a methodical review and elimination of the provided choices. | `considering the options`, `evaluating each choice`, `among the options` |
| Explicit Justification | Language that directly justifies the selection of the final answer. | `most suitable`, `the best fit`, `fits the description best` |

Table 8: Keywords for Logical Rigor & Causal Reasoning ($R_{\text{logic}}$).

| Category | Description | Example Keywords / Phrases |
|---|---|---|
| Premise & Deduction | Establishes a logical premise and draws a conclusion from it. | `given`, `based on`, `since`, `therefore`, `thus`, `hence`, `so` |
| Evidential Support | Links acoustic evidence from the audio signal to an inference. | `indicates`, `suggests`, `is consistent with`, `as evidenced by` |
| Hypothetical Reasoning | Terms used for suppositions or stating general principles. | `assume`, `suppose`, `typically`, `generally`, `it is likely that` |

Table 9: Keywords for Domain Knowledge Integration ($R_{\text{domain}}$).

| Category | Description | Example Keywords / Phrases |
|---|---|---|
| Acoustic Properties | Basic terminology related to the physical properties of sound. | `sound`, `audio`, `noise`, `pitch`, `volume`, `timbre`, `rhythm`, `frequency` |
| Environmental & Animal Sounds | Vocabulary for specific non-speech, non-music sound events. | `bell`, `ring`, `hooves`, `engine`, `siren`, `animal`, `clip-clop`, `moo` |
| Musical Concepts | Specialized terminology for analyzing musical content. | `chord`, `note`, `melody`, `harmony`, `instrument`, `major`, `minor`, `P5` |
| Speech Analysis | Terms used to describe and analyze human vocal characteristics. | `voice`, `speech`, `tone`, `intonation`, `male`, `female`, `shouting`, `whisper` |

## B.7 REPRODUCIBILITY

We have made comprehensive efforts to ensure reproducibility of our work. Our complete methodology is detailed in Section 3, with step-by-step algorithmic implementation provided in Appendix C. All experimental configurations are thoroughly documented in Section 4, with hyperparameter settings specified in Section B.4. As Appendix B, our training pipeline builds upon the open-source codebase (i.e., Ke-Omni-R (Zhao et al., 2025)) using publicly available base models (i.e., Qwen2.5-Omni-7B (Xu et al., 2025)) and training datasets. Data augmentation procedures are described in Section B.3. Evaluation benchmarks are all publicly available. Additional implementation details, including computational requirements and reward function specifications, are provided in Appendix B. All source code and trained models will be made publicly available upon publication to facilitate reproducibility and future research.

## C  ALGORITHM PSEUDOCODE

In this section, we provide the detailed pseudocode for the CESAR framework. Algorithm 1 outlines the main online reinforcement learning loop using Group Relative Policy Optimization (GRPO). To enhance clarity, we use the superscript 'ex' (e.g., $\mathcal{L}^{ex}_{GRPO}$) to denote a value calculated for a single training *example*, distinguishing it from values aggregated over an entire mini-batch. Algorithm 2 then specifies the computation of our multi-faceted, process-oriented reward, which is central to cultivating robust reasoning capabilities.

---

**Algorithm 1** CESAR Training via Group Relative Policy Optimization (GRPO)

---

1: **Require:** Audio LLM policy $\pi_\theta$ to be fine-tuned, reference policy $\pi_{ref}$.
2: **Require:** Training dataset $\mathcal{D} = \{(a_i, q_i, \mathcal{C}_i, y_i)\}_{i=1}^{N}$.
3: **Require:** Number of samples per input $K$.
4: **Require:** Reward weights $\{\alpha_j\}_{j=1}^{5}$, learning rate $\eta$, KL regularization weight $\beta$.
5: Initialize policy parameters $\theta$ from a pre-trained Audio LLM.
6: **for** each training iteration **do**
7:     Sample a mini-batch $B = \{(a, q, \mathcal{C}, y)\}$ from $\mathcal{D}$.
8:     Initialize gradients $\nabla_\theta \mathcal{L} \leftarrow 0$.
9:     **for** each training example $(a, q, \mathcal{C}, y)$ in $B$ **do**
10:         // Step 1: Sample K responses from the current policy $\pi_\theta$.
11:         Sample a set of $K$ responses $\mathcal{S} = \{s^{(k)} = (t^{(k)}, \hat{y}^{(k)})\}_{k=1}^{K} \sim \pi_\theta(\cdot|a, q, \mathcal{C})$.
12:         // Step 2: Calculate the total reward for each of the K responses.
13:         Initialize a rewards list $R \leftarrow []$.
14:         **for** $k = 1$ to $K$ **do**
15:             $R^{(k)}_{total} \leftarrow$ CALCULATETOTALREWARD$(s^{(k)}, y, q, \mathcal{C})$         ▷ See Algorithm 2
16:             Append $R^{(k)}_{total}$ to $R$.
17:         **end for**
18:         // Step 3: Compute the advantage using the mean reward as a baseline.
19:         $\bar{R} \leftarrow \frac{1}{K} \sum_{j=1}^{K} R^{(j)}_{total}$.
20:         Initialize policy gradient loss for the example $\mathcal{L}^{ex}_{PG} \leftarrow 0$.
21:         **for** $k = 1$ to $K$ **do**
22:             $A(s^{(k)}) \leftarrow R^{(k)}_{total} - \bar{R}$.         ▷ Advantage of response $k$
23:             $\mathcal{L}^{ex}_{PG} \leftarrow \mathcal{L}^{ex}_{PG} - A(s^{(k)}) \log \pi_\theta(s^{(k)}|a, q, \mathcal{C})$.
24:         **end for**
25:         // Step 4: Calculate the full loss and accumulate gradients.
26:         $\mathcal{L}_{KL} \leftarrow \mathbb{E}_{\pi_\theta} \left[ \log \frac{\pi_\theta(\cdot|a,q,\mathcal{C})}{\pi_{ref}(\cdot|a,q,\mathcal{C})} \right]$.
27:         $\mathcal{L}^{ex}_{GRPO} \leftarrow \frac{1}{K}\mathcal{L}^{ex}_{PG} + \beta \cdot \mathcal{L}_{KL}$.
28:         Accumulate gradients: $\nabla_\theta \mathcal{L} \leftarrow \nabla_\theta \mathcal{L} + \nabla_\theta \mathcal{L}^{ex}_{GRPO}$.
29:     **end for**
30:     // Step 5: Update the policy parameters.
31:     $\theta \leftarrow \theta - \eta \cdot \frac{1}{|B|} \nabla_\theta \mathcal{L}$.
32: **end for**
33: **return** Optimized policy parameters $\theta$.

---

---

**Algorithm 2** Multi-Faceted Reward Calculation

---

1: **function** CALCULATETOTALREWARD$(s, y, q, \mathcal{C})$
2:     **Input:** A single response $s = (t, \hat{y})$, ground-truth answer $y$, question $q$, choices $\mathcal{C}$.
3:     **Input:** Reward weights $\{\alpha_j\}_{j=1}^{5}$.
4:     // — 1. Verifiable Rewards —
5:     $R_{\text{acc}} \leftarrow \mathbb{I}[\hat{y} = y]$.                                                       ▷ Accuracy
6:     $R_{\text{format}} \leftarrow \mathbb{I}[\text{ValidFormat}(s)]$.                     ▷ XML structure compliance
7:     // — 2. Reasoning Process Rewards —
8:     $Q \leftarrow (q, \mathcal{C})$.                                               ▷ Full question context
9:     $R_{\text{consistency}} \leftarrow \text{Sim}_{\text{semantic}}(t, \hat{y}) + \text{Sim}_{\text{semantic}}(t, Q)$.       ▷ Semantic alignment
10:     $R_{\text{pattern}} \leftarrow$ CALCULATEKEYWORDSCORE$(t, \text{PatternKeywords})$.       ▷ See Table 7
11:     $R_{\text{logic}} \leftarrow$ CALCULATEKEYWORDSCORE$(t, \text{LogicKeywords})$.         ▷ See Table 8
12:     $R_{\text{domain}} \leftarrow$ CALCULATEKEYWORDSCORE$(t, \text{DomainKeywords})$.      ▷ See Table 9
13:     $R_{\text{keywords}} \leftarrow R_{\text{pattern}} + R_{\text{logic}} + R_{\text{domain}}$.
14:     $R_{\text{overthinking}} \leftarrow 1 - \frac{\text{length}(t)}{L_{\text{max\_output}}}$.                    ▷ Penalty for verbosity
15:     // — 3. Compute Total Weighted Reward —
16:     $R_{\text{total}} \leftarrow \alpha_1 R_{\text{acc}} + \alpha_2 R_{\text{format}} + \alpha_3 R_{\text{consistency}} + \alpha_4 R_{\text{keywords}} + \alpha_5 R_{\text{overthinking}}$.
17:     **return** $R_{\text{total}}$.
18: **end function**

---

# D ADDITIONAL EXPERIMENTAL RESULTS

## D.1 BENCHMARK RESULTS ON MMAU-PRO

To further validate the robustness and generalization of our process-oriented reward framework, we conduct an extended evaluation on the highly challenging **MMAU-Pro benchmark** (Kumar et al., 2025). MMAU-Pro is a comprehensive testbed for holistic audio intelligence, meticulously designed to evaluate models on complex, realistic auditory scenarios that are explicitly underserved by existing benchmarks. It consists of 5,305 expert-annotated instances, where questions are designed to require deliberate, multi-hop reasoning. Its audio is sourced directly "from the wild" to prevent data contamination and test true generalization. Critically, it introduces novel tasks that directly probe the limits of current models, including long-form audio, multi-audio reasoning, spatial audio, complex mixtures, open-ended QA, and instruction following. The benchmark is explicitly designed to minimize reliance on "language priors" and demand "genuine audio-grounded reasoning," making it an ideal testbed to validate our central thesis.

Table 10: Performance on the MMAU-Pro Benchmark. Best scores are highlighted in blue , second-best scores in green . All values are accuracy (%). All results show accuracy (%). Human performance is included as an upper bound reference. We report the performance of Ke-Omni-R (Zhao et al., 2025) and Qwen2.5-Omni-7B (Xu et al., 2025) from our own reproductions under the same protocol; all other baseline results are taken from the MMAU Pro paper (Kumar et al., 2025).

| Model | Sound | Music | Speech | Sound–Music | Speech–Music | Speech–Sound | S–M–Speech | Spatial | Voice | Multi-Audio | Open-ended | IF | Average |
|---|---|---|---|---|---|---|---|---|---|---|---|---|---|
| **Our Proposed Method** | | | | | | | | | | | | | |
| CESAR (Ours) | 54.1 | 63.5 | 64.0 | 48.0 | 43.5 | 53.4 | 71.4 | 40.6 | 54.5 | 34.2 | 62.4 | 35.6 | **56.4** |
| **Audio RL Baseline** | | | | | | | | | | | | | |
| Ke-Omni-R | 46.9 | 64.3 | 61.8 | 48.0 | 47.8 | 51.1 | 57.1 | 49.2 | 47.2 | 35.6 | 59.2 | 24.1 | 54.5 |
| **Base Model** | | | | | | | | | | | | | |
| Qwen2.5-Omni-7B (Base) | 43.1 | 55.6 | 54.2 | 32.0 | 45.7 | 46.6 | 28.6 | 37.2 | 51.0 | 33.3 | 58.4 | 31.0 | 49.1 |
| **Large Audio Language Models** | | | | | | | | | | | | | |
| GPT-4o Audio | 44.7 | 63.1 | 68.2 | 40.4 | 43.5 | 62.5 | 57.1 | 21.4 | 57.5 | 32.6 | 43.2 | 82.5 | 52.5 |
| Audio Flamingo 3 | 55.9 | 61.7 | 58.8 | 40.0 | 41.3 | 47.7 | 57.1 | 26.8 | 58.6 | 26.0 | 44.2 | 33.3 | 51.7 |
| GPT-4o-mini-Audio | 40.2 | 59.7 | 66.1 | 35.3 | 42.2 | 55.9 | 42.8 | 12.0 | 52.7 | 22.4 | 41.6 | 79.7 | 48.3 |
| Kimi-Audio | 46.0 | 57.6 | 52.2 | 46.0 | 54.3 | 48.9 | 42.8 | 43.7 | 50.6 | 17.2 | 34.5 | 42.3 | 46.6 |
| Audio Flamingo 2 | 39.5 | 55.7 | 43.0 | 36.0 | 34.8 | 29.5 | 14.8 | 44.1 | 37.2 | 15.5 | 43.2 | 29.6 | 42.6 |
| DeSTA2.5-Audio | 35.7 | 48.2 | 49.9 | 22.0 | 36.9 | 35.2 | 28.6 | 28.0 | 51.0 | 19.8 | 36.4 | 46.5 | 40.6 |
| Gemma-3n-E4B-it | 42.4 | 46.4 | 44.9 | 38.0 | 45.6 | 31.8 | 57.1 | 21.8 | 58.3 | 19.6 | 28.5 | 36.4 | 39.7 |
| SALMONN 13B | 43.6 | 47.2 | 37.3 | 28.0 | 47.8 | 38.4 | 42.8 | 30.8 | 53.2 | 17.4 | 33.6 | 38.5 | 39.6 |
| Phi4-MM | 25.7 | 47.8 | 47.6 | 30.0 | 39.1 | 30.1 | 28.6 | 39.7 | 42.7 | 11.4 | 42.5 | 65.4 | 38.7 |
| DeSTA2 | 31.0 | 43.3 | 46.5 | 32.6 | 47.8 | 39.7 | 42.8 | 32.6 | 54.8 | 13.2 | 25.4 | 41.5 | 36.7 |
| Gemma-3n-E2B-it | 40.1 | 44.1 | 41.3 | 26.0 | 33.2 | 30.6 | 28.6 | 12.0 | 51.4 | 11.4 | 23.2 | 29.6 | 35.4 |
| SALMONN 7B | 32.2 | 44.9 | 38.3 | 22.0 | 34.8 | 28.4 | 14.8 | 26.5 | 36.5 | 11.4 | 31.2 | 33.9 | 34.5 |
| GAMA | 45.4 | 41.2 | 29.8 | 24.0 | 27.9 | 27.3 | 14.8 | 12.0 | 28.4 | 20.2 | 24.2 | 31.7 | 33.2 |
| **Large Audio Reasoning Models** | | | | | | | | | | | | | |
| Audio-Reasoner | 34.2 | 50.1 | 44.0 | 26.0 | 36.9 | 43.2 | 28.6 | 20.3 | 43.4 | 22.6 | 38.6 | 43.4 | 39.5 |
| R1-AQA | 47.9 | 31.9 | 33.7 | 32.0 | 36.9 | 20.4 | 28.5 | 23.6 | 32.7 | 11.4 | 38.5 | 44.2 | 34.1 |
| Mellow | 27.6 | 32.9 | 27.9 | 24.0 | 34.8 | 27.3 | 14.3 | 23.7 | 28.3 | 20.8 | 21.4 | 23.5 | 27.5 |
| **Omni Models** | | | | | | | | | | | | | |
| Gemini-2.5 Flash | 51.9 | 64.9 | 73.4 | 42.8 | 58.7 | 61.3 | 42.8 | 36.3 | 71.7 | 21.2 | 67.5 | 95.1 | 59.2 |
| Gemini-2.0 Flash | 48.4 | 56.9 | 69.5 | 39.6 | 57.6 | 55.9 | 42.8 | 34.6 | 68.6 | 26.5 | 66.8 | 94.2 | 55.7 |
| Ming-Lite-Omni-1.5 | 47.9 | 56.2 | 49.1 | 30.0 | 39.1 | 45.4 | 42.8 | 31.7 | 44.5 | 37.4 | 42.7 | 48.2 | 47.4 |
| Baichuan-Omni-1.5 | 34.6 | 32.5 | 36.5 | 30.0 | 19.5 | 30.7 | 28.5 | 21.2 | 40.0 | 28.8 | 39.7 | 47.2 | 33.9 |
| **Cascaded Systems** | | | | | | | | | | | | | |
| Caption + GPT-4o | 38.6 | 40.6 | 38.4 | 21.6 | 38.2 | 25.5 | 28.6 | 9.5 | 38.6 | 24.7 | 27.6 | 88.2 | 35.3 |
| Captions + Qwen235B-A22B | 36.4 | 41.3 | 36.1 | 18.6 | 37.4 | 24.5 | 14.3 | 5.8 | 35.6 | 22.5 | 25.6 | 85.5 | 33.7 |
| **Baselines** | | | | | | | | | | | | | |
| Human | 78.2 | 70.5 | 82.3 | 79.3 | 78.5 | 82.4 | 85.7 | 88.2 | 68.4 | 79.8 | 77.3 | 100.0 | 77.9 |
| Random Choice | 28.3 | 26.1 | 29.4 | 24.2 | 25.2 | 30.5 | 14.8 | 21.2 | 29.3 | 25.2 | — | — | 23.4 |

**Analysis of Results.** Our evaluation on the MMAU-Pro benchmark validates the significant advantages of our process-oriented reward framework. As shown in Table 10, CESAR achieves an overall average score of **56.4%**, establishing it as the highest-performing model in the 7B category. This performance not only represents a substantial absolute improvement over its base model (Qwen2.5-Omni-7B), but also a clear gain over Ke-Omni-R, the outcome-only RL baseline. This directly confirms that rewarding the reasoning *process*—including consistency and structure—builds a more robust and capable model than rewarding the *result* alone.

CESAR's average score is highly competitive, surpassing other prominent audio LLMs like Audio Flamingo 3 and powerful proprietary models such as GPT-4o Audio. Its performance ranks just below that of ultra-large-scale proprietary models like Gemini 2.5 Flash, demonstrating that a 7B model with superior reasoning training can effectively challenge models many times its size.

This strong average score is driven by superior performance on genuine audio-related tasks, where CESAR's reasoning capabilities begin to close the gap with human-level performance. The comparison against our Ke-Omni-R baseline is particularly insightful. Compared to baselines like Qwen2.5-Omni-7B (Xu et al., 2025) and Ke-Omni-R (Zhao et al., 2025), CESAR shows dramatic gains in reasoning-heavy tasks that demand structure and coherence, such as 'Open-ended QA', and the highly complex 'Sound–Music–Speech' mixture task. This demonstrates a superior ability to disentangle complex acoustic scenes and formulate structured responses. Furthermore, CESAR shows broad improvements across foundational reasoning in 'Sound', 'Speech', and 'Voice' tasks, confirming the wide-ranging benefits of our approach in cultivating a more genuine and generalizable audio reasoning capability.

### D.2 HUMAN EVALUATION: VALIDATING REASONING QUALITY THROUGH EXPERT JUDGMENT

To provide robust validation of our reasoning quality improvements beyond automated metrics, we conduct a comprehensive human evaluation study comparing CESAR's reasoning processes against two baselines: the base Qwen2.5-Omni-7B model and the Ke-Omni-R baseline. This evaluation directly assesses the quality of the *reasoning process* itself (i.e., reasoning capability).

#### D.2.1 HUMAN EVALUATION SETUP AND METHODOLOGY

**Data Collection and Preparation.** We collect reasoning traces and final answers from different models (CESAR, Qwen2.5-Omni-7B, and Ke-Omni-R) on all questions in the MMAU Test-mini benchmark. Each model generates its thinking process along with the final answer for each question. These model outputs are then prepared for human evaluation through careful anonymization and randomization.

**Task Design.** Human evaluators are presented with audio clips from the MMAU Test-mini benchmark along with questions and answer choices. For each question, evaluators review the reasoning traces and final answers generated by two models (presented in randomized order as "Model 1" and "Model 2") and select which model demonstrates superior reasoning capability. **Critically, evaluators are kept blind to the correct answers** to eliminate potential bias—this ensures that judgments are based purely on reasoning quality rather than being influenced by answer correctness. Evaluators are also blind to which model produced which reasoning trace. Evaluators are explicitly instructed to focus on the quality of the thinking process, assessing four key dimensions: (1) **Audio Understanding** - whether the model correctly perceives the acoustic content; (2) **Logic** - whether the reasoning follows a coherent, step-by-step progression relevant to the question; (3) **Clarity** - whether the explanation is easy to follow; and (4) **Consistency** - whether the reasoning aligns with the final answer.

**Evaluation Protocol.** We employ a rigorous multi-annotator protocol to ensure accuracy and fairness. **Each question in MMAU Test-mini is evaluated by three independent expert judges**, eliminating single-annotator bias and providing robust consensus. We report results using two aggregation methods: (1) **Per-Vote** evaluation, where each individual judgment from all three annotators is counted separately, providing fine-grained insight into reasoning quality across all evaluations (yielding 3× the number of questions in total judgments); and (2) **Majority-Vote** evaluation, where the final judgment for each question is determined by the majority decision among the three annotators, representing a more conservative consensus-based assessment. This dual-reporting approach ensures both comprehensive coverage and robust validation of our findings.

**Coverage.** Our evaluation spans the full diversity of the MMAU Test-mini benchmark, encompassing all three audio modalities (Sound, Music, Speech), three difficulty levels (Easy, Medium, Hard), and 27 distinct reasoning sub-categories, ensuring comprehensive assessment across the entire spectrum of audio reasoning tasks.

#### D.2.2 MAIN HUMAN EVALUATION RESULTS: COMMANDING WIN RATES VALIDATE SUPERIOR REASONING QUALITY

The human evaluation results provide decisive empirical evidence that CESAR cultivates verifiably superior reasoning processes. Tab. 11 and Tab. 12 present the aggregate results using per-vote and majority-vote protocols respectively. From the human perspective, CESAR's reasoning processes are consistently judged as superior to both baselines across all evaluation scenarios. Against the base Qwen2.5-Omni-7B model, human evaluators demonstrate an overwhelming preference for CESAR's reasoning, with this preference strengthening further under the conservative majority-vote protocol. Even more critically, when compared against the strong Ke-Omni-R baseline—which also employs reinforcement learning but with outcome-only rewards—CESAR maintains a clear and consistent advantage. This validates our central hypothesis: rewarding the reasoning *process* yields qualitatively superior reasoning compared to optimizing solely for final answer correctness.

Table 11: Human Evaluation Results - Overall Performance (Per-Vote Protocol). Each individual judgment from three annotators per question is counted. Best win rates are highlighted in **blue** . All values are win rates (%).

| Category | CESAR vs. Qwen2.5-Omni | | | CESAR vs. Ke-Omni-R | | |
|---|---|---|---|---|---|---|
| | CESAR | Baseline | Tie | CESAR | Baseline | Tie |
| **Overall** | **79.07** | 16.30 | 4.63 | **58.47** | 25.77 | 15.77 |
| By Audio Modality | | | | | | |
| Music | **78.84** | 16.47 | 4.69 | **59.18** | 25.25 | 15.57 |
| Sound | **78.88** | 16.32 | 4.80 | **61.06** | 25.33 | 13.61 |
| Speech | **79.48** | 16.12 | 4.40 | **55.16** | 26.73 | 18.12 |
| By Difficulty Level | | | | | | |
| Easy | **76.64** | 18.30 | 5.06 | **59.82** | 25.45 | 14.73 |
| Medium | **80.37** | 15.37 | 4.26 | **57.22** | 26.60 | 16.17 |
| Hard | **78.39** | 16.53 | 5.08 | **60.03** | 24.15 | 15.82 |

Table 12: Human Evaluation Results - Overall Performance (Majority-Vote Protocol). The final judgment for each question is determined by the majority decision among three annotators. Best win rates are highlighted in **blue** . All values are win rates (%).

| Category | CESAR vs. Qwen2.5-Omni | | | CESAR vs. Ke-Omni-R | | |
|---|---|---|---|---|---|---|
| | CESAR | Baseline | Tie | CESAR | Baseline | Tie |
| **Overall** | **88.60** | 6.60 | 4.80 | **63.10** | 14.80 | 22.10 |
| By Audio Modality | | | | | | |
| Music | **88.62** | 6.29 | 5.09 | **64.37** | 14.37 | 21.26 |
| Sound | **87.69** | 7.21 | 5.11 | **66.07** | 14.11 | 19.82 |
| Speech | **89.49** | 6.31 | 4.20 | **58.86** | 15.92 | 25.23 |
| By Difficulty Level | | | | | | |
| Easy | **87.50** | 7.14 | 5.36 | **66.07** | 14.73 | 19.20 |
| Medium | **89.07** | 6.85 | 4.07 | **60.19** | 16.11 | 23.70 |
| Hard | **88.56** | 5.51 | 5.93 | **66.95** | 11.86 | 21.19 |

**Robustness Across Modalities and Difficulty Levels.** The breakdown by audio modality (Music, Sound, Speech) and difficulty level (Easy, Medium, Hard) in Tab. 11 and Tab. 12 reveals the remarkable robustness of our reasoning improvements. From the human evaluators' perspective, CESAR consistently demonstrates superior reasoning across all categories, with only minor variations. This consistency is particularly significant given that evaluators were blind to the correct answers, confirming that the preference for CESAR's reasoning stems from genuine quality improvements rather than correlation with answer correctness. Notably, against Ke-Omni-R, our advantages are particularly pronounced in Sound tasks and Hard tasks, suggesting that process-oriented rewards are especially beneficial for challenging reasoning scenarios requiring nuanced acoustic analysis.

### D.2.3    FINE-GRAINED ANALYSIS: REASONING QUALITY ACROSS TASK SUB-CATEGORIES

To provide deeper insight into where and how our reasoning improvements manifest, we present detailed breakdowns across all 27 reasoning sub-categories in the MMAU benchmark. Tables 13 through 16 demonstrate that from the human evaluators' perspective, CESAR's reasoning advantages are not confined to specific task types but rather represent a broad, systematic improvement in reasoning capability across the entire spectrum of audio understanding challenges.

Table 13: Human Evaluation Results by Sub-Category: CESAR vs. Qwen2.5-Omni-7B (Per-Vote Protocol). All values are win rates (%). Best scores are highlighted in  blue .

| Sub-Category | CESAR | Qwen2.5-Omni-7B | Tie |
|---|---|---|---|
| Acoustic Scene Reasoning | **82.64** | 15.28 | 2.08 |
| Acoustic Source Inference | **80.56** | 13.19 | 6.25 |
| Ambient Sound Interpretation | **71.53** | 20.83 | 7.64 |
| Conversational Fact Retrieval | **81.82** | 15.15 | 3.03 |
| Counting | **83.91** | 16.09 | 0.00 |
| Dissonant Emotion Interpretation | **89.52** | 6.67 | 3.81 |
| Eco-Acoustic Knowledge | **78.72** | 17.02 | 4.26 |
| Emotion Flip Detection | **80.00** | 20.00 | 0.00 |
| Emotion State Summarisation | **79.55** | 16.67 | 3.79 |
| Emotional Tone Interpretation | **81.82** | 14.14 | 4.04 |
| Event-Based Knowledge Retrieval | **82.83** | 14.14 | 3.03 |
| Event-Based Sound Reasoning | **83.33** | 9.72 | 6.94 |
| Harmony and Chord Progressions | **71.72** | 23.23 | 5.05 |
| Instrumentation | **77.14** | 19.05 | 3.81 |
| Key Highlight Extraction | **87.30** | 6.35 | 6.35 |
| Lyrical Reasoning | **70.00** | 23.33 | 6.67 |
| Melodic Structure Interpretation | **79.80** | 14.14 | 6.06 |
| Multi-Speaker Role Mapping | **72.84** | 18.52 | 8.64 |
| Musical Genre Reasoning | **75.49** | 19.61 | 4.90 |
| Musical Texture Interpretation | **81.37** | 13.73 | 4.90 |
| Phonemic Stress Pattern Analysis | **72.96** | 22.01 | 5.03 |
| Phonological Sequence Decoding | **73.47** | 19.05 | 7.48 |
| Rhythm and Tempo Understanding | **86.23** | 6.52 | 7.25 |
| Socio-Cultural Interpretation | **85.00** | 10.00 | 5.00 |
| Sound-Based Event Recognition | **73.91** | 24.64 | 1.45 |
| Temporal Event Reasoning | **81.25** | 13.89 | 4.86 |
| Temporal Reasoning | **75.60** | 22.62 | 1.79 |

**Systematic Improvements Across All Reasoning Types.**    The sub-category analysis reveals several critical insights. First, from the human evaluators' perspective, CESAR's reasoning is consistently preferred across virtually all sub-categories in both comparisons (against Qwen2.5-Omni-7B and Ke-Omni-R), demonstrating systematic superiority rather than task-specific advantages. Second, the improvements are particularly pronounced in categories requiring complex, multi-step reasoning. For instance, against Qwen2.5-Omni, CESAR achieves exceptional preference in tasks such as "Dissonant Emotion Interpretation" and "Key Highlight Extraction"—categories that demand sophisticated understanding of nuanced acoustic cues and their semantic implications. Third, even against the strong Ke-Omni-R baseline, CESAR maintains substantial advantages in challenging categories such as "Ambient Sound Interpretation" and "Emotion Flip Detection", demonstrating that process-oriented rewards are especially effective for tasks requiring detection of subtle changes and contextual understanding.

Table 14: Human Evaluation Results by Sub-Category: CESAR vs. Qwen2.5-Omni-7B (Majority-Vote Protocol). All values are win rates (%). Best scores are highlighted in blue .

| Sub-Category | CESAR | Qwen2.5-Omni | Tie |
|---|---|---|---|
| Acoustic Scene Reasoning | 89.58 | 6.25 | 4.17 |
| Acoustic Source Inference | 91.67 | 4.17 | 4.17 |
| Ambient Sound Interpretation | 81.25 | 8.33 | 10.42 |
| Conversational Fact Retrieval | 90.91 | 4.55 | 4.55 |
| Counting | 93.10 | 6.90 | 0.00 |
| Dissonant Emotion Interpretation | 97.14 | 2.86 | 0.00 |
| Eco-Acoustic Knowledge | 85.11 | 10.64 | 4.26 |
| Emotion Flip Detection | 90.00 | 10.00 | 0.00 |
| Emotion State Summarisation | 88.64 | 4.55 | 6.82 |
| Emotional Tone Interpretation | 93.94 | 6.06 | 0.00 |
| Event-Based Knowledge Retrieval | 90.91 | 6.06 | 3.03 |
| Event-Based Sound Reasoning | 91.67 | 0.00 | 8.33 |
| Harmony and Chord Progressions | 81.82 | 12.12 | 6.06 |
| Instrumentation | 82.86 | 11.43 | 5.71 |
| Key Highlight Extraction | 95.24 | 4.76 | 0.00 |
| Lyrical Reasoning | 80.00 | 0.00 | 20.00 |
| Melodic Structure Interpretation | 93.94 | 3.03 | 3.03 |
| Multi-Speaker Role Mapping | 85.19 | 7.41 | 7.41 |
| Musical Genre Reasoning | 85.29 | 8.82 | 5.88 |
| Musical Texture Interpretation | 88.24 | 5.88 | 5.88 |
| Phonemic Stress Pattern Analysis | 84.91 | 9.43 | 5.66 |
| Phonological Sequence Decoding | 85.71 | 6.12 | 8.16 |
| Rhythm and Tempo Understanding | 93.48 | 0.00 | 6.52 |
| Socio-Cultural Interpretation | 95.00 | 0.00 | 5.00 |
| Sound-Based Event Recognition | 84.78 | 13.04 | 2.17 |
| Temporal Event Reasoning | 89.58 | 8.33 | 2.08 |
| Temporal Reasoning | 87.50 | 8.93 | 3.57 |

### D.2.4 KEY INSIGHTS AND IMPLICATIONS

The human evaluation provides decisive validation of our central thesis through three critical findings:

**Process-Oriented Training Cultivates Human-Preferred Reasoning.** The overwhelming and consistent preference for CESAR's reasoning across all comparisons demonstrates that explicitly rewarding the reasoning process results in qualitative improvements that are immediately recognizable to human experts. Crucially, these preferences emerged under blind evaluation conditions where annotators had no knowledge of correct answers, confirming that the superior reasoning quality is intrinsic to CESAR's thinking process rather than an artifact of answer correctness correlation. This validates our approach of moving beyond outcome-only optimization.

**The Limitations of Outcome-Only Reinforcement Learning.** The substantial gap between CESAR and Ke-Omni-R—despite both employing reinforcement learning on the same base model and training data—provides causal evidence that rewarding answer correctness alone is insufficient. From the human evaluators' perspective, CESAR consistently produces better reasoning processes across all tasks, demonstrating that high-quality reasoning requires explicit process-level supervi-

Table 15: Human Evaluation Results by Sub-Category: CESAR vs. Ke-Omni-R (Per-Vote Protocol). All values are win rates (%). Best scores are highlighted in blue .

| Sub-Category | CESAR | Ke-Omni-R | Tie |
|---|---|---|---|
| Acoustic Scene Reasoning | **59.03** | 33.33 | 7.64 |
| Acoustic Source Inference | **49.31** | 38.89 | 11.81 |
| Ambient Sound Interpretation | **72.22** | 20.14 | 7.64 |
| Conversational Fact Retrieval | **50.00** | 37.88 | 12.12 |
| Counting | **44.83** | 35.63 | 19.54 |
| Dissonant Emotion Interpretation | **52.38** | 25.71 | 21.90 |
| Eco-Acoustic Knowledge | **63.12** | 16.31 | 20.57 |
| Emotion Flip Detection | **71.67** | 13.33 | 15.00 |
| Emotion State Summarisation | **53.79** | 25.76 | 20.45 |
| Emotional Tone Interpretation | **65.66** | 23.23 | 11.11 |
| Event-Based Knowledge Retrieval | **51.52** | 26.26 | 22.22 |
| Event-Based Sound Reasoning | **63.19** | 19.44 | 17.36 |
| Harmony and Chord Progressions | **63.64** | 24.24 | 12.12 |
| Instrumentation | **60.00** | 23.81 | 16.19 |
| Key Highlight Extraction | **57.14** | 31.75 | 11.11 |
| Lyrical Reasoning | **66.67** | 13.33 | 20.00 |
| Melodic Structure Interpretation | **54.55** | 31.31 | 14.14 |
| Multi-Speaker Role Mapping | **50.62** | 17.28 | 32.10 |
| Musical Genre Reasoning | **57.84** | 25.49 | 16.67 |
| Musical Texture Interpretation | **60.78** | 21.57 | 17.65 |
| Phonemic Stress Pattern Analysis | **62.26** | 23.27 | 14.47 |
| Phonological Sequence Decoding | **56.46** | 30.61 | 12.93 |
| Rhythm and Tempo Understanding | **57.25** | 23.19 | 19.57 |
| Socio-Cultural Interpretation | **63.33** | 25.00 | 11.67 |
| Sound-Based Event Recognition | **57.25** | 22.46 | 20.29 |
| Temporal Event Reasoning | **63.19** | 26.39 | 10.42 |
| Temporal Reasoning | **53.57** | 30.36 | 16.07 |

sion targeting consistency, logical structure, and analytical depth. The three-annotator consensus protocol ensures these findings are robust and not dependent on individual annotator preferences.

**Robustness and Generalization of Reasoning Quality.**    The consistency of improvements across all audio modalities, difficulty levels, and 27 distinct reasoning sub-categories demonstrates that our framework cultivates genuinely robust reasoning skills rather than task-specific heuristics. The blind evaluation protocol—where evaluators judge reasoning quality without knowledge of correct answers—eliminates potential biases and confirms that CESAR's advantages reflect fundamental improvements in reasoning capability. This comprehensive validation across the full spectrum of audio understanding challenges, combined with rigorous multi-annotator evaluation, establishes CESAR as a principled approach to building controllable, high-quality reasoning in multimodal AI systems.

Table 16: Human Evaluation Results by Sub-Category: CESAR vs. Ke-Omni-R (Majority-Vote Protocol). All values are win rates (%). Best scores are highlighted in  blue .

| Sub-Category | CESAR | Ke-Omni-R | Tie |
|---|---|---|---|
| Acoustic Scene Reasoning | **62.50** | 25.00 | 12.50 |
| Acoustic Source Inference | **47.92** | 33.33 | 18.75 |
| Ambient Sound Interpretation | **79.17** | 6.25 | 14.58 |
| Conversational Fact Retrieval | **50.00** | 31.82 | 18.18 |
| Counting | **41.38** | 31.03 | 27.59 |
| Dissonant Emotion Interpretation | **60.00** | 17.14 | 22.86 |
| Eco-Acoustic Knowledge | **72.34** | 4.26 | 23.40 |
| Emotion Flip Detection | **85.00** | 0.00 | 15.00 |
| Emotion State Summarisation | **59.09** | 15.91 | 25.00 |
| Emotional Tone Interpretation | **75.76** | 12.12 | 12.12 |
| Event-Based Knowledge Retrieval | **48.48** | 18.18 | 33.33 |
| Event-Based Sound Reasoning | **72.92** | 6.25 | 20.83 |
| Harmony and Chord Progressions | **72.73** | 12.12 | 15.15 |
| Instrumentation | **74.29** | 14.29 | 11.43 |
| Key Highlight Extraction | **61.90** | 19.05 | 19.05 |
| Lyrical Reasoning | **70.00** | 0.00 | 30.00 |
| Melodic Structure Interpretation | **57.58** | 18.18 | 24.24 |
| Multi-Speaker Role Mapping | **44.44** | 3.70 | 51.85 |
| Musical Genre Reasoning | **67.65** | 14.71 | 17.65 |
| Musical Texture Interpretation | **70.59** | 8.82 | 20.59 |
| Phonemic Stress Pattern Analysis | **73.58** | 7.55 | 18.87 |
| Phonological Sequence Decoding | **59.18** | 18.37 | 22.45 |
| Rhythm and Tempo Understanding | **56.52** | 10.87 | 32.61 |
| Socio-Cultural Interpretation | **75.00** | 10.00 | 15.00 |
| Sound-Based Event Recognition | **52.17** | 10.87 | 36.96 |
| Temporal Event Reasoning | **75.00** | 12.50 | 12.50 |
| Temporal Reasoning | **46.43** | 25.00 | 28.57 |

### D.3 Extended Evaluation on MMAU Full Test Set

To provide a more comprehensive evaluation of our method's performance and generalization capabilities, we extend our experiments to the complete MMAU Test Set, which comprises approximately 9,000 audio question-answering samples spanning the full spectrum of audio understanding tasks across speech, sound, and music modalities. This large-scale evaluation serves as a critical validation of our framework's robustness and scalability beyond the Test-mini subset used in our main experiments.

**Consistent State-of-the-Art Performance at Scale.** The results presented in Tab. 17 demonstrate that CESAR maintains its commanding performance advantage when evaluated on the substantially larger test set. Our method achieves the highest overall accuracy, establishing new state-of-the-art results on this comprehensive benchmark. Importantly, this performance advantage holds across all three audio modalities, with particularly strong results in sound understanding tasks. The consistency between our Test-mini and Full Test Set results validates that our process-oriented training approach cultivates genuinely robust reasoning capabilities rather than overfitting to specific evaluation scenarios.

**Robustness of Process-Oriented Rewards.** The extended evaluation further confirms the critical importance of our multi-faceted reward suite. Both CESAR variants (with and without the Overthinking Penalty) significantly outperform the Ke-Omni-R baseline, which employs outcome-only rewards. This performance gap—maintained across thousands of diverse test cases—provides decisive evidence that rewarding the reasoning process itself yields superior and more generalizable audio understanding capabilities. The sustained advantage over strong proprietary models and specialized audio systems further validates our framework's effectiveness.

**Synergistic Effects Across Reasoning Modes.** Consistent with our findings on the Test-mini subset, the Full Test Set results reveal that process-oriented training creates beneficial synergies across both reasoning and non-reasoning inference modes. Our models demonstrate substantial improvements over the base Qwen2.5-Omni-7B model in both settings, confirming that cultivating high-quality reasoning processes fundamentally enhances the model's underlying audio understanding capabilities. This synergistic effect—where training to reason better simultaneously improves direct answering performance—represents a key advantage of our approach over traditional supervised fine-tuning methods.

**Implications for Real-World Deployment.** The strong performance on this large-scale benchmark has important practical implications. With nearly 9,000 diverse test cases covering a wide range of audio understanding scenarios, tasks, and difficulty levels, these results provide confidence in the real-world applicability of our framework. The consistency of our method's advantages across both small-scale (Test-mini) and large-scale (Full Test Set) evaluations suggests that CESAR-trained models can be reliably deployed in production environments where they will encounter diverse and unpredictable audio reasoning challenges.

Table 17: MMAU Full Test Set Results (9k samples). We evaluate our method against a comprehensive set of audio models on the complete MMAU test set. Best scores are highlighted in blue, second-best scores in green. All results show accuracy (%). Models are sorted by overall performance. We report the performance of Qwen2.5-Omni-7B (Xu et al., 2025) and Ke-Omni-R (Zhao et al., 2025) from our own reproductions under the same protocol; all other baseline results are taken from the MMAU paper (Sakshi et al., 2025).

| Method | Reasoning | Sound | Speech | Music | Overall Accuracy |
|--------|-----------|-------|--------|-------|------------------|
| **Our Proposed Methods** | | | | | |
| **CESAR** | ✓ | 77.60 | 76.09 | 67.77 | 73.79 |
| **CESAR** | ✗ | 76.57 | 75.50 | 65.70 | 72.55 |
| **CESAR w/o OP** | ✓ | 77.70 | 75.81 | 67.10 | 73.51 |
| **CESAR w/o OP** | ✗ | 76.13 | 75.60 | 66.77 | 72.80 |
| **RL Baseline Methods** | | | | | |
| Ke-Omni-R | ✓ | 75.37 | 73.77 | 66.73 | 71.94 |
| Ke-Omni-R | ✗ | 74.00 | 73.70 | 66.83 | 71.49 |
| **Base Model** | | | | | |
| Qwen2.5-Omni-7B | ✓ | 68.87 | 68.11 | 55.77 | 64.20 |
| Qwen2.5-Omni-7B | ✗ | 67.97 | 70.18 | 63.53 | 67.19 |
| **Proprietary and Open-Source Models** | | | | | |
| MiMo-Audio | - | 77.20 | 70.77 | 69.73 | 72.59 |
| Audio Flamingo 3 | - | 75.83 | 66.97 | 74.47 | 72.42 |
| Step-Audio-2-mini | - | 75.57 | 66.49 | 66.85 | 70.23 |
| Gemini 2.5 Pro | - | 70.63 | 72.67 | 64.77 | 69.36 |
| Gemini 2.5 Flash | - | 69.50 | 68.27 | 69.40 | 67.39 |
| Gemini 2.0 Flash | - | 68.93 | 72.87 | 59.30 | 67.03 |
| DeSTA2.5-Audio | - | 66.83 | 71.94 | 57.10 | 65.21 |
| Kimi-Audio | - | 70.70 | 56.57 | 65.93 | 64.40 |
| Audio Reasoner | - | 67.27 | 62.53 | 61.53 | 63.78 |
| Phi-4-multimodal | - | 62.67 | 63.80 | 61.97 | 62.81 |
| Gemini 2.5 Flash Lite | - | 62.50 | 67.47 | 54.87 | 61.61 |
| Audio Flamingo 2 | - | 68.13 | 44.87 | 70.20 | 61.06 |
| GPT-4o Audio | - | 63.20 | 69.33 | 49.93 | 60.82 |
| Qwen2-Audio-Instruct | - | 61.17 | 55.37 | 55.67 | 57.40 |
| Gemma 3n-4B | - | 50.27 | 62.13 | 53.20 | 55.20 |
| GPT-4o mini Audio | - | 49.67 | 67.47 | 35.97 | 51.03 |
| M2UGen | - | 44.97 | 35.77 | 38.53 | 39.76 |
| MusiLingo | - | 41.93 | 31.70 | 41.23 | 38.29 |
| SALMONN | - | 42.10 | 28.77 | 37.83 | 36.23 |
| MuLLaMa | - | 30.97 | 17.10 | 29.67 | 25.91 |
| GAMA-IT | - | 32.73 | 11.57 | 22.37 | 22.22 |
| GAMA | - | 30.73 | 16.97 | 17.33 | 21.68 |
| LTU | - | 20.67 | 15.33 | 15.68 | 17.23 |
| Audio Flamingo Chat | - | 23.33 | 7.67 | 15.77 | 15.59 |

## D.4 BENCHMARK RESULTS ON MMAU TEST-MINI

Table 18: MMAU Test-mini Benchmark Results. We evaluate our method against state-of-the-art proprietary and open-source audio models. Best scores are highlighted in blue , second-best scores in green . Accuracy (%) is reported. We report the performance of Qwen2.5-Omni-7B (Xu et al., 2025) and Ke-Omni-R (Zhao et al., 2025) from our own reproductions under the same protocol; all other baseline results are taken from the MMAU paper (Sakshi et al., 2025).

| Method | Reasoning | Sound | Music | Speech | Total Accuracy |
|---|---|---|---|---|---|
| **Our Proposed Methods** | | | | | |
| **CESAR** | ✓ | **83.48** | **73.05** | 74.77 | **77.10** |
| **CESAR** | ✗ | 79.88 | 67.96 | 73.27 | 73.70 |
| **CESAR w/o OP** | ✓ | **81.98** | 70.06 | **77.48** | **76.50** |
| **CESAR w/o OP** | ✗ | 80.48 | 70.06 | 74.47 | 75.00 |
| **RL Baseline Methods** | | | | | |
| Ke-Omni-R | ✓ | 79.28 | 70.06 | 74.47 | 74.60 |
| Ke-Omni-R | ✗ | 78.38 | **70.96** | 74.17 | 74.50 |
| **Base Models** | | | | | |
| Qwen2.5-Omni-7B | ✓ | 69.07 | 59.58 | 66.97 | 65.20 |
| Qwen2.5-Omni-7B | ✗ | 72.37 | 64.37 | 69.07 | 68.60 |
| **Proprietary Models** | | | | | |
| Gemini 2.5 Pro | - | 75.08 | 68.26 | 71.47 | 71.60 |
| Gemini 2.5 Flash | - | 73.27 | 65.57 | **76.58** | 71.80 |
| Gemini 2.0 Flash | - | 71.17 | 65.27 | 75.08 | 70.50 |
| GPT-4o Audio | - | 64.56 | 56.29 | 66.67 | 62.50 |
| GPT-4o mini Audio | - | 50.75 | 39.22 | 69.07 | 53.00 |
| **Open-Source Audio Models** | | | | | |
| Kimi-Audio | - | 75.68 | 66.77 | 62.16 | 68.20 |
| Audio Reasoner | - | 67.87 | 69.16 | 66.07 | 67.70 |
| Phi-4-multimodal | - | 65.47 | 64.37 | 67.27 | 65.70 |
| Audio Flamingo 2 | - | 71.47 | **70.96** | 44.74 | 62.40 |
| Qwen2-Audio-Instruct | - | 67.27 | 56.29 | 55.26 | 59.60 |

Our evaluation on the MMAU Test-mini benchmark, with comprehensive results presented in Tab. 18, not only establishes a new state-of-the-art performance but, more importantly, a deeper analysis of these results uncovers several critical insights into the nature of reasoning in Audio LLMs and the means by which it can be effectively cultivated.

**An Insight on Scaling Reasoning Process vs. Scaling Parameters.** The first critical insight from these results emerges from the clear superiority of scaling up the *reasoning process* over simply scaling up model parameters. CESAR, at just 7B parameters, achieves its state-of-the-art 77.10% accuracy not by possessing a larger architecture, but by effectively scaling its cognitive process at inference time—a latent capability unlocked by our training and fully realized through test-time analysis of reasoning length. This performance decisively surpasses that of proprietary models like the Gemini 2.5 series and GPT-4o Audio, whose primary scaling axis is their vast parameter count. This finding strongly suggests that a new paradigm for performance enhancement is not only viable but superior: instead of relying on brute-force parameter scaling, strategically cultivating and dynamically scaling a model's reasoning process offers a more efficient and effective path to advanced capabilities.

**The Symbiotic Rise of Reasoning and Intuition.** Beyond sheer performance, the results reveal a more subtle and perhaps more profound insight into the effects of our training paradigm. Our process-oriented RL training not only enhances the model's explicit, step-by-step reasoning but

also substantially elevates its direct, intuitive answering capability. This is evident as our model without reasoning ('CESAR w/o Reasoning') achieves 73.70% accuracy, a score far superior to the base model's 68.60% in the same setting. This suggests that incentivizing high-quality reasoning pathways does more than just teach a model to generate a thinking monologue; it fundamentally refines the model's core representations of the acoustic world. This discovery carries significant implications for practical deployment, as it enables highly efficient inference through fast, direct answers while retaining a powerful, on-demand reasoning faculty for more complex challenges.

**Transforming Reasoning from Detriments into Gains.** Finally, the data provides a clear narrative on the evolution of reasoning itself. The base Qwen2.5-Omni-7B model exemplifies the critical problem of uncontrolled reasoning, where performance catastrophically drops by 3.4 points when it is prompted to "think" (from 68.60% to 65.20%). This is a textbook case of the test-time inverse scaling problem. In stark contrast, CESAR systematically reverses this trend, gaining a robust 3.4 points under the same conditions (from 73.70% to 77.10%). This transforms the act of reasoning from a high-risk gamble into a reliable and scalable tool for performance enhancement. With this newfound stability, reasoning is no longer an unpredictable behavior but a controllable capability. We therefore suggest that practitioners can now confidently employ reasoning and, by combining it with test-time scaling analysis, identify the model-specific "sweet spot" to unlock its full, calibrated potential. This marks a pivotal shift, firmly establishing reasoning as a core asset for advancing multimodal understanding.

### D.5 BEYOND AGGREGATE SCORES: A TASK-LEVEL ANALYSIS OF CONTROLLABLE REASONING CAPABILITY

While aggregate scores (Table 18) establish the state-of-the-art performance of our method, they fundamentally mask the most profound discovery of our work: the systematic emergence of controllable reasoning archetypes. A granular, multi-faceted analysis is essential to understand not merely the quantitative superiority, but the qualitative transformation of reasoning from a randomly emergent phenomenon into a precisely engineerable capability.

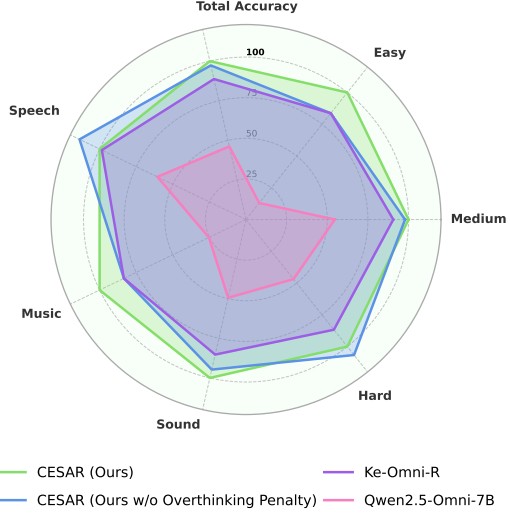

Figure 4: Normalized multi-dimensional performance comparison on the MMAU Test-mini benchmark. Performance is scaled relative to CESAR (Ours), which constitutes the 100% baseline on each axis. This visualization reveals the emergence of distinct reasoning specializations across task difficulties.

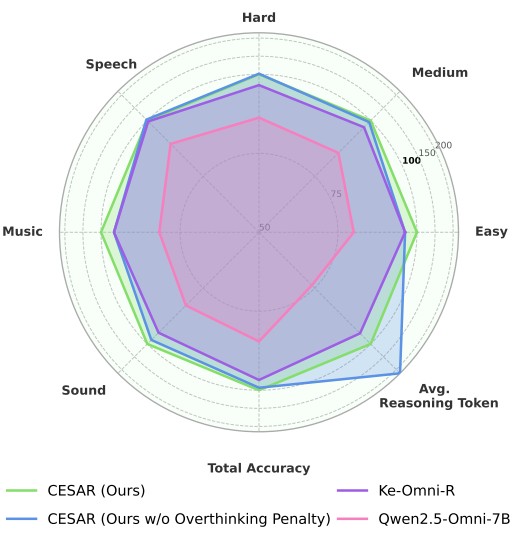

Figure 5: Extended radar analysis including reasoning token efficiency. This enhanced visualization provides clearer visibility of the fundamental trade-offs between reasoning depth and computational cost, unveiling two fundamentally different reasoning archetypes optimized for distinct cognitive scenarios.

To reveal this transformation, we present a comprehensive radar chart analysis in Fig. 4, where the performance of our full model, CESAR (Ours), serves as the 100% baseline across seven key eval-

uation dimensions. This normalization strategy exposes not merely superior performance, but the emergence of fundamentally different cognitive architectures that our process-oriented framework has systematically cultivated.

**The Emergence of Two Reasoning Archetypes: A Tale of Cognitive Specialization.** The radar charts provide irrefutable visual evidence for our central hypothesis while simultaneously unveiling an unexpected discovery: process-oriented supervision does not simply improve reasoning—it enables the systematic engineering of distinct reasoning archetypes. The performance polygons of our two CESAR variants (green and blue) dominate both visualizations (Fig. 4 and Fig. 5), covering significantly larger areas than the strong RL baseline Ke-Omni-R (purple) and the base model Qwen2.5-Omni-7B (pink). This validates our core claim that genuine reasoning emerges only when supervision targets the reasoning *process*, not just final outcomes.

However, the most profound insight emerges from the striking divergence between our two CESAR variants—a divergence that reveals the existence of a fundamental trade-off in reasoning system design. The CESAR (w/o Overthinking Penalty) variant exhibits a distinctive cognitive profile: exceptional performance on *Hard* tasks, consistently exceeding even our 100% baseline, but at a deliberate cost of efficiency on simpler problems. This model represents what we term a **depth specialist**—an archetype that favors exhaustive, thorough analysis over computational efficiency.

In stark contrast, the full CESAR (Ours) model demonstrates a fundamentally different cognitive architecture. It forms a perfectly calibrated profile across all difficulty levels, showing exceptional stability on *Easy* and *Medium* tasks while maintaining competitive performance on *Hard* problems. This represents a **calibrated generalist**—an archetype optimized for consistent, efficient reasoning across diverse problem complexities.

**The Fundamental Performance-Depth Trade-off: Efficiency vs. Thoroughness.** The extended analysis including reasoning token efficiency (Fig. 5) exposes the computational mechanics underlying this cognitive divergence. The depth specialist achieves its superior performance on challenging tasks by investing substantially more reasoning tokens—engaging in extensive, multi-step analytical processes that thoroughly explore problem spaces. Conversely, the calibrated generalist demonstrates remarkable efficiency, achieving comparable overall performance while operating under strict computational constraints imposed by the overthinking penalty.

This discovery challenges conventional assumptions about reasoning optimization and reveals a fundamental principle: **there exists an inherent tension between reasoning depth and computational efficiency, and optimal performance emerges when models are explicitly trained to navigate this trade-off according to task requirements**. The depth specialist excels precisely because it is willing to invest computational resources in exhaustive analysis when problems demand it. The generalist succeeds by learning to apply just enough analytical effort to solve problems effectively without wasteful over-elaboration.

**Engineering Controllable Cognitive Architectures.** The emergence of these distinct reasoning archetypes represents far more than an interesting experimental observation—it provides definitive proof that reasoning has been transformed from an unpredictable emergent property into a controllable, engineerable capability. The stark differences between our variants are not accidental byproducts of training, but the direct result of our process-oriented reward architecture functioning as precision engineering tools for cognitive behavior.

By systematically modulating a single reward component—the overthinking penalty—we have demonstrated the ability to produce models with predictably different reasoning profiles, each optimized for distinct deployment scenarios. The depth specialist thrives in research environments where thorough analysis justifies computational cost, making it ideal for complex analytical tasks requiring maximum cognitive depth. The calibrated generalist excels in production systems where efficiency and consistency are paramount, delivering reliable performance across diverse problem types without excessive resource consumption.

This unprecedented level of control demonstrates that CESAR transcends being merely a high-performing model—it represents a comprehensive framework for engineering the next generation of controllable audio reasoners with specific, desirable cognitive traits. We have moved decisively beyond the traditional paradigm of hoping for beneficial reasoning patterns to emerge spontaneously,

entering an era where cognitive capabilities can be systematically specified, implemented, and validated with the same precision as traditional software systems.

The dual visualization provides definitive empirical proof that our methodology enables the systematic exploration of the reasoning capability space, allowing researchers and practitioners to engineer cognitive systems precisely tailored to their specific requirements and constraints. This work establishes both the theoretical foundation and practical methodology for controllable AI development, where reasoning behavior becomes a design parameter rather than an emergent accident.

## D.6 BENCHMARK RESULTS ON MMSU

Table 19: MMSU Benchmark Results. We evaluate our method against state-of-the-art audio models across perception and reasoning tasks in speech understanding. Best scores are highlighted in **blue** , second-best scores in **green** . All results show accuracy (%). Human performance is included as an upper bound reference. We report the performance of Ke-Omni-R (Zhao et al., 2025) from our own reproductions under the same protocol; all other baseline results are taken from the MMSU paper (Wang et al., 2025) (including Qwen2.5-Omni-7B (Xu et al., 2025).). The results of Audio Flamingo 3 are taken from their paper (Goel et al., 2025b).

| Models | Perception Tasks | | | | Reasoning Tasks | | | | Overall |
|---|---|---|---|---|---|---|---|---|---|
| | Semantics | Phonology | Paralinguistics | Avg | Semantics | Phonology | Paralinguistics | Avg | |
| *Our Proposed Method* | | | | | | | | | |
| **CESAR** | 60.16 | 50.16 | 39.50 | 48.45 | 88.72 | 80.66 | 57.01 | 81.07 | 64.24 |
| *Audio RL Baseline* | | | | | | | | | |
| Ke-Omni-R | 58.74 | 46.31 | 40.50 | 47.09 | 86.82 | 74.31 | 60.00 | 78.06 | 62.08 |
| *Proprietary Models* | | | | | | | | | |
| Gemini 1.5 Pro | 57.06 | 53.60 | 31.23 | 46.10 | 79.47 | 83.46 | 46.33 | 76.16 | 60.68 |
| Qwen2.5-Omni-7B | 55.12 | 37.33 | 39.35 | 42.50 | 88.00 | 81.37 | 48.36 | 79.83 | 60.57 |
| Kimi-Audio | 57.64 | 42.30 | 35.74 | 43.52 | 81.77 | 76.65 | 55.22 | 76.03 | 59.28 |
| GPT-4o Audio | 59.70 | 41.56 | 21.44 | 39.67 | 80.83 | 78.74 | 26.25 | 71.96 | 56.38 |
| Qwen2-Audio-Instruct | 52.14 | 32.87 | 35.56 | 39.02 | 77.62 | 64.81 | 46.67 | 68.90 | 53.27 |
| Gemini 2.0 Flash | 47.17 | 41.30 | 30.62 | 40.83 | 70.69 | 70.69 | 36.16 | 47.83 | 51.03 |
| *Open-Source Audio Models* | | | | | | | | | |
| Audio Flamingo 3 | – | – | – | – | – | – | – | – | 62.30 |
| MiniCPM | 56.56 | 34.05 | 36.48 | 40.54 | 80.71 | 74.72 | 46.71 | 73.57 | 56.53 |
| MERA LION | 54.49 | 33.69 | 25.84 | 35.74 | 80.32 | 77.18 | 41.49 | 73.68 | 54.10 |
| Qwen-Audio-Chat | 57.21 | 38.52 | 24.70 | 35.69 | 58.61 | 59.78 | 25.60 | 55.93 | 46.92 |
| DIVA | 44.36 | 33.72 | 27.45 | 33.95 | 62.32 | 74.24 | 40.00 | 65.04 | 48.31 |
| Megrez-3B-Omni | 41.36 | 32.52 | 26.35 | 32.48 | 73.53 | 66.11 | 40.42 | 67.05 | 49.03 |
| Step-Audio | 31.56 | 29.39 | 24.01 | 28.72 | 49.10 | 50.09 | 45.27 | 47.27 | 37.42 |
| BLSP | 31.35 | 20.96 | 23.75 | 28.36 | 47.91 | 42.31 | 42.08 | 44.97 | 35.96 |
| GLM-4-Voice | 27.80 | 24.52 | 27.34 | 26.18 | 46.10 | 48.16 | 44.35 | 46.76 | 35.51 |
| *Human Performance (Upper Bound)* | | | | | | | | | |
| Human | 87.10 | 94.32 | 92.88 | 91.24 | 82.16 | 87.60 | 89.12 | 86.77 | 89.72 |
| *Random Baselines* | | | | | | | | | |
| Most Frequent Choice | 26.20 | 26.04 | 27.83 | 29.83 | 28.30 | 28.30 | 30.10 | 28.41 | 28.06 |
| Random Guess | 24.30 | 25.70 | 26.10 | 24.90 | 23.80 | 25.40 | 25.40 | 25.02 | 25.37 |

The MMSU benchmark, with its unique split between perception and reasoning tasks, provides a granular lens for a multi-faceted analysis of a model's capabilities. Our examination of the results in Tab. 19 reveals several critical findings, starting with the validation of our method's reasoning capabilities, followed by an exploration of its surprising efficiency and broader impacts, and concluding with an identification of key frontiers for future research.

**The Initial Breakthrough: Reasoning Closes to the Human Level.** The analysis first confirms the efficacy of CESAR in its target domain. The data shows the model achieves superior performance in reasoning tasks, where its average score of 81.07% not only establishes a new state-of-the-art but also approaches the human benchmark of 86.77%. This proficiency is particularly pronounced in Semantic Reasoning, where CESAR achieves a super-human score of 88.72%. This is a direct and powerful validation that our process-oriented training is exceptionally effective at cultivating the kind of deep, nuanced understanding that was previously the domain of human cognition.

**A Small Model Can also Win.** A more compelling finding emerges, however, one that challenges the foundational assumptions of the field. The data in Tab. 19 reveals that CESAR, a 7B model, not only establishes superior performance in reasoning over significantly larger proprietary models like Gemini 1.5 Pro, but also—unexpectedly—surpasses them in average perception tasks (48.45% vs. 46.10%). This is a pivotal finding: it demonstrates that a smaller model, when endowed with superior reasoning, can outperform larger competitors across *both* cognitive and perceptual dimensions. This result provides strong evidence for a new, more efficient scaling paradigm.

**The Ripple Effect: How Better Thinking Creates Better Hearing.** The model's unexpected strength in perception suggests a deeper mechanism is at play. The data points to an unanticipated

synergy: enhancing reasoning seems to have a beneficial ripple effect on foundational perception. While our rewards explicitly target higher-order thinking, the process of learning to form consistent and logical connections appears to refine the model's underlying representations of the acoustic world. This finding suggests that our methodology improves not just how a model interprets sounds, but how effectively it processes them at a more fundamental level, explaining how a targeted cognitive enhancement can lead to broader, more holistic improvements in multimodal understanding.

**A New Vista: Beyond the Perceptual Bottleneck.**    Ultimately, the success in reasoning brings a critical challenge for the field into sharp relief. The asymmetry between our model's near-human reasoning and its still-developing perception (which, while outperforming its peers, still lags far behind the human score of 91.24%) illuminates a clear "perceptual bottleneck." This should not be viewed as a limitation of our method, but rather as a key finding that clarifies the path for future research. It reinforces our central thesis: the future lies not in the resource-intensive race of simply scaling up model size, but in the new science of scaling specific *capabilities*. Our work demonstrates that reasoning can be scaled to near-human levels within a compact model; the clear next step is to apply a similarly focused, principled approach to perception, paving the way for a future of AI that is not only more powerful but also dramatically more efficient and accessible.

**Content Grounding: Validating Synergistic Perception Improvements.**    To further validate the synergistic perception improvements observed in Table 19, we analyze CESAR's performance on MMSU's **Content Grounding** task, which evaluates accurate content transcription from speech (e.g., "Which sentence is the correct transcription of the audio?"). This task provides an effective diagnostic for assessing transcription-related capabilities within the benchmark framework. As shown in Table 20, CESAR achieves 90.80% accuracy, substantially outperforming Ke-Omni-R (72.50%) by 18.3 points and the base model (59.60%) by 31.2 points. This dramatic improvement demonstrates that our process-oriented training creates synergistic effects that extend beyond reasoning to measurably improve fundamental speech understanding capabilities, including semantic-level transcription accuracy and robustness to acoustic variations—providing concrete evidence that cultivating high-quality reasoning processes simultaneously refines the model's perceptual foundations.

Table 20: Content Grounding Performance on MMSU. Best score is highlighted in blue .

| Method | Content Grounding Accuracy (%) |
|---|---|
| **CESAR (Ours)** | **90.80** |
| Ke-Omni-R | 72.50 |
| Qwen2.5-Omni-7B | 59.60 |

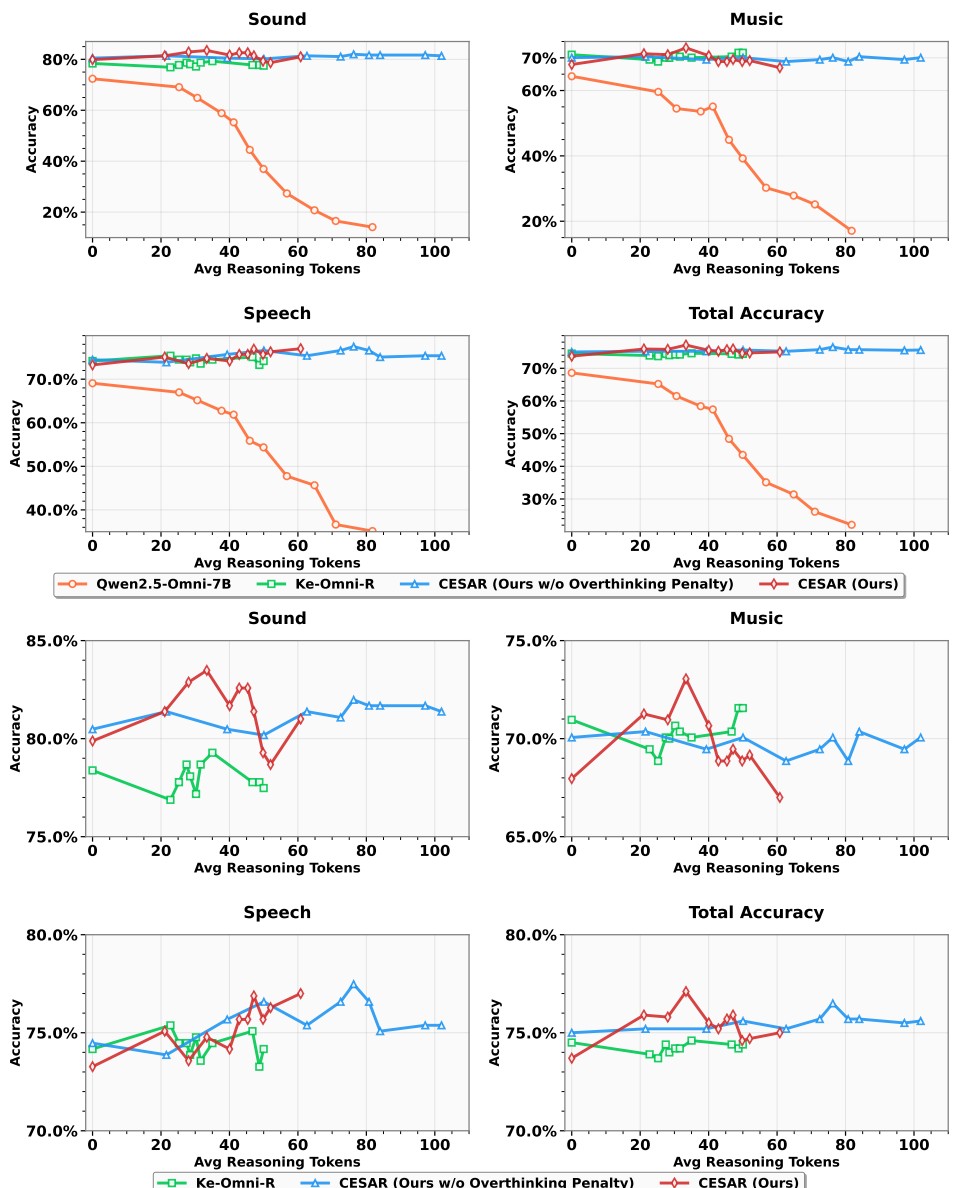

Figure 6: **Test-Time Scaling Curves of Reasoning.** Accuracy is plotted against the average length of the reasoning chain (in used tokens). **(Top Row)** The full comparison reveals a catastrophic performance collapse of the base Qwen2.5-Omni-7B model as it generates longer reasoning chains, empirically demonstrating the test-time inverse scaling problem. In contrast, all RL-trained models remain robust. **(Bottom Row)** A zoomed-in view of the RL models highlights the performance peak of our full method (i.e., CESAR (Ours)), which discovers a "reasoning sweet spot". It consistently outperforms both the version without the Overthinking Penalty reward (i.e., CESAR (Ours w/o Overthinking Penalty)) and the Ke-Omni-R baseline.

Aggregate accuracy scores, while informative, obscure a critical underlying dynamic: the **test-time inverse scaling problem** that plagues audio language models lacking explicit reasoning training. This phenomenon is twofold: first, prompting such models to generate a chain of thought often yields worse results than direct, zero-shot answering. Second, their performance degrades precipitously as

the reasoning chain lengthens. To rigorously investigate these dynamics and validate our solution, we introduce a **test-time scaling analysis**.

This training-free inference methodology allows us to probe a model's reasoning capability as a function of its computational budget. We achieve this by systematically varying an upper bound on reasoning length, specifically by adjusting the `max_think_len` parameter within the prompt across a range (e.g., 25, 50, ..., 250) (Zhao et al., 2025). Critically, this parameter does not force a fixed output length; rather, it provides a ceiling, allowing the model to autonomously determine an appropriate reasoning depth based on the problem's demands and its own intrinsic capabilities. We then plot accuracy against the *actual average number of reasoning tokens* generated. As illustrated in Fig. 6, this analysis provides a granular view into each model's reasoning behavior, reveals profound differences in their underlying skills, and demonstrates how to fully unlock the latent reasoning potential cultivated by our framework.

**The Peril of Under-Optimized Reasoning: A Case of Test-Time Inverse Scaling.** The most dramatic finding, shown in the top row of Fig. 6, is the **catastrophic performance collapse** of the base Qwen2.5-Omni-7B model. While it begins with a respectable accuracy at zero reasoning tokens (i.e., direct answering), its performance enters a free fall as it is prompted to generate longer reasoning processes. This provides powerful empirical evidence for the test-time inverse scaling problem: for a model that has not been explicitly trained *how* to reason, "thinking more" is actively harmful. Each additional token of unguided "reasoning" introduces new opportunities for error accumulation and hallucination, transforming a decent zero-shot guesser into a demonstrably poor reasoner.

**Curing Test-Time Inverse Scaling through Process-Oriented RL.** In stark contrast, all models trained with our process-oriented reinforcement learning framework are completely immune to this collapse. The bottom row of Fig. 6 shows that our models (CESAR and its variant) and the Ke-Omni-R baseline all maintain remarkable stability as reasoning length increases. This result establishes a fundamental principle: **robust training cultivates robust scaling**. By receiving granular feedback on the reasoning *process*, our models learn to generate coherent, logically sound, and self-consistent reasoning processes that do not derail. This learned robustness is the key to curing the test-time inverse scaling fragility observed in the base model, transforming reasoning from detriments into gains.

**Unlocking Calibrated Reasoning and Model-Specific "Sweet Spots".** The zoomed-in analysis reveals the final and most profound layer of insight. While the Ke-Omni-R baseline is stable, its performance is noisy and fails to consistently benefit from longer reasoning. This comparison highlights the limitations of outcome-only rewards. Our process-oriented rewards, however, cultivate distinct and controllable reasoning styles. The CESAR (w/o Overthinking Penalty) variant, for instance, exhibits a preference for longer reasoning chains, showing sustained high performance as token count increases.

Most significantly, our full method, CESAR, demonstrates a form of learned metacognition. It discovers a model-specific **"reasoning sweet spot,"** where its performance actively *peaks* at an optimal reasoning depth (around 30-40 tokens) before gracefully stabilizing. This behavior is a direct consequence of the 'Overthinking Penalty' reward, which trains the model to balance analytical sufficiency with conciseness. It learns not only *how* to reason, but also *how much* to reason. This finding challenges the monolithic "longer is better" assumption, proving that reasoning is a trainable skill. The test-time scaling analysis is therefore not merely an evaluation tool; it is the key to unlocking these cultivated, model-specific reasoning capabilities at inference time, allowing each model to achieve its peak performance.

## D.8  Quantifying Reasoning Quality beyond Accuracy: An AI-as-Judge Framework

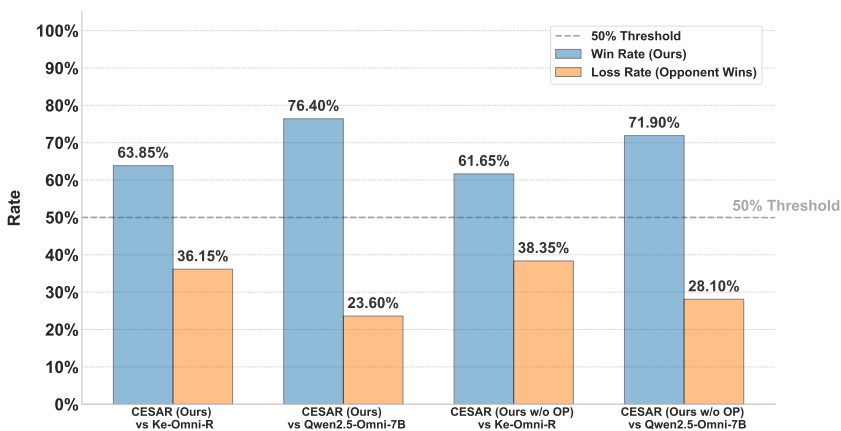

Figure 7: **Quantitative Analysis of Reasoning Quality via AI-as-Judge.** Win rate of our model's reasoning process when compared head-to-head against the base model (Qwen2.5-Omni-7B) and the Ke-Omni-R baseline. The dominant win rates provide strong quantitative evidence of superior reasoning quality, a metric that accuracy scores alone cannot capture. Throughout, OP denotes the overthinking penalty defined in equation 8.

Evaluating the quality of a model's reasoning process—distinct from the correctness of its final answer—poses a significant challenge in multimodal AI. Current evaluation paradigms are insufficient: outcome-based accuracy is a coarse proxy that can reward correct answers derived from flawed or fallacious logic, while anecdotal qualitative analysis is inherently unscalable and subjective. This methodological gap makes it difficult to determine whether a model is developing genuine analytical skill or simply becoming a more effective test-taker.

To address this, we introduce a scalable and rigorous framework for the quantitative evaluation of reasoning quality itself. We employ a powerful, state-of-the-art multimodal model, akin to GPT-4o Audio (Hurst et al., 2024), as an expert adjudicator. For each comparison, after providing the judge with the full context (audio, question, choices, and correct answer) and the two reasoning traces, we use the following direct prompt for evaluation: *Given the audio context and two reasoning processes from Model A and Model B, try to determine which process is superior. A superior process is more logical, faithful to the audio, and follows a clearer analytical path. Focus on the quality of the reasoning, not just the final answer's correctness, and conclude with 'Model A Wins', 'Model B Wins', or 'Tie'.* This approach ensures a consistent and targeted assessment focused squarely on the analytical process.

Noting that to enhance the clarity and readability of the final results of AI-as-Judge, we distribute 'Tie' outcomes equally for the final win-rate calculation (e.g., an initial outcome where Model A wins 40%, Model B wins 20%, and 40% are ties is converted to a final win rate of 60% for A and 40% for B).

The results of AI-as-Judge, presented in Fig. 7, are decisive. Against the base Qwen2.5-Omni-7B model, CESAR's reasoning is judged superior in a commanding **76.4%** of comparisons. This offers the first quantitative proof that our process-centric rewards cultivate a fundamentally more robust and logical reasoning architecture, rather than merely improving final-answer accuracy. Even more critically, when pitted against Ke-Omni-R—a strong baseline also trained with reinforcement learning but with outcome-only rewards—our model's reasoning prevails in a significant **63.85%** of cases. This result starkly illustrates the limitations of simplistic, outcome-based RL and validates the necessity of our multi-faceted reward suite for shaping genuinely high-quality reasoning. By filling a critical methodological gap, our work provides the field with a scalable tool to assess reasoning quality directly, proving that our process-centric framework cultivates verifiably superior analytical capabilities.

### D.9 QUALITATIVE ANALYSIS: THE ANATOMY OF CULTIVATED REASONING

Beyond quantitative benchmarks, a granular examination of the reasoning traces reveals the concrete mechanisms through which CESAR transforms a model's cognitive behavior. The following head-to-head comparisons are not merely anecdotal evidence of better performance; they are a microscopic validation of our central thesis. They illustrate precisely how our process-oriented rewards eradicate the critical failure modes—hallucination, logical fallacies, and inconsistency—that plague models trained without such guidance, thereby cultivating reasoning that is demonstrably more **E**ffective, **C**onsistent, and **L**ogical.

#### D.9.1 COMPARISON WITH BASE MODEL (QWEN2.5-OMNI-7B): UNDER-OPTIMIZED VS. CULTIVATED REASONING

This comparison starkly contrasts the brittle, uncontrolled output of a powerful base model against the robust, structured thought process instilled by CESAR. The examples demonstrate that merely prompting a model to "think" is insufficient; without explicit training on *how* to reason, the process itself becomes a source of error, leading to failures that range from high-level factual invention to low-level execution mistakes.

---

**Example 1: Robustness against Hallucination**

**Question:** Based on the given audio, identify the source of the speech.

**Choices:** man, woman, child, robot

**Correct Answer:** man

**Task ID:** 72fb5481-73ae-409d-8e16-c94ac48d2ee4

| CESAR (Correct) | Qwen2.5-Omni-7B (Wrong) |
|---|---|
| **Reasoning:** *Given the options, the speech is likely from a man. The **tone** and **volume suggest** an adult male. The other options are less probable.* | **Reasoning:** *The audio is a speech, and the voice is male. The options provided do not include 'man', so the closest match is 'robot'.* |
| **Answer:** man | **Answer:** robot |

---

**Grounded Reasoning vs. Factual Hallucination.** The base model often fails to ground its reasoning in the provided context. In Example 1, after correctly identifying the acoustic evidence ("the voice is male"), it invents a false premise—that 'man' is absent from the choices—and proceeds to an illogical conclusion. This reveals a fundamental inability to tether its internal state to the problem's explicit constraints. CESAR, by contrast, exhibits a learned, systematic methodology. Its trace begins with a grounding step (**Given the options**), explicitly acknowledging the context, and then connects acoustic cues to an inference using relational language (**suggests**)—showcasing an **effective** and sound analytical process directly instilled by our 'Keywords' reward.

**Logical Deduction vs. Flawed Inference.** The base model struggles to interpret complex acoustic scenes. In Example 2, it oversimplifies a rich soundscape into a generic, low-information label ("chaotic noise"), from which it makes a spurious logical leap. This highlights a core weakness: the inability to decompose and analyze multi-element audio. CESAR demonstrates a far more sophisticated, multi-step **logical** process. It correctly identifies the constituent acoustic parts (**music, and shouting**), synthesizes them into a high-level hypothesis ("a large-scale event"), and systematically evaluates the choices to find the one that **fits the description best**. This is a true deductive capability, a direct outcome of rewarding structured analytical patterns.

**Precise Execution vs. Minor Inconsistency.** Example 3 reveals a subtle but critical failure in precision. While the base model's reasoning is perceptually and logically correct, it stumbles at the final execution step, introducing a fatal typo (_click_click) into its answer. This shows that a sound thought process is nullified if the model cannot reliably translate it into a correct final output. CESAR's end-to-end process rewards, which value not just the reasoning trace but its faithful conversion into a final answer, ensure this level of precision. Its reasoning trace also exhibits the

**Example 2: Logical Deduction from Acoustic Cues**

| | |
|---|---|
| **Question:** What kind of event could be taking place in the audio? | |
| **Choices:** A quiet library during study hours, A heated argument escalating to violence, A military parade with a marching band, A gentle breeze blowing through trees | |
| **Correct Answer:** A military parade with a marching band | |
| **Task ID:** 3690d238-8902-4217-a068-80a56972842e | |
| **CESAR (Correct)** | **Qwen2.5-Omni-7B (Wrong)** |
| **Reasoning:** *The presence of marching, **music, and shouting suggests** a large-scale event. The options are a quiet library, a heated argument, a military parade, and a gentle breeze. A military parade **fits the description best**, as it involves a marching band and organized **shouting**.* | **Reasoning:** *The audio contains the sound of a man speaking, followed by a loud, chaotic noise. The options are a quiet library, a heated argument, a military parade, and a gentle breeze. The chaotic noise could be interpreted as a violent escalation, so a heated argument is the most likely scenario.* |
| **Answer:** A military parade with a marching band | **Answer:** A heated argument escalating to violence |

**Example 3: Reasoning-Answer Consistency**

| | |
|---|---|
| **Question:** What was the order of the sounds in the sequence? | |
| **Choices:** ['light_switch_clicking', 'boiling_water', 'doorbell_ringing', 'clock_ticking'], ['boiling_water', 'light_switch_clicking', 'clock_ticking', 'doorbell_ringing'], ['clock_ticking', 'doorbell_ringing', 'boiling_water', 'light_switch_clicking'], ['doorbell_ringing', 'clock_ticking', 'light_switch_clicking', 'boiling_water'] | |
| **Correct Answer:** ['light_switch_clicking', 'boiling_water', 'doorbell_ringing', 'clock_ticking'] | |
| **Task ID:** bbabe360-0573-43d4-b2e6-6892150cbdcd | |
| **CESAR (Correct)** | **Qwen2.5-Omni-7B (Wrong)** |
| **Reasoning:** ***Considering the sequence**, the light switch clicking is **followed by** boiling water, **then** a doorbell rings, and **finally**, a clock ticks. This order fits the given options.* | **Reasoning:** *First, there's a light switch click, then boiling water sound, followed by a doorbell ringing, and finally, a clock ticking.* |
| **Answer:** ['light_switch_clicking', 'boiling_water', 'doorbell_ringing', 'clock_ticking'] | **Answer:** ['light_switch_click_click', 'boiling_water', 'doorbell_ringing', 'clock_ticking'] |

sequential pattern keywords (**Considering the sequence... followed by... then... finally**) that our framework explicitly incentivizes.

### D.9.2 COMPARISON WITH RL BASELINE (KE-OMNI-R): THE NECESSITY OF PROCESS REWARDS

This comparison isolates the crucial contribution of our multi-faceted, process-oriented reward suite against a baseline also trained with RL but guided only by the correctness of the final answer. These examples provide causal evidence that optimizing for outcomes alone is insufficient, as it permits and may even reinforce broken cognitive processes.

**Enforcing Reasoning-Answer Consistency.** Example 4 is a textbook failure of reasoning-answer consistency. Ke-Omni-R correctly identifies the acoustic fact in its reasoning ("rings three times") but then inexplicably provides a contradictory answer (2). An outcome-only reward system is blind to this internal contradiction; on trials where the model might guess the correct final answer despite such a flaw, it would be rewarded, reinforcing this broken cognitive process. CESAR's 'Reasoning Consistency Reward' directly targets this failure mode, ensuring the final answer is a direct and faithful consequence of the thought process. This makes the reasoning verifiably **consistent** and trustworthy.

**Example 4: Superior Reasoning-Answer Consistency**

**Question:** How many times does the telephone ring in the audio?

**Choices:** 2, 4, 5, 3

**Correct Answer:** 3

**Task ID:** b132f501-53cd-4e78-84e3-ac65c5588260

| CESAR (Correct) | Ke-Omni-R (Wrong) |
|---|---|
| **Reasoning:** *Considering the options, the telephone rings three times. The audio clearly shows three distinct rings.* | **Reasoning:** *The telephone rings three times, as indicated by the three distinct ring sounds in the audio.* |
| **Answer:** 3 | **Answer:** 2 |

**Example 5: Concise vs. Convoluted Reasoning**

**Question:** What is the characteristic of the chord played from 0:02 to 0:03 in the audio?

**Choices:** A:min/P5, E:maj/P1, D:maj/M3, G:maj/P1

**Correct Answer:** A:min/P5

**Task ID:** 7333dffb-368f-44cc-adb4-b4e9805164a3

| CESAR (Correct) | Ke-Omni-R (Wrong) |
|---|---|
| **Reasoning:** *Considering the options, 'A:min/P5' seems the most fitting as it's a minor chord and 'P5' indicates a perfect fifth. The others don't fit the context as well.* | **Reasoning:** *The chord progression from 0:02 to 0:03 involves a progression from a lower note to a higher one... The progression is from a lower note to a lower one... The progression is from a lower note to a lower one... The progression is from a lower note to a higher one...* |
| **Answer:** A:min/P5 | **Answer:** E:maj/P1 |

**Efficient Reasoning vs. Convoluted Overthinking.** Example 5 highlights the problem of unconstrained, inefficient reasoning. The Ke-Omni-R baseline engages in a convoluted and self-contradictory monologue (a loop of opposing statements). This "overthinking" is a direct symptom of a reward function that does not value efficiency. CESAR, shaped by our Overthinking Penalty and Keywords rewards, produces an **effective** and parsimonious trace. It efficiently uses domain knowledge (**minor chord**, **perfect fifth**) and structured evaluation (**most fitting**) to arrive at the answer, demonstrating a learned ability to allocate cognitive effort purposefully.

**Example 6: Robustness against Factual Hallucination**

**Question:** Given that an infant is crying, name a famous person known for their association with the animal heard in the background.

**Choices:** George Orwell, Ernest Hemingway, Charles Darwin, Charles Dickens

**Correct Answer:** Ernest Hemingway

**Task ID:** 255742c0-cc69-4be5-891e-a3c796904934

| CESAR (Correct) | Ke-Omni-R (Wrong) |
|---|---|
| **Reasoning:** *Among the options, Ernest Hemingway is known for his love of animals, particularly cats. The animal sound could be a cat. George Orwell and Charles Dickens are not known for their association with cats. Charles Darwin is known for his work on evolution, not for cats.* | **Reasoning:** *Considering the context of the question, the animal in the background is a pig. Ernest Hemmanway is famously associated with pigs in his works, particularly in "The Old Man and the Sea."* |
| **Answer:** Ernest Hemingway | **Answer:** Ernest Hemmanway |

**Grounded Knowledge vs. Factual Hallucination.** Finally, Example 6 reveals a spectacular, multi-layered failure of factual grounding from the RL baseline. Ke-Omni-R not only misidenti-

fies the animal sound as a "pig," but proceeds to invent a completely false piece of literary trivia to justify its choice, incorrectly linking Hemingway to pigs in "The Old Man and the Sea." This shows how an outcome-only agent can generate syntactically plausible reasoning that is semantically untethered from reality. The failure is compounded by an execution error, as it ultimately misspells the author's name (Hemmanway) in the final answer. CESAR's reasoning, by contrast, is sound. It correctly identifies a plausible animal (cat), recalls Hemingway's true association with it, and systematically eliminates other options. Our holistic reward suite prevents this kind of unconstrained hallucination, ensuring the reasoning process is both internally and externally valid.

### D.9.3 EFFICIENT REASONING VS. PATHOLOGICAL OVERTHINKING: THE CRITICAL ROLE OF THE OVERTHINKING PENALTY

This final comparison isolates the specific contribution of our 'Overthinking Penalty' by presenting a case where its absence leads to outright failure. Here, the full CESAR model arrives at the correct answer through a clear and evidence-based analytical path. The 'CESAR w/o OP' variant, however, initially identifies the correct acoustic evidence but then engages in a long, convoluted reasoning process. This overthinking causes it to fixate on an irrelevant, imagined context, ultimately leading it to discard the correct answer in favor of a flawed one. This example provides dramatic proof that overthinking is not a benign inefficiency but a pathological failure mode that can derail the entire analytical process. It demonstrates that the 'Overthinking Penalty' is not merely a regularizer for efficiency, but a critical component for ensuring robustness and correctness.

---

**Example 7: Grounded Acoustic Analysis vs. Contextual Hallucination**

**Question:** What is the sound event taking place in the audio?

**Choices:** A person crying, A person laughing, A person coughing, A person yawning

**Correct Answer:** A person laughing

**Task ID:** 01a7e75e-302a-464a-959c-6b3a04213d2f

| CESAR (Correct) | CESAR w/o OP (Wrong) |
|---|---|
| **Reasoning:** *The audio contains a series of sharp, rhythmic vocal bursts. The pitch and cadence are consistent with joyful laughter. It does not match the sound patterns of crying (more sustained wailing), coughing (a forceful expulsion of air), or yawning (a long, deep inhalation and exhalation). Therefore, the sound is a person laughing.* | **Reasoning:** *The audio clearly contains a human vocalization. It's a series of short, sharp, and somewhat rhythmic sounds. This could be interpreted as either laughing or crying, as both can have a similar staccato pattern. Let's analyze the context. While it sounds like laughter, this type of abrupt, almost gasping sound can also be associated with someone trying to suppress tears or crying in a very distressed way. Crying is a more primal and urgent emotional expression than laughter. Given the ambiguity, the more intense and emotionally significant possibility is often a safer bet in interpretation, as it might signify a situation requiring attention. The sound has a slightly strained quality that leans more towards distress than pure joy.* |
| **Answer:** A person laughing | **Answer:** A person crying |

---

**Evidence-Based Deduction vs. Over-analysis of Ambiguity.** The Example 7 provides a textbook case of a model "thinking itself into a mistake." Both models correctly identify the core acoustic features ("sharp, rhythmic vocal bursts"). The full CESAR model follows a direct and logical path: it compares these features against the prototypes of each option and correctly concludes they are "consistent with joyful laughter." Its reasoning is grounded entirely in the acoustic evidence.

The 'CESAR w/o OP' model, however, falls into a trap of pathological overthinking. After acknowledging the sound could be laughter, it begins to construct an elaborate, unfalsifiable alternative narrative. It introduces irrelevant external concepts like "*primal and urgent emotional expression*" and "*safer bet in interpretation*." It imagines a "strained quality" to support a "distress" hypothesis that is not strongly grounded in the audio. By lacking a penalty for this verbose and speculative detour, the

model gives undue weight to a complex, imagined scenario over the most direct interpretation of the sound itself. This demonstrates perfectly how the 'Overthinking Penalty' is crucial for keeping the model's reasoning tethered to evidence, preventing it from spiraling into contextual hallucinations that corrupt an initially correct perception.

### D.10 ABLATION STUDY: DECONSTRUCTING THE SOURCES OF IMPROVED REASONING CAPABILITY IN CESAR

To precisely quantify the contribution of each component within the CESAR framework, we conduct a progressive ablation study, systematically deconstructing our full model to isolate the impact of each design choice. The results, presented in Tab. 21, provide a comprehensive validation of our methodology, revealing not only that each component is effective, but also how they synergize to cultivate robust reasoning.

Table 21: Progressive Ablation Study Results. We systematically remove components from our full method to demonstrate their individual contributions. "Reasoning" refers to the chain-of-thought reasoning mechanism. Best scores are highlighted in blue , second-best scores in green .

| Method | Ablation Components | Reasoning? | RL Post-training | Technical Components | | | | Performance (%) | | | |
|---|---|---|---|---|---|---|---|---|---|---|---|
| | | | | Consistency | Key Words | Data Augmentation | Overthinking Penalty | Sound | Music | Speech | Total Accuracy |
| **Our Proposed Methods** | | | | | | | | | | | |
| CESAR | None (Full Method) | ✗ | ✓ | ✓ | ✓ | ✓ | ✓ | 79.88 | 67.96 | 73.27 | 73.70 |
| | | ✓ | ✓ | ✓ | ✓ | ✓ | ✓ | 83.48 | 73.05 | 74.77 | 77.10 |
| | Overthinking Penalty | ✗ | ✓ | ✓ | ✓ | ✓ | ✗ | 80.48 | 70.06 | 74.47 | 75.00 |
| | | ✓ | ✓ | ✓ | ✓ | ✓ | ✗ | 81.98 | 70.06 | 77.48 | 76.50 |
| | Data Augmentation | ✗ | ✓ | ✓ | ✓ | ✗ | ✗ | 80.18 | 68.86 | 75.38 | 74.80 |
| | | ✓ | ✓ | ✓ | ✓ | ✗ | ✗ | 82.28 | 68.86 | 77.48 | 76.20 |
| | Key Words | ✗ | ✓ | ✓ | ✗ | ✗ | ✗ | 80.48 | 66.77 | 74.77 | 74.00 |
| | | ✓ | ✓ | ✓ | ✗ | ✗ | ✗ | 80.78 | 68.86 | 75.98 | 75.20 |
| **Baseline Methods** | | | | | | | | | | | |
| Ke-Omni-R | Consistency | ✗ | ✓ | ✗ | ✗ | ✗ | ✗ | 78.38 | 70.96 | 74.17 | 74.50 |
| | | ✓ | ✓ | ✗ | ✗ | ✗ | ✗ | 79.28 | 70.06 | 74.47 | 74.60 |
| Qwen2.5-Omni-7B | RL Post-training | ✗ | ✗ | ✗ | ✗ | ✗ | ✗ | 72.37 | 64.37 | 69.07 | 68.60 |
| | | ✓ | ✗ | ✗ | ✗ | ✗ | ✗ | 69.07 | 59.58 | 66.97 | 65.20 |

**The Foundational Role of Process-Oriented RL.** The ablation starkly confirms that online reinforcement learning is the core mechanism transforming the model's fundamental capabilities. Removing RL post-training entirely—reverting to the base Qwen2.5-Omni-7B model—causes the most significant performance drop, a staggering 9.4 points in reasoning accuracy (from 74.60% of Ke-Omni-R to 65.20%). More critically, this is not merely a quantitative drop but a qualitative reversal: the base model is the only variant that exhibits test-time inverse scaling, where enabling reasoning **degrades** performance. In contrast, every single RL-trained variant sees a performance **gain** from reasoning. This demonstrates that RL is not an incremental improvement but the essential catalyst that turns reasoning from detriments into gains, a prerequisite for any further refinement.

**The Synergy of Process-Specific Rewards.** Our results clearly show that high-quality reasoning emerges from a synergy of process-oriented rewards, with each component providing a distinct and vital contribution. The most significant gains over the outcome-only RL baseline (Ke-Omni-R) come from our two core process rewards. First, removing the *Consistency* reward (which effectively reduces our model to the Ke-Omni-R baseline's level of process supervision) leads to a significant performance drop. As confirmed in our qualitative analysis (See App. D.9), a model lacking this reward can produce reasoning traces that are completely disconnected from the final answer. If reasoning is not required to be consistent with the answer, it becomes an unreliable, and potentially harmful, cognitive artifact. Second, removing the *Key Words* reward causes a further substantial drop of 1.0% in accuracy (from 76.20% to 75.20%). This demonstrates that explicitly rewarding logical structure and domain-specific terminology is a powerful driver of effective analytical strategies. Together, these rewards target distinct but complementary facets of high-quality reasoning—internal coherence and logical structure—proving that a multi-faceted approach is significantly more effective than optimizing for any single aspect alone.

**The Importance of Calibrated Training.** The final components, while having a smaller numerical impact, are crucial for calibrating and robustifying the learned skills. Although removing *Data Augmentation* only results in a minor accuracy drop, its role in exposing the model to linguistic diversity is essential for generalization and preventing the learning of superficial correlations in real-world scenarios. Most interestingly, the *Overthinking Penalty* serves a dual purpose. While its removal leads to a 0.6% performance drop, it also reveals a deeper insight into the value of reasoning. In our full method, the performance gap between reasoning and non-reasoning modes is 3.4 points (77.10% vs 73.70%). Without the penalty, this gap narrows to just 1.5 points (76.50% vs 75.00%). This shows that by penalizing inefficient thought, the model learns to better distinguish

when a simple, intuitive answer is insufficient and a more deliberate reasoning process is required. This enlarges the space where the cultivated reasoning skill provides a distinct advantage, underscoring the importance of not only knowing how to reason, but also when.

Ultimately, the ablation validates our holistic approach. Our framework comprehensively elevates performance in both non-reasoning and reasoning settings, with each component proving its value in building a model that is not only more accurate but reasons in a more effective, consistent, and logical manner.

D.11 BENCHMARK RESULTS ON MMAR

The MMAR benchmark, designed to test deep, multi-step reasoning on longer audio, exposes the limits of current models. Our analysis of the results in Tab. 22 reveals a clear fragmentation in reasoning capabilities across the field and highlights distinct challenges for future research.

Table 22: MMAR Benchmark Results. We evaluate our method against state-of-the-art audio models across single and mixed audio modalities. Best scores are highlighted in blue , second-best scores in green . All results show accuracy (%). Mix-S-M: Sound+Music, Mix-S-Sp: Sound+Speech, Mix-M-Sp: Music+Speech, Mix-S-M-Sp: Sound+Music+Speech. We report the performance of Ke-Omni-R (Zhao et al., 2025) from our own reproductions under the same protocol; all other baseline results are taken from the MMAR paper (Ma et al., 2025b) (including Qwen2.5-Omni-7B (Xu et al., 2025)).

| Method | Size | Sound | Music | Speech | Mix-S-M | Mix-S-Sp | Mix-M-Sp | Mix-S-M-Sp | Overall |
|---|---|---|---|---|---|---|---|---|---|
| **Our Proposed Method** | | | | | | | | | |
| **CESAR** | **7B** | **66.06** | **55.83** | 62.24 | 63.64 | 67.43 | 60.98 | 66.67 | 62.70 |
| **Base Model** | | | | | | | | | |
| Qwen2.5-Omni-7B | 7B | 58.79 | 40.78 | 59.86 | 54.55 | 61.93 | 67.07 | 58.33 | 56.70 |
| **Audio RL Baseline** | | | | | | | | | |
| Ke-Omni-R | 7B | 63.64 | 47.09 | 62.93 | 63.64 | 68.35 | 67.07 | 45.83 | 60.90 |
| **Proprietary Models** | | | | | | | | | |
| Gemini 2.0 Flash | - | 61.21 | 50.97 | 72.11 | 81.82 | 72.48 | 65.85 | 70.83 | 65.60 |
| GPT-4o Audio | - | 53.94 | 50.97 | 70.41 | 63.64 | 72.48 | 62.20 | 75.00 | 63.50 |
| GPT-4o mini Audio | - | 38.79 | 35.92 | 58.84 | 45.45 | 60.09 | 57.32 | 50.00 | 50.60 |
| Baichuan-Omni-1.5 | 11B | 41.21 | 33.01 | 40.48 | 36.36 | 48.62 | 39.02 | 41.67 | 40.70 |
| **Large Audio Reasoning Models (LARMs)** | | | | | | | | | |
| Audio-Reasoner | 8.4B | 43.64 | 33.50 | 32.99 | 45.45 | 42.66 | 31.71 | 25.00 | 36.80 |
| Audio-CoT | 8.4B | 35.76 | 25.24 | 34.01 | 9.09 | 30.73 | 30.49 | 37.50 | 31.30 |
| **Large Audio Language Models (LALMs)** | | | | | | | | | |
| Qwen2.5-Omni-3B | 3B | 53.94 | 46.12 | 53.74 | 36.36 | 60.09 | 57.32 | 58.33 | 53.80 |
| SALAMONN (13B) | 13B | 30.30 | 31.07 | 34.69 | 9.09 | 34.86 | 35.37 | 41.67 | 33.20 |
| SALAMONN (7B) | 7B | 30.91 | 29.61 | 34.35 | 9.09 | 37.61 | 28.05 | 37.50 | 32.80 |
| Audio Flamingo | 2.2B | 32.73 | 21.84 | 24.83 | 18.18 | 30.28 | 24.39 | 25.00 | 26.60 |
| Audio Flamingo 2 | 0.5B | 20.61 | 20.39 | 24.15 | 27.27 | 23.85 | 26.83 | 25.00 | 23.00 |
| Audio Flamingo 2 | 1.5B | 26.67 | 20.87 | 22.79 | 9.09 | 22.94 | 23.17 | 20.83 | 22.90 |
| Audio Flamingo 2 | 3B | 24.85 | 17.48 | 20.75 | 18.18 | 26.61 | 23.17 | 8.33 | 21.90 |
| LTU | 7B | 19.39 | 19.90 | 13.95 | 18.18 | 24.77 | 21.95 | 16.67 | 19.20 |
| LTU-AS | 7B | 20.00 | 14.08 | 19.05 | 9.09 | 20.64 | 28.05 | 12.50 | 19.00 |
| MusiLingo | 7B | 9.09 | 7.28 | 4.08 | 9.09 | 6.88 | 7.32 | 8.33 | 6.60 |
| MU-LLaMa | 7B | 13.94 | 13.59 | 14.97 | 9.09 | 12.39 | 14.63 | 16.67 | 13.90 |
| **Large Reasoning Models (LRMs) + Audio Caption** | | | | | | | | | |
| Caption + OpenAI o3 | - | 49.70 | 41.75 | 63.95 | 36.36 | 60.09 | 52.44 | 54.17 | 54.70 |
| Caption + DeepSeek-R1 | 671B | 46.67 | 49.51 | 62.59 | 45.45 | 58.72 | 56.10 | 54.17 | 55.50 |
| Caption + OpenAI o1 | - | 48.48 | 43.20 | 63.61 | 18.18 | 56.88 | 45.12 | 45.83 | 53.00 |
| Caption + GPT-4o | - | 46.06 | 40.29 | 60.88 | 27.27 | 53.67 | 46.34 | 45.83 | 50.70 |
| Caption + DeepSeek-V3 | 671B | 42.42 | 40.78 | 56.12 | 18.18 | 50.00 | 45.12 | 37.50 | 47.60 |
| **Random Baseline** | | | | | | | | | |
| Random Guess | - | 29.39 | 25.88 | 31.48 | 25.00 | 29.30 | 31.10 | 28.13 | 29.32 |

**The Acoustic-Linguistic Divide.** The MMAR results first expose a stark divergence between linguistic and acoustic reasoning capabilities across current models. While top proprietary systems show strong performance in tasks dominated by *Speech*, our model, CESAR, achieves the highest scores in the non-linguistic domains of *Sound* (66.06%) and *Music* (55.83%). This bifurcation is further illuminated by the "Caption + LRM" methods; their reliance on text transcripts allows them to perform reasonably well on speech-centric tasks but leaves them unable to compete on acoustic tasks where critical, non-transcribable information is paramount. This demonstrates that advanced audio reasoning is not a monolithic capability and that true progress requires models that can reason over the raw acoustic signal, not just its textual representation.

**Superiority of Process-Oriented Reinforcement Learning.** Second, the benchmark's difficulty serves as a critical test of training methodologies, revealing the limitations of prevalent supervised fine-tuning (SFT) paradigms. Our method, trained with process-oriented reinforcement learning, establishes a commanding lead over other Large Audio Reasoning Models (LARMs) like Audio-Reasoner, with a performance chasm of nearly 26 points (62.70% vs. 36.80%). Models trained via SFT on static CoT datasets prove to be brittle, failing to generalize to the complex, multi-hop reasoning required by MMAR. This vast performance gap strongly suggests that robust reasoning skills cannot be effectively learned through imitation alone; they require the interactive, process-focused feedback inherent to our RL framework.

**The Frontier of Mixed-Modality Reasoning.** Finally, MMAR underscores the profound challenge of reasoning over mixed-modality audio streams. Across the board, even top-performing models, including ours and leading proprietary systems, show high variance and struggle for consistent dominance in the "Mix-" categories. This indicates that while models may handle individual audio types, the compositional understanding and temporal grounding of multiple, overlapping audio sources (e.g., background music, foreground speech, and intermittent sound effects) remains a formidable challenge. This area represents the next clear frontier for the field of audio intelligence.

### D.12 QUANTIFYING TEST-TIME SCALING: LINEAR REGRESSION ANALYSIS

To rigorously quantify the test-time inverse scaling phenomenon, we perform linear regression analysis on the test-time scaling curves from Section 4. We model the relationship between accuracy ($P$) and average reasoning tokens ($L$) as: $P(L) = P(0) + \beta \cdot L$, where $P(0)$ is the non-reasoning baseline accuracy and the **Scaling Slope ($\beta$)** measures the performance change per additional reasoning token. This formulation directly captures whether reasoning improves or degrades performance relative to the baseline.

**Interpreting the Scaling Slope.**

- **Negative slope ($\beta < 0$):** test-time inverse scaling, where longer reasoning actively harms performance
- **Positive slope ($\beta > 0$):** effective test-time scaling, where reasoning improves performance
- **Zero slope ($\beta = 0$):** ineffective reasoning that fails to utilize additional compute

**Results and Analysis.** Figure 8 presents the linear fits for each model on MMAU Test-mini, with computed slopes quantitatively confirming our findings:

1. **Qwen2.5-Omni-7B (Base Model):** $\beta = -0.5083$. Severe test-time inverse scaling with a steep negative slope, showing catastrophic performance degradation as reasoning length increases. The model's accuracy drops from its non-reasoning baseline to less than one-third of its initial performance, confirming that under-optimized reasoning is actively harmful.

2. **Ke-Omni-R (RL Baseline):** $\beta = -0.0070$. Slightly negative slope showing a downward trend with volatile performance. The outcome-only rewards still fail to reverse the inverse scaling phenomenon, with performance gradually declining or stagnating around the non-reasoning baseline.

3. **CESAR (w/o Overthinking Penalty):** $\beta = +0.0084$. Positive slope demonstrating that process-oriented rewards enable effective reasoning. Performance steadily improves above the non-reasoning baseline, validating that rewarding the reasoning process transforms reasoning from a liability into a gain.

4. **CESAR (Full Method):** $\beta = +0.0383$. Strong positive slope—the most favorable scaling coefficient among all methods. Performance rises substantially above the non-reasoning baseline, reaching an optimal peak before gracefully stabilizing. The higher $\beta$ demonstrates that the Overthinking Penalty enhances both scaling efficiency and the magnitude of reasoning benefits.

**Efficiency Trade-offs.** We measured inference latency on MMAU Test-mini using a single NVIDIA H200 GPU. At the optimal "reasoning sweet spot" ($L \approx 35\text{-}40$ tokens), CESAR adds only **0.08 seconds (1.8% overhead)** per query compared to non-reasoning inference (4.39s vs. 4.47s). This negligible latency increase yields substantial accuracy gains (+3.4%), and the best scaling coefficient among all methods—demonstrating a highly favorable efficiency-performance trade-off.

In general, our linear regression analysis demonstrates that test-time inverse scaling is not inherent to reasoning but stems from inadequate training. By rewarding the reasoning process, CESAR transforms the scaling coefficient from severely negative ($\beta = -0.51$) to substantially positive ($\beta = +0.038$), enabling efficient and effective test-time scaling with clear optimal operating points.

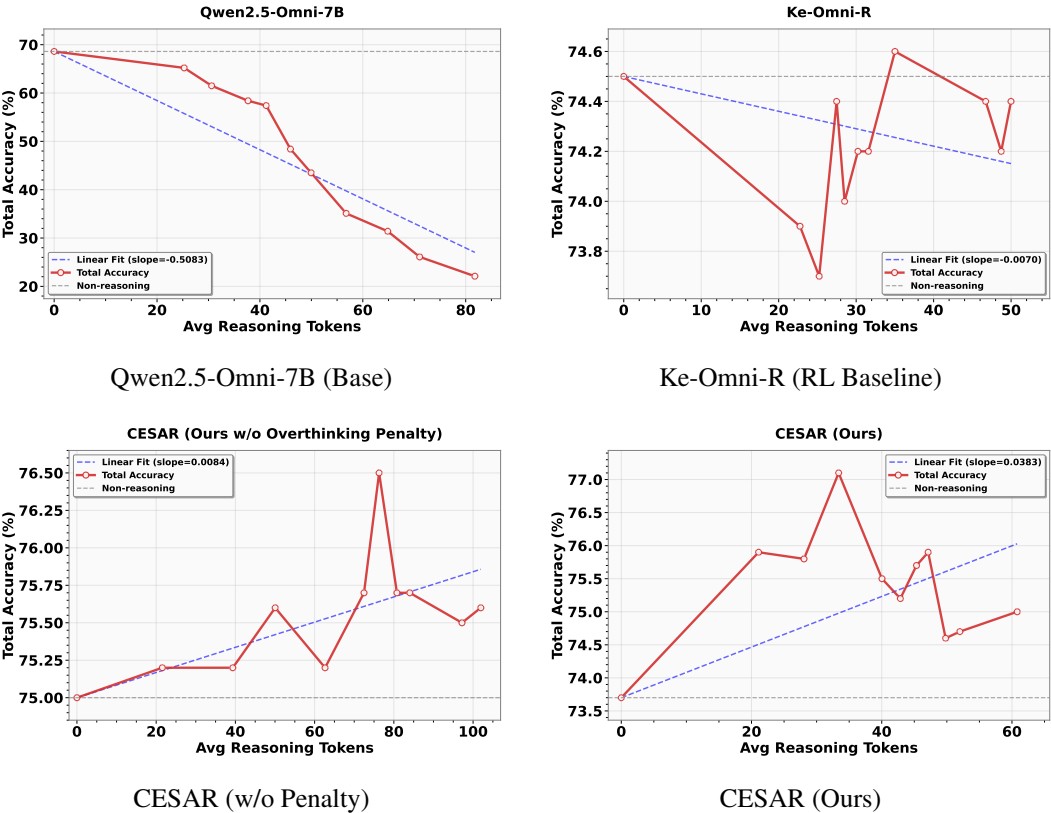

Figure 8: **Linear Regression Analysis of Test-Time Scaling.** We fit a linear model to the accuracy vs. reasoning tokens curve for each method. The **Scaling Slope** ($\beta$) quantifies the scaling behavior: a negative slope confirms test-time inverse scaling (Base Model), while a positive slope indicates effective reasoning scaling (CESAR). Note how CESAR reverses the negative trend of the base model into a positive trajectory.

### D.13 TRAINING DYNAMICS AND STABILITY

To provide further insight into the stability and convergence properties of our method, we visualize the training process in Figure 9. The curve plots the training accuracy over all training steps.

As shown in the figure, CESAR exhibits a stable and consistent learning trajectory. The model's performance (accuracy) improves steadily from an initial state, demonstrating effective optimization under our multi-faceted reward framework. The smoothed curve (red line) highlights a clear upward trend without significant oscillations or collapse, validating the robustness of our GRPO-based training with process-oriented rewards. This confirms that our approach not only achieves high final performance but also maintains training stability throughout the optimization process.

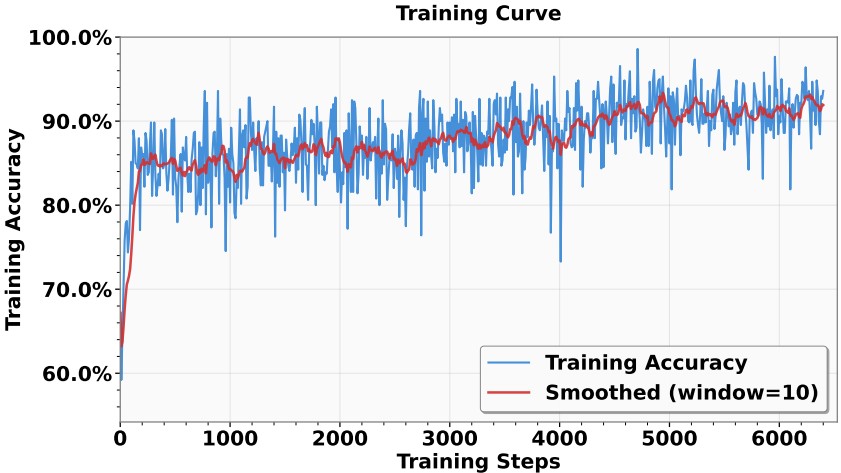

Figure 9: **Training Curve of CESAR.** The plot shows the training accuracy across all training steps. The light blue line represents the raw accuracy at each evaluation step, while the red line shows the smoothed trend (moving average). The consistent upward trajectory demonstrates the stability and effectiveness of our training process.

### D.14 HYPERPARAMETER SENSITIVITY ANALYSIS

To validate the robustness of our reward configuration and understand the impact of the accuracy reward weight, we conduct a sensitivity analysis by varying $\alpha_1$ (the weight for $R_{\text{acc}}$) while keeping other reward weights fixed at $\alpha_{2-5} = 1.0$. The results on MMAU Test-mini are presented in Table 23.

Table 23: Sensitivity Analysis of Accuracy Reward Weight ($\alpha_1$). We evaluate CESAR with different values of $\alpha_1$ while keeping all other reward weights at 1.0. Best scores are highlighted in blue . All results show accuracy (%).

| Alpha Configuration | Sound | Music | Speech | Overall |
|---|---|---|---|---|
| $\alpha_1 = 1, \alpha_{2-5} = 1$ | 82.58 | 70.36 | 75.08 | 76.00 |
| $\alpha_1 = 5, \alpha_{2-5} = 1$ (main) | **83.48** | **73.05** | 74.77 | **77.10** |
| $\alpha_1 = 10, \alpha_{2-5} = 1$ | 79.88 | 71.86 | **76.88** | 76.20 |

**Analysis.** The results reveal that our choice of $\alpha_1 = 5$ achieves the optimal balance between answer correctness and reasoning process quality. When $\alpha_1 = 1$ (equal weighting), the model achieves competitive but slightly lower overall performance (76.00%), suggesting that stronger emphasis on correctness is beneficial. Conversely, when $\alpha_1 = 10$ (excessive emphasis on correctness), performance degrades to 76.20%, indicating that over-prioritizing final answer correctness at the expense of process rewards can harm the model's ability to develop robust reasoning strategies.

This analysis confirms that our main configuration ($\alpha_1 = 5$) provides the most effective trade-off, allowing the model to prioritize correctness while still benefiting substantially from the reasoning process rewards. The relatively stable performance across different $\alpha_1$ values also demonstrates the robustness of our framework to hyperparameter variations.

