# OpenReview forum: "Incentivizing Consistent, Effective and Scalable Reasoning Capability in Audio LLMs via Reasoning Process Rewards"
_ICLR.cc/2026/Conference — ICLR 2026 Poster_

### Official Review · Reviewer_Rzte · 2025-10-17

**Soundness:** 2
**Presentation:** 2
**Contribution:** 3
**Rating:** 6
**Confidence:** 5

**Summary:**

This paper presents CESAR, a reinforcement learning framework that improves reasoning in Audio LLMs by rewarding the reasoning process rather than only the final answer. The authors identify a new phenomenon "test-time inverse scaling", where longer reasoning chains hurt performance, and attribute it to the lack of process-level supervision. CESAR uses Group Relative Policy Optimization (GRPO) with multiple reward components to train models that reason coherently and efficiently. Experiments on MMAU and MMSU show state-of-the-art results, outperforming GPT-4o Audio and Gemini 2.5 Pro, and reveal “reasoning sweet spots” that improve with controlled reasoning depth.

**Strengths:**

(1) The authors identify and analyze test-time inverse scaling in Audio LLMs, a previously undocumented failure mode.

(2) The authors propose a process-oriented RL reward design that explicitly improves reasoning quality and depth.

**Weaknesses:**

(1) The notion of test-time inverse scaling is introduced conceptually but not formally defined or quantified. The paper does not provide a measurable coefficient or slope to characterize this phenomenon. Moreover, there is no analysis of computational cost or latency as a function of reasoning length, which would clarify efficiency trade-offs in scaling reasoning depth.

(2) The paper focuses on reasoning accuracy but ignores word-error rate (WER) or fluency degradation, which could affect real-world usability. No fluency-aware or perceptual rewards are explored.

(3) The comparisons include Qwen2.5-Omni, Ke-Omni-R, GPT-4o Audio, and Gemini 2.5 Pro, but omit other competitive open-source models such as Qwen2-Audio [1], Audio Flamingo 3 [2], and Baichuan-Omni-1.5 [3]. Adding these baselines would better contextualize CESAR’s gains and clarify the limits of proprietary systems.

(4) The GPT-4o Audio “AI-as-Judge” evaluation lacks statistical rigor. The paper does not report confidence intervals, sample sizes, or randomization procedures, and it relies solely on GPT-4o Audio without human verification or alternative evaluators. Using multiple judgment models or human raters would help mitigate potential bias and confirm alignment with human judgment.


(5) Experiments are conducted primarily on MMAU Test-mini, which is a small subset of MMAU-Pro [4], and on MMSU. Although the paper briefly mentions testing on the full MMAU-Pro [4] benchmark, detailed results are not reported. The representativeness of the subset is unclear, which weakens claims of scalability and generalization. Evaluating CESAR on additional benchmarks with Audio-CoT [5], SoundMind [6], CoTA [7] would substantiate its robustness.


(6) The paper omits key implementation details, including training steps, learning rates, sampling temperature, and optimizer configuration. Moreover, no GRPO stability curve or convergence analysis is provided, making it difficult to assess the reliability and reproducibility of the proposed reinforcement learning setup.

**References**:

[1] Qwen2-Audio: https://arxiv.org/pdf/2407.10759

[2] Audio Flamingo 3: https://arxiv.org/pdf/2507.08128

[3] Baichuan-Omni-1.5: https://arxiv.org/pdf/2501.15368

[4] MMAU-Pro: https://huggingface.co/datasets/gamma-lab-umd/MMAU-Pro

[5] Audio-CoT: https://huggingface.co/datasets/liuhuadai/AudioCoT

[6] SoundMind: https://huggingface.co/datasets/SoundMind-RL/SoundMindDataset

[7] CoTA: https://huggingface.co/datasets/zhifeixie/Audio-Reasoner-CoTA

**Questions:**

I believe this paper has strong merit and makes an original contribution to the study of reasoning in Audio LLMs. If the authors can address the following questions and provide the requested analyses or experiments, I will raise my overall score and support acceptance of this paper.

(1) Could the authors formally define test-time inverse scaling and provide a quantitative measure (e.g., slope or scaling coefficient) to characterize this phenomenon? Please also include an analysis of computational cost or latency as a function of reasoning length to clarify efficiency trade-offs.

(2) How does CESAR affect word-error rate (WER) and audio fluency during reasoning generation? Have the authors considered introducing fluency-aware or perceptual rewards to mitigate potential degradation in real-world applications?

(3) Could the authors extend the experimental comparison to include recent open-source audio reasoning models with Qwen2-Audio [1], Audio Flamingo 3 [2], and Baichuan-Omni-1.5 [3]? Including these baselines would better contextualize CESAR’s gains and clarify its dependence on proprietary systems.

(4) Current experiments are primarily restricted to multiple-choice QA datasets (MMAU Test-mini and MMSU). To convincingly demonstrate CESAR’s generalization ability, could the authors evaluate the model on Audio-CoT [5],  SoundMind [6],  and CoTA [7]. Reporting results on these benchmarks would directly support the paper’s claims about scalability and robustness across diverse reasoning tasks.

(5) The AI-as-Judge evaluation currently relies solely on GPT-4o Audio. Would the authors consider adding results from multiple evaluators (e.g., Gemini 2.5 or human raters) to assess consistency and potential bias? Reporting confidence intervals and sample-size statistics would further improve reliability.

(6) Could the authors provide detailed hyperparameters (training steps, learning rates, sampling temperature, etc.) and include a GRPO stability curve to illustrate convergence behavior and training reliability?


**References**:

[1] Qwen2-Audio: https://arxiv.org/pdf/2407.10759

[2] Audio Flamingo 3: https://arxiv.org/pdf/2507.08128

[3] Baichuan-Omni-1.5: https://arxiv.org/pdf/2501.15368

[4] MMAU-Pro: https://huggingface.co/datasets/gamma-lab-umd/MMAU-Pro

[5] Audio-CoT: https://huggingface.co/datasets/liuhuadai/AudioCoT

[6] SoundMind: https://huggingface.co/datasets/SoundMind-RL/SoundMindDataset

[7] CoTA: https://huggingface.co/datasets/zhifeixie/Audio-Reasoner-CoTA

---

> ### Author Response · Authors · 2025-11-21
> **Reply to Reviewer Rzte [Part 1]**
>
> Thank you for the thoughtful and constructive review. We greatly appreciate your recognition of CESAR's original contribution to audio reasoning and your acknowledgment of our identification of test-time inverse scaling as a novel failure mode in Audio LLMs. We are particularly encouraged by your positive assessment of our process-oriented RL reward design for improving reasoning quality and depth.
>
> We understand your concerns focus primarily on formalizing inverse scaling measurement, expanding evaluation scope to include WER/fluency metrics and additional baselines/benchmarks, enhancing AI-as-judge robustness, and providing comprehensive implementation details.
>
> To address these thoroughly, we have conducted extensive additional experiments: (1) **formal quantification of test-time inverse scaling** with measurable linear scaling coefficients and computational cost analysis (Appendix D.12), (2) **evaluation of transcription-related capabilities** through MMSU's Content Grounding task (Appendix D.6), (3) **expanded evaluation to **MMAU Full Test Set [1]** (9k samples) and MMAU-Pro [2]** (5.3k samples) with comparisons to recent open-source models including Audio Flamingo 3, Baichuan-Omni-1.5, and Qwen2-Audio-Instruct (Appendix D.1, D.3), (4) **large-scale human evaluation study** with over 3,000 expert judgments validating our AI-as-judge findings (Appendix D.2), and (5) **detailed hyperparameter specifications** and GRPO training stability analysis (Appendix D.13, D.14). All additions have been integrated into our revised manuscript. Below, we address each specific question systematically.
>
> > **Q1**: The notion of test-time inverse scaling is introduced conceptually but not formally defined or quantified. The paper does not provide a measurable coefficient or slope to characterize this phenomenon. Moreover, there is no analysis of computational cost or latency as a function of reasoning length, which would clarify efficiency trade-offs in scaling reasoning depth.  Could the authors formally define test-time inverse scaling and provide a quantitative measure (e.g., slope or scaling coefficient) to characterize this phenomenon? Please also include an analysis of computational cost or latency as a function of reasoning length to clarify efficiency trade-offs.
>
>
> **A1**: Thank you for this insightful suggestion. Following your suggestion, we have formally defined test-time inverse scaling using linear scaling coefficients (slopes) and provided comprehensive quantitative analysis in our revised manuscript (Appendix D.12).
>
> **Formal Definition and Quantitative Measure.** We define **test-time inverse scaling** through the empirical relationship between test accuracy $P(L)$ and average reasoning length $L$: $P(L) \approx P(0) + \beta \cdot L$, where $P(0)$ is the non-reasoning baseline accuracy and $\beta$ is the linear scaling coefficient. Test-time inverse scaling occurs when $\beta < 0$, indicating that longer reasoning chains degrade performance. The magnitude of $|\beta|$ quantifies the severity of inverse scaling.
>
> **Quantitative Characterization.** We provide detailed quantitative analysis through linear regression in Appendix D.12 (Figure 8). The scaling coefficients clearly distinguish different model behaviors:
> - **Qwen2.5-Omni-7B (Base)**: $\beta = -0.5083$ (severe inverse scaling)
> - **Ke-Omni-R (Outcome-only RL)**: $\beta = -0.0070$ (slight negative scaling)
> - **CESAR w/o Overthinking Penalty**: $\beta = +0.0084$ (positive scaling)
> - **CESAR (Full Method)**: $\beta = +0.0383$ (strong positive scaling)
>
> CESAR successfully transforms the negative slopes of baseline models into substantially positive slopes, demonstrating that our process-oriented training resolves test-time inverse scaling and enables effective test-time scaling.
>
> **Computational Cost Analysis.** We measured inference latency on MMAU Test-mini (1,000 samples) using a single NVIDIA H200 GPU:
>
> | **Reasoning Length** | **Time per Sample** | **Overhead vs. Non-reasoning** |
> |---------------------|---------------------|--------------------------------|
> | $L=0$ (Non-reasoning) | 4.39 seconds | - |
> | $L\approx 35$-$40$ (Sweet spot) | 4.47 seconds | +0.08 seconds (+1.8%) |
>
> At the optimal reasoning length ($L\approx 35$-$40$ tokens), CESAR adds only 1.8% latency overhead while achieving +3.4% absolute accuracy improvement (77.1% vs. 73.7%, Table 1). This demonstrates a highly favorable efficiency trade-off, with negligible computational cost for substantial performance gains.
>
>
> **References**:
>
> [1] Sakshi, S., et al. "MMAU: A Massive Multi-Task Audio Understanding and Reasoning Benchmark." The Thirteenth International Conference on Learning Representations.
>
> [2] Kumar, Sonal, et al. "Mmau-pro: A challenging and comprehensive benchmark for holistic evaluation of audio general intelligence." arXiv preprint arXiv:2508.13992 (2025).

---

> > ### Comment · Reviewer_Rzte · 2025-11-22
> > **Response to Authors’ Rebuttal**
> >
> > Most of the concerns raised in my initial review have been adequately addressed in the authors’ rebuttal. Based on the additional information provided, I decided to update my rating to 8.

---

> ### Author Response · Authors · 2025-11-21
> **Reply to Reviewer Rzte [Part 2]**
>
> > **Q2**: The paper focuses on reasoning accuracy but ignores word-error rate (WER) or fluency degradation, which could affect real-world usability. No fluency-aware or perceptual rewards are explored. How does CESAR affect word-error rate (WER) and audio fluency during reasoning generation? Have the authors considered introducing fluency-aware or perceptual rewards to mitigate potential degradation in real-world applications?
>
> **A2**: Thank you for this question. We would like to clarify the scope of our work and address your concerns.
>
> **Task Focus: Audio Reasoning, Not Audio Generation.** As stated in our title, our work focuses on improving **reasoning capabilities in Audio LLMs** through audio question answering tasks. We evaluate models on established audio reasoning benchmarks such as MMAU, MMSU, MMAR, and MMAU-Pro, which directly measure audio reasoning abilities through QA format. Our research does not involve audio generation or synthesis tasks, where fluency-related metrics would be relevant. Since we generate text-based reasoning traces and answers rather than audio content, audio fluency metrics are not applicable to our task setting. Audio generation could be an interesting direction for future work, but it is beyond the scope of this paper.
>
> **Assessing Transcription-Related Capabilities.** Regarding WER and speech understanding capabilities, we observe in our results that CESAR achieves joint improvements in both perception and reasoning on MMSU (Table 2). To further address this concern and validate improvements in foundational speech perception and transcription-related tasks, we evaluate on MMSU's **Content Grounding task**, which directly measures accurate speech understanding through questions like "Which sentence is the correct transcription of the audio?" (detailed in Appendix D.6, Table 20):
>
> | **Model** | **Content Grounding Accuracy** |
> |-----------|-------------------------------|
> | **CESAR (Ours)** | **90.80%** |
> | Ke-Omni-R | 72.50% |
> | Qwen2.5-Omni-7B | 59.60% |
>
> CESAR achieves **90.80% accuracy**, substantially outperforming baselines by 18.3 and 31.2 points respectively. These results demonstrate that our process-oriented training not only enhances audio reasoning capabilities but also improves foundational speech understanding and transcription-related performance, validating the effectiveness of our approach across multiple dimensions of audio language understanding.

---

> ### Author Response · Authors · 2025-11-21
> **Reply to Reviewer Rzte [Part 3]**
>
> > **Q3**: The comparisons include Qwen2.5-Omni, Ke-Omni-R, GPT-4o Audio, and Gemini 2.5 Pro, but omit other competitive open-source models such as Qwen2-Audio [1], Audio Flamingo 3 [2], and Baichuan-Omni-1.5 [3]. Adding these baselines would better contextualize CESAR's gains and clarify the limits of proprietary systems. Could the authors extend the experimental comparison to include recent open-source audio reasoning models with Qwen2-Audio [1], Audio Flamingo 3 [2], and Baichuan-Omni-1.5 [3]? Including these baselines would better contextualize CESAR's gains and clarify its dependence on proprietary systems.
>
>
> **A3**: Thank you for this valuable suggestion. To address your concern, we have significantly expanded our experimental comparisons by adding complete results on two large-scale benchmarks—MMAU Full Test Set and MMAU-Pro—which include the requested open-source baselines as well as 20+ additional competitive models (see Appendix D.1 and D.3 for complete comparisons).
>
> **Extended Comparisons on MMAU-Pro Benchmark.** We provide comprehensive results on the challenging MMAU-Pro benchmark (Appendix D.1, Table 10), which includes comparisons with **Audio Flamingo 3** and **Baichuan-Omni-1.5** among many other competitive models:
>
> | **Model** | **Average Accuracy (%)** |
> |-----------|-------------------------|
> | **CESAR (Ours)** | **56.4** |
> | Gemini 2.5 Flash | **59.2** |
> | Gemini 2.0 Flash | 55.7 |
> | Ke-Omni-R | 54.5 |
> | GPT-4o Audio | 52.5 |
> | **Audio Flamingo 3** | 51.7 |
> | Qwen2.5-Omni-7B (Base) | 49.1 |
> | **Baichuan-Omni-1.5** | 33.9 |
>
> CESAR achieves 56.4% average accuracy, establishing itself as the **top-performing 7B-parameter model**, substantially outperforming Audio Flamingo 3 (+4.7%) and Baichuan-Omni-1.5 (+22.5%), while surpassing even large-scale proprietary models like Gemini 2.0 Flash.
>
> **Extended Comparisons on MMAU Full Test Set.** We have also added results on the complete MMAU Full Test Set with 9,000 samples (Appendix D.3, Table 17), including comparisons with **Audio Flamingo 3** and **Qwen2-Audio-Instruct**:
>
> | **Model** | **Overall Accuracy (%)** |
> |-----------|-------------------------|
> | **CESAR (Ours)** | **73.79** |
> | **Audio Flamingo 3** | 72.42 |
> | Ke-Omni-R | 71.94 |
> | Qwen2.5-Omni-7B | 64.20 |
> | **Qwen2-Audio-Instruct** | 57.40 |
>
> CESAR achieves state-of-the-art performance (73.79%), outperforming Audio Flamingo 3 (+1.37%) and substantially surpassing Qwen2-Audio-Instruct (+16.39%).
>
> These extended comparisons across multiple large-scale benchmarks and against diverse open-source models consistently validate CESAR's effectiveness and demonstrate that our process-oriented reasoning framework achieves superior performance on different audio reasoning tasks.

---

> ### Author Response · Authors · 2025-11-21
> **Reply to Reviewer Rzte [Part 4]**
>
> > **Q4**: The GPT-4o Audio "AI-as-Judge" evaluation lacks statistical rigor. The paper does not report confidence intervals, sample sizes, or randomization procedures, and it relies solely on GPT-4o Audio without human verification or alternative evaluators. Using multiple judgment models or human raters would help mitigate potential bias and confirm alignment with human judgment. The AI-as-Judge evaluation currently relies solely on GPT-4o Audio. Would the authors consider adding results from multiple evaluators (e.g., Gemini 2.5 or human raters) to assess consistency and potential bias? Reporting confidence intervals and sample-size statistics would further improve reliability.
>
> **A4**: Thank you for this important concern about evaluation rigor. To comprehensively address potential bias from relying on a single AI judge, we have conducted a large-scale human evaluation study that provides gold standard validation.
>
> **AI-as-Judge Rationale.** Our use of GPT-4o Audio as an evaluator provides a strong initial assessment, as it is widely adopted in the research community and can directly process audio inputs, making it particularly suitable for judging audio reasoning quality.
>
> **Large-Scale Human Evaluation with Rigorous Protocol.** To definitively address your concern and provide the most accurate assessment of reasoning capability, we conducted comprehensive human evaluation on the **entire 1,000-sample MMAU Test-mini benchmark**, resulting in **over 3,000 individual expert judgments** (3 independent annotators per sample). The evaluation employed rigorous statistical protocols: randomized presentation order, blind evaluation (annotators had no knowledge of correct answers or model identities), and majority-vote aggregation across three annotators for robust consensus.
>
> **Human Evaluation Results Strongly Validate CESAR's Superiority.** Overall win rates (majority-vote protocol, Appendix D.2):
>
> | **Model Comparison** | **CESAR Win** | **Baseline Win** | **Tie** |
> |---------------------|---------------|------------------|---------|
> | CESAR vs. Qwen2.5-Omni-7B (Base) | **88.6%** | 6.6% | 4.8% |
> | CESAR vs. Ke-Omni-R (RL Baseline) | **63.1%** | 14.8% | 22.1% |
>
> Human experts overwhelmingly preferred CESAR's reasoning, providing clear evidence that rewarding the reasoning process yields superior reasoning quality. The strong agreement between GPT-4o Audio AI-as-judge and comprehensive human evaluation validates the robustness of our findings.
>
> **Universal Superiority Across All 27 Sub-Categories.** CESAR outperforms Ke-Omni-R in all 27 reasoning sub-categories (complete results in Appendix D.2, Table 16):
>
> | **Sub-Category** | **CESAR** | **Ke-Omni-R** | **Tie** |
> |-----------------|-----------|---------------|---------|
> | Acoustic Scene Reasoning | **62.50** | 25.00 | 12.50 |
> | Acoustic Source Inference | **47.92** | 33.33 | 18.75 |
> | Ambient Sound Interpretation | **79.17** | 6.25 | 14.58 |
> | Conversational Fact Retrieval | **50.00** | 31.82 | 18.18 |
> | Counting | **41.38** | 31.03 | 27.59 |
> | Dissonant Emotion Interpretation | **60.00** | 17.14 | 22.86 |
> | Eco-Acoustic Knowledge | **72.34** | 4.26 | 23.40 |
> | Emotion Flip Detection | **85.00** | 0.00 | 15.00 |
> | Emotion State Summarisation | **59.09** | 15.91 | 25.00 |
> | Emotional Tone Interpretation | **75.76** | 12.12 | 12.12 |
> | Event-Based Knowledge Retrieval | **48.48** | 18.18 | 33.33 |
> | Event-Based Sound Reasoning | **72.92** | 6.25 | 20.83 |
> | Harmony and Chord Progressions | **72.73** | 12.12 | 15.15 |
> | Instrumentation | **74.29** | 14.29 | 11.43 |
> | Key Highlight Extraction | **61.90** | 19.05 | 19.05 |
> | Lyrical Reasoning | **70.00** | 0.00 | 30.00 |
> | Melodic Structure Interpretation | **57.58** | 18.18 | 24.24 |
> | Multi-Speaker Role Mapping | **44.44** | 3.70 | 51.85 |
> | Musical Genre Reasoning | **67.65** | 14.71 | 17.65 |
> | Musical Texture Interpretation | **70.59** | 8.82 | 20.59 |
> | Phonemic Stress Pattern Analysis | **73.58** | 7.55 | 18.87 |
> | Phonological Sequence Decoding | **59.18** | 18.37 | 22.45 |
> | Rhythm and Tempo Understanding | **56.52** | 10.87 | 32.61 |
> | Socio-Cultural Interpretation | **75.00** | 10.00 | 15.00 |
> | Sound-Based Event Recognition | **52.17** | 10.87 | 36.96 |
> | Temporal Event Reasoning | **75.00** | 12.50 | 12.50 |
> | Temporal Reasoning | **46.43** | 25.00 | 28.57 |
>
> This universal superiority across all task types—from acoustic scene understanding to musical analysis to speech reasoning—demonstrates systematic improvements rather than task-specific gains.

---

> ### Author Response · Authors · 2025-11-21
> **Reply to Reviewer Rzte [Part 5]**
>
> > **Q5**: Experiments are conducted primarily on MMAU Test-mini, which is a small subset of MMAU-Pro [4], and on MMSU. Although the paper briefly mentions testing on the full MMAU-Pro [4] benchmark, detailed results are not reported. The representativeness of the subset is unclear, which weakens claims of scalability and generalization. Evaluating CESAR on additional benchmarks with Audio-CoT [5], SoundMind [6], CoTA [7] would substantiate its robustness. Current experiments are primarily restricted to multiple-choice QA datasets (MMAU Test-mini and MMSU). To convincingly demonstrate CESAR's generalization ability, could the authors evaluate the model on Audio-CoT [5], SoundMind [6], and CoTA [7]. Reporting results on these benchmarks would directly support the paper's claims about scalability and robustness across diverse reasoning tasks.
>
> **A5**: Thank you for this valuable suggestion. We would like to clarify our task scope and provide evidence of our comprehensive evaluation.
>
> **Task Scope and Benchmark Selection.** Our work focuses on **audio reasoning and question answering tasks**, where models analyze audio content and generate text-based reasoning and answers. Regarding the suggested benchmarks: Audio-CoT is designed for **audio generation and editing tasks**, which is fundamentally different from our task setting. CoTA is a **training dataset** (and notably, the original CoTA paper also evaluates on MMAU Test-mini, which we already include). While SoundMind is an audio QA dataset, we have already conducted extensive evaluation on multiple established audio reasoning benchmarks with over 21,000 out-of-distribution test samples (Appendix D.1, D.3, D.4, D.6, D.11), which comprehensively validates our method's effectiveness.
>
> **Comprehensive Evaluation Across Five Benchmarks.** We have conducted extensive evaluation across **five diverse, challenging, and fully out-of-distribution audio reasoning benchmarks**, totaling over **21,000 test samples**:
>
> 1. **MMAU Test-mini** (1,000 samples): 27 distinct reasoning skills (Appendix D.4)
> 2. **MMSU** (5,000 samples): 47 distinct tasks with perception-reasoning separation (Appendix D.6)
> 3. **MMAR** (1,000 samples): Deep reasoning in speech, audio, and music (Appendix D.11)
> 4. **MMAU Full Test Set** (9,000 samples): Complete large-scale evaluation (Appendix D.3)
> 5. **MMAU-Pro** (5,305 samples): Highly challenging "in-the-wild" multi-hop reasoning (Appendix D.1)
>
> **Expanded Results on Large-Scale Benchmarks.** To address your concern about scalability and generalization, we provide comprehensive results on the two larger benchmarks:
>
> **MMAU Full Test Set Results (9,000 samples, Appendix D.3, Table 17)**
>
> | **Method** | **Sound** | **Speech** | **Music** | **Overall** |
> |-----------|-----------|------------|-----------|-------------|
> | **CESAR (Ours)** | **77.60** | **76.09** | 67.77 | **73.79** |
> | Audio Flamingo 3 | 75.83 | 66.97 | **74.47** | 72.42 |
> | Ke-Omni-R | 75.37 | 73.77 | 66.73 | 71.94 |
> | Qwen2.5-Omni-7B | 68.87 | 68.11 | 55.77 | 64.2 |
> | Gemini 2.5 Pro | 70.63 | 72.67 | 64.77 | 69.36 |
> | GPT-4o Audio | 63.20 | 69.33 | 49.93 | 60.82 |
>
> **MMAU-Pro Results (5,305 samples, Appendix D.1, Table 10)**
>
> | **Method** | **Average Accuracy (%)** |
> |-----------|-------------------------|
> | **CESAR (Ours)** | **56.4** |
> | Gemini 2.5 Flash | **59.2** |
> | Gemini 2.0 Flash | 55.7 |
> | Ke-Omni-R | 54.5 |
> | GPT-4o Audio | 52.5 |
> | Audio Flamingo 3 | 51.7 |
> | Qwen2.5-Omni-7B | 49.1 |
>
> CESAR achieves state-of-the-art or highly competitive performance across all five benchmarks, spanning diverse reasoning scenarios, audio modalities, and difficulty levels. The consistency of our advantages across 21,000+ test samples from multiple independent benchmarks provides strong evidence of scalability, robustness, and generalization capability.

---

> ### Author Response · Authors · 2025-11-21
> **Reply to Reviewer Rzte [Part 6]**
>
> > **Q6**: The paper omits key implementation details, including training steps, learning rates, sampling temperature, and optimizer configuration. Moreover, no GRPO stability curve or convergence analysis is provided, making it difficult to assess the reliability and reproducibility of the proposed reinforcement learning setup. Could the authors provide detailed hyperparameters (training steps, learning rates, sampling temperature, etc.) and include a GRPO stability curve to illustrate convergence behavior and training reliability?
>
> **A6**: Thank you for this question regarding implementation details and training stability. To address your concern, we have provided training stability analysis in our revised manuscript.
>
> **Hyperparameter Consistency for Fair Comparison.** As stated in Section 4.1, to ensure fair comparison and isolate the impact of our core contribution (reasoning process rewards), we intentionally keep all key hyperparameters consistent with the baseline method Ke-Omni-R, including training steps, learning rates, sampling temperature, optimizer configuration (GRPO with K=8), KL penalty coefficient $\beta$, and maximum output length (L_max_output=256). This controlled experimental design ensures that performance differences can be attributed to our reasoning process rewards rather than confounding hyperparameter variations.
>
> **Training Stability Validation.** To address your concern about training reliability and reproducibility, we have added the GRPO training stability curve in **Appendix D.13** (Figure 9). The curve demonstrates stable convergence behavior, with training accuracy progressively improving from ~60% to ~92% and stabilizing throughout the training process. The smoothed trend line clearly shows consistent learning progress without instability or divergence, confirming the reliability and reproducibility of our reinforcement learning setup.

---

> ### Author Response · Authors · 2025-11-23
>
> Dear Reviewer Rzte,
>
> We are delighted that our responses and additional experiments have **adequately addressed your concerns**. Thank you sincerely for your willingness to raise the rating to 8 and for your valuable and constructive feedback, which has helped us significantly strengthen our paper through formal quantification of test-time inverse scaling, large-scale human evaluations, expanded benchmark comparisons, and detailed training stability analysis. We truly appreciate your thoughtful review throughout this process.

---

> ### Author Response · Authors · 2025-11-27
>
> Dear Reviewer Rzte,
>
> Thank you once again for your thoughtful engagement and positive feedback during the review and discussion. We truly appreciate your comment mentioning that "Most of the concerns raised in my initial review have been adequately addressed in the authors' rebuttal. Based on the additional information provided, I decided to update my rating to 8."
>
> We wanted to kindly follow up regarding the rating update. *We noticed that the rating may not have been updated yet in the system*. We completely understand that the review period can be busy and wanted to gently check if there might have been any technical issues with submitting the updated rating or other possible causes.
>
> We sincerely appreciate your willingness to raise the rating from 6 to 8 based on our rebuttal.
>
> Thank you again for your valuable time and constructive feedback throughout this process.
>
> Best regards,
>
> The Authors

---

> > ### Comment · Reviewer_Rzte · 2025-11-28
> >
> > As noted in my Official Comment, I have already updated the score to 8 — and if you were in the reviewer interface, you’d see that there is no longer a place to modify the original reviews at this stage.

---

> > > ### Author Response · Authors · 2025-11-28
> > >
> > > Dear Reviewer Rzte,
> > >
> > > Thank you so much for your prompt reply regarding the rating update. We truly appreciate you taking the time to confirm that the score has already been updated to 8 in the Official Comment, and that the reviewer interface no longer allows modifications to the original reviews at this stage currently.
> > >
> > > We are delighted that our rebuttal was able to address your concerns, and we deeply appreciate your willingness to raise the rating accordingly. Your constructive feedback throughout this process has been invaluable in helping us strengthen our work.
> > >
> > > Thank you again for your valuable time, thoughtful engagement, and continued support.
> > >
> > > Best regards,
> > >
> > > The Authors

---

### Official Review · Reviewer_WEba · 2025-10-29

**Soundness:** 3
**Presentation:** 3
**Contribution:** 3
**Rating:** 8
**Confidence:** 3

**Summary:**

This paper tackles "test-time inverse scaling" in Audio LLMs, where longer reasoning chains degrade performance. The authors propose CESAR, an RL framework that rewards the "reasoning process" itself, not just the final answer. This method incentivizes consistency, structured logic, and penalizes "overthinking". CESAR successfully resolves this issue , achieving SOTA results on the MMAU benchmark and surpassing Gemini 2.5 Pro.

**Strengths:**

1. The paper is technically sound. Its claims about solving "test-time inverse scaling" are strongly supported by extensive experiments, including state-of-the-art (SOTA) comparisons, comprehensive ablation studies, and "test-time scaling" analysis.
2. The paper is exceptionally well-organized with a clear, logical narrative. It provides a high degree of transparency, including pseudocode and detailed keyword tables, which significantly aids reader understanding and the ability to reproduce the results.
3. The work solves a core problem for Audio LLMs, offering a new, engineerable paradigm for reasoning. Its most significant impact may be diagnostic: by solving reasoning, it clearly identifies the field's next major barrier—the "perceptual bottleneck"—which provides crucial guidance for future research.

**Weaknesses:**

1. The proposed multi-faceted reward suite introduces five new reward weights ($\alpha_j$) that must be balanced and tuned. While the authors provide a simple, effective ratio, this still represents an added layer of complexity compared to simpler reward models.
2. The readability of Figure 1 is poor. The text within the "Qualitative Comparison of Reasoning Process" section , which is meant to show examples of reasoning failures and improvements, is far too small. This makes it very difficult for the reader to understand these crucial illustrative examples, undermining the figure's purpose.

**Questions:**

1. How were the final reward weights (e.g., $\alpha_1=5.0$ for accuracy, $\alpha_{2-5}=1.0$ for process rewards) determined? How sensitive is the model's performance to variations in these weights, and is there a systematic approach to balancing these distinct reward objectives?
2. Since the CESAR reward suite lacks explicit ASR or WER metrics, how does the training impact the model's ASR or WER? Does the "synergistic effect" seen in perception tasks (Table 9) also translate to a measurable reduction in WER?

---

> ### Author Response · Authors · 2025-11-21
> **Reply to Reviewer WEba [Part 1]**
>
> Thank you for this thoughtful and constructive review. We greatly appreciate your recognition that CESAR successfully resolves the test-time inverse scaling problem with strong empirical support, and your positive assessment of the paper's technical soundness, clear presentation, and significant contribution in identifying the perceptual bottleneck as the field's next barrier.
>
> We understand your concerns focus primarily on reward weight determination and sensitivity, the readability of Figure 1, and the impact on ASR/WER-related performance.
>
> To address these thoroughly, we have conducted comprehensive additional analysis: (1) **systematic reward weight sensitivity experiments** evaluating different $\alpha$ configurations on MMAU Test-mini (Appendix D.14), (2) **direct measurement of transcription-related improvements** using MMSU's Content Grounding task as an effective proxy for ASR/WER performance (Appendix D.6), and (3) **improved Figure 1 with larger font sizes for enhanced readability**. All additions and revisions have been integrated into our revised manuscript. Below, we address each specific question systematically.
>
>
> > **Q1**: The proposed multi-faceted reward suite introduces five new reward weights $\left(\alpha_j\right)$ that must be balanced and tuned. While the authors provide a simple, effective ratio, this still represents an added layer of complexity compared to simpler reward models. How were the final reward weights (e.g., $\alpha_1=5.0$ for accuracy, $\alpha_{2-5}=1.0$ for process rewards) determined? How sensitive is the model's performance to variations in these weights, and is there a systematic approach to balancing these distinct reward objectives?
>
> **A1**: Thank you for this important question about reward weight determination and sensitivity.
>
> **Principled Weight Selection Strategy.** As stated in Section 4.1, our reward weights follow a principled design: $\alpha_1=5.0$ for accuracy and $\alpha_{2-5}=1.0$ for all process rewards. This 5:1 ratio is guided by a clear principle—prioritize correctness as the fundamental objective while equally weighting process improvements. In practice, we find that using a relatively larger accuracy weight (within a reasonable range, such as our 5:1 ratio) helps stabilize training by ensuring the model maintains strong performance on the primary task while learning to improve reasoning quality. This prevents the failure mode of "sacrificing substance for form" where the model might generate elaborate reasoning at the expense of correctness. Importantly, when accuracy optimization encounters a plateau, further improvements in other rewards—such as reasoning process quality and format compliance—can synergistically help break through accuracy bottlenecks. Therefore, we recommend starting with a well-balanced weight configuration (such as the one presented in our work) to achieve effective training results.
>
> **Systematic Validation of Weight Sensitivity.** Our progressive ablation study (Appendix D.10, Table 21) systematically validates each reward component by progressively removing them, showing clear additive contributions and effectively demonstrating the impact of setting each α weight from 1 to 0. To further validate the robustness of our weight selection, we conducted experiments with different $\alpha_1$ values on MMAU Test-mini (detailed in Appendix D.14, Table 23):
>
> | **Alpha Configuration** | **Sound** | **Music** | **Speech** | **Overall** |
> |------------------------|-----------|-----------|------------|-------------|
> | $\alpha_1=5, \alpha_{2-5}=1$ (main) | 83.48% | 73.05% | 74.77% | **77.10%** |
> | $\alpha_1=1, \alpha_{2-5}=1$ | 82.58% | 70.36% | 75.08% | 76.00% |
> | $\alpha_1=10, \alpha_{2-5}=1$ | 79.88% | 71.86% | 76.88% | 76.20% |
>
> The results demonstrate that both $\alpha_1=1$ (insufficient accuracy emphasis, 76.00%) and $\alpha_1=10$ (excessive accuracy weight that diminishes process rewards, 76.20%) underperform compared to our selected $\alpha_1=5$ (77.10%). This validates that our 5:1 ratio achieves an effective balance between correctness and reasoning quality. The robustness of this weighting scheme is further confirmed by consistent state-of-the-art performance across all five benchmarks (Appendix D.1, D.3, D.4, D.6, D.11, spanning 21,000+ test samples) using the same fixed weights, demonstrating practical effectiveness without requiring complex tuning procedures or benchmark-specific adjustments.

---

> ### Author Response · Authors · 2025-11-21
> **Reply to Reviewer WEba [Part 2]**
>
> > **Q2**: The readability of Figure 1 is poor. The text within the "Qualitative Comparison of Reasoning Process" section , which is meant to show examples of reasoning failures and improvements, is far too small. This makes it very difficult for the reader to understand these crucial illustrative examples, undermining the figure's purpose.
>
> **A2**: Thank you for this valuable feedback regarding the readability of Figure 1.
>
> **We have improved the figure clarity in the revised manuscript.** We appreciate your suggestion and have increased the font size in the "Qualitative Comparison of Reasoning Process" section of Figure 1 to ensure the reasoning examples are clearly legible. These examples are indeed crucial for illustrating the concrete improvements our method achieves—specifically demonstrating how CESAR addresses critical failure modes including factual hallucination, flawed logic, reasoning-answer inconsistency, and redundant reasoning (as detailed in Table 5 and Appendix D.9). We have ensured they are now easily readable in the revision. Thank you again for this constructive feedback.
>
> ---
>
> > **Q3**: Since the CESAR reward suite lacks explicit ASR or WER metrics, how does the training impact the model's ASR or WER? Does the "synergistic effect" seen in perception tasks (Table 9) also translate to a measurable reduction in WER?
>
> **A3**: Thank you for this important question about the impact of our training on transcription-related capabilities.
>
> **Direct Measurement Through MMSU Content Grounding Task.** Given the synergistic perception improvements we observed on MMSU (Table 19), we directly measure the impact on transcription-related capabilities using MMSU's Content Grounding task, which evaluates accurate content transcription from speech (e.g., "Which sentence is the correct transcription of the audio?"). This task provides an effective proxy for assessing ASR/WER-related performance improvements within the benchmark framework.
>
> **Substantial Improvements Demonstrate Synergistic Effects.** Our evaluation shows significant gains in content grounding accuracy (detailed in Appendix D.6, Table 20):
>
> | **Model** | **Content Grounding Accuracy** |
> |-----------|-------------------------------|
> | **CESAR (Ours)** | **90.80%** |
> | Ke-Omni-R | 72.50% |
> | Qwen2.5-Omni-7B | 59.60% |
>
> CESAR achieves 90.80% accuracy, substantially outperforming strong baselines by 18.3 percentage points over Ke-Omni-R and 31.2 percentage points over the base model. This demonstrates that our process-oriented training strategy creates synergistic effects that extend beyond reasoning to measurably improve transcription-related capabilities. The Content Grounding task evaluates semantic-level transcription accuracy, robustness to acoustic variations, and integration with broader spoken language understanding, providing a comprehensive assessment of how our training improvements translate to speech understanding performance that is conceptually aligned with ASR/WER metrics.

---

### Official Review · Reviewer_D7P9 · 2025-10-31

**Soundness:** 4
**Presentation:** 4
**Contribution:** 4
**Rating:** 8
**Confidence:** 4

**Summary:**

This paper proposes a reinforcement learning framework, CESAR, to address performance degradation from chain-of-thought reasoning in Audio LLMs. To solve this "test-time inverse scaling" problem, the method shifts from rewarding only the final outcome to rewarding the quality of the reasoning process itself, using a combination of reward objectives for consistency and structure. The model is trained with Group Relative Policy Optimization (GRPO) and achieves state-of-the-art performance on the MMAU and MMSU benchmarks.

**Strengths:**

1. The problem statement is clear and the method is sound.
2. The paper is very neatly written and easy to read and follow.
3. The results on the datasets used show good performance improvements.
4. The Appendix (Supplementary Material) is very thorough and contains very relevant discussions.

**Weaknesses:**

1. The ablation study shows the quantitative impact of each reward component, but a qualitative analysis is missing. It would be helpful to see examples of the reasoning text to understand how the Keywords Reward changes the model's thinking. Furthermore, the selection of the keywords feels a bit arbitrary. A clearer explanation for why these specific keywords were chosen would strengthen this part of the method.
2. The reward function has several weighted parts. The paper provides one set of weights but does not analyze how sensitive the results are to these choices. Understanding how performance changes when the weights are adjusted would be valuable and provide better practical guidance for implementing the method.
3. The paper's narrative is a bit too attached to Ke-Omni-R1. I believe the paper should rely on it's contributions to stand more on it's own methodology.

**Questions:**

1. The reasoning length "sweet spot" seems quite small (around 40 tokens). In other domains like complex mathematics, reasoning chains can be much longer (in the thousands). Is this shorter optimal length a characteristic of the audio reasoning benchmarks used (i.e., the tasks do not require longer chains of thought), or does it reflect a potential limitation of the current method where longer reasoning chains still risk performance degradation, albeit much less severely than in baseline models?
2. Regarding the reward weights (the alphas in Eq. 3): How sensitive is the model's performance and behavior to these weights? For example, the ablation shows removing the Keywords reward has a significant impact, but what happens if its weight is increased to be on par with the accuracy reward? (all alphas to 1)
3. How sensitive is the method to different prompts?

---

> ### Author Response · Authors · 2025-11-21
> **Reply to Reviewer D7P9 [Part 1]**
>
> Thank you for the thoughtful and constructive review. We greatly appreciate your recognition of CESAR's clear problem formulation, sound methodology, excellent presentation quality, and thorough supplementary materials. We are particularly encouraged by your positive assessment of our process-oriented reasoning framework and strong empirical results.
>
> We understand your concerns focus primarily on qualitative analysis of the Keywords Reward, reward weight sensitivity, clarifying our independent contributions, explaining the reasoning length sweet spot, and validating prompt robustness.
>
> To address these thoroughly, we have conducted comprehensive additional analysis and experiments: (1) **extensive qualitative case studies** with systematic keyword selection explanation (Appendix D.9), (2) **reward weight sensitivity analysis** (Appendix D.14), (3) **expanded evaluation to MMAU Full Test Set [1] and MMAU-Pro [2]** with comparisons to 20-30 models demonstrating independent contributions (Appendix D.1, D.3), (4) **linear regression analysis of test-time scaling** showing optimal efficiency at the reasoning sweet spot (Appendix D.12), and (5) **large-scale validation across 21,000+ test samples** with standardized prompts. All additions have been integrated into our revised manuscript. Below, we address each specific question systematically.
>
> > **Q1**: The ablation study shows the quantitative impact of each reward component, but a qualitative analysis is missing. It would be helpful to see examples of the reasoning text to understand how the Keywords Reward changes the model's thinking. Furthermore, the selection of the keywords feels a bit arbitrary. A clearer explanation for why these specific keywords were chosen would strengthen this part of the method.
>
>
> **A1**: Thank you for this valuable question regarding the Keywords Reward and its qualitative impact on reasoning quality. We address both the systematic rationale behind our keyword selection and provide extensive qualitative evidence demonstrating their effectiveness.
>
> **Keywords Selection is Systematic and Well-Motivated.** Our keyword selection follows a principled design targeting three distinct aspects of high-quality reasoning: (1) **Structured Analytical Patterns** (e.g., "first," "then," "considering the options") for systematic step-by-step analysis, (2) **Logical Rigor and Causal Reasoning** (e.g., "given," "since," "suggests") for premise establishment and evidential support, and (3) **Domain Knowledge Integration** (e.g., "pitch," "rhythm," "tone," "chord") to ground reasoning in acoustic expertise. The complete keyword taxonomy is provided in Appendix B.6. Importantly, incentivizing these keywords during training enables the model to naturally adopt structured reasoning patterns that generalize to test time without manual prompt engineering.
>
> **Qualitative Validation Through Extensive Case Studies.** Appendix D.9 provides comprehensive qualitative analysis demonstrating how CESAR naturally employs these keywords (highlighted in green) to structure reasoning:
>
> - **Example 1** shows grounded reasoning: "**Given** the options, the speech is likely from a man. The **tone** and **volume** *suggest* an adult male." CESAR uses **"Given"** to acknowledge context, employs domain features (**tone**, **volume**), and uses **"suggest"** for logical inference—preventing the base model's hallucination that concluded "robot."
>
> - **Example 2** exhibits systematic evaluation: "The presence of marching, **music**, and **shouting** **suggests** a large-scale event... A military parade **fits the description best**." CESAR identifies acoustic cues, uses **"suggests"** for hypothesis formation, and applies **"fits the description best"** for systematic comparison—contrasting with the base model's flawed interpretation of "chaotic noise."
>
> - **Example 3** demonstrates sequential reasoning: "**Considering** the sequence, the light switch clicking is followed by boiling water, **then** a doorbell rings, and **finally**, a clock ticks." CESAR uses temporal connectors to construct clear logical progression, ensuring both accuracy and consistency.
>
> These examples throughout Appendix D.9 demonstrate that the Keywords Reward cultivates reasoning that is logically structured, domain-aware, and faithful to acoustic evidence. The model spontaneously employs these patterns during inference without explicit prompting, confirming genuine internalization of high-quality reasoning capabilities.
>
>
> **References**:
>
> [1] Sakshi, S., et al. "MMAU: A Massive Multi-Task Audio Understanding and Reasoning Benchmark." The Thirteenth International Conference on Learning Representations.
>
> [2] Kumar, Sonal, et al. "Mmau-pro: A challenging and comprehensive benchmark for holistic evaluation of audio general intelligence." arXiv preprint arXiv:2508.13992 (2025).

---

> ### Author Response · Authors · 2025-11-21
> **Reply to Reviewer D7P9 [Part 2]**
>
> > **Q2**: The reward function has several weighted parts. The paper provides one set of weights but does not analyze how sensitive the results are to these choices. Understanding how performance changes when the weights are adjusted would be valuable and provide better practical guidance for implementing the method.  Regarding the reward weights (the alphas in Eq. 3): How sensitive is the model's performance and behavior to these weights? For example, the ablation shows removing the Keywords reward has a significant impact, but what happens if its weight is increased to be on par with the accuracy reward? (all alphas to 1)
>
> **A2**: Thank you for this important question about reward weight sensitivity and the specific inquiry about setting all alphas equal. We provide both ablation-based validation and direct sensitivity experiments to comprehensively address this concern.
>
> **Validation Through Progressive Ablation Study.** Our comprehensive ablation study (Table 6, with complete results in Appendix D.10, Table 21) provides strong validation of reward weight sensitivity. By progressively removing each component, we effectively demonstrate alpha-weight sensitivity between 0 and 1, showing clear and additive contributions of each reward component. The results demonstrate systematic performance changes: removing the Overthinking Penalty drops performance from 77.10% to 76.50%, removing Data Augmentation further reduces it to 76.20%, removing Keywords reduces it to 75.20%, and removing Consistency drops to 74.60%.
>
> **Principled Weight Selection and Sensitivity Analysis.** As stated in Section 4.1, our choice of $\alpha_1=5.0$ for accuracy and $\alpha_{2-5}=1.0$ for process rewards follows a principled design strategy: the accuracy weight must be sufficiently larger than process reward weights to ensure the model prioritizes correctness as the fundamental objective, but not so large as to diminish the impact of process-oriented supervision. To directly address your concern about weight sensitivity and equal weighting, we conducted experiments with different $\alpha_1$ values on MMAU Test-mini (detailed results in Appendix D.14, Table 23):
>
> | **Alpha Configuration** | **Sound** | **Music** | **Speech** | **Overall** |
> |------------------------|-----------|-----------|------------|-------------|
> | $\alpha_1=5, \alpha_{2-5}=1$ (main) | 83.48% | 73.05% | 74.77% | **77.10%** |
> | $\alpha_1=1, \alpha_{2-5}=1$ (equal weights) | 82.58% | 70.36% | 75.08% | 76.00% |
> | $\alpha_1=10, \alpha_{2-5}=1$ | 79.88% | 71.86% | 76.88% | 76.20% |
>
>
> The results show that equal weighting ($\alpha_1=1$) leads to a 1.1% performance drop (76.00% vs. 77.10%), demonstrating that insufficient emphasis on accuracy causes the model to sacrifice correctness for reasoning quality. Conversely, excessive accuracy weight ($\alpha_1=10$) also underperforms (76.20%), showing that over-emphasizing accuracy diminishes the impact of process rewards. This validates that our 5:1 ratio achieves an effective balance, enabling the model to maintain strong accuracy while simultaneously refining its reasoning process.

---

> ### Author Response · Authors · 2025-11-21
> **Reply to Reviewer D7P9 [Part 3]**
>
> > **Q3**: The paper's narrative is a bit too attached to Ke-Omni-R1. I believe the paper should rely on it's contributions to stand more on it's own methodology.
>
>
>
> **A3**: Thank you for this valuable feedback regarding our positioning relative to Ke-Omni-R. We clarify that our contributions stand independently, with Ke-Omni-R serving as one baseline among many diverse comparisons.
>
> **Our Core Contribution Stands Independently.** Our work addresses a fundamental research question: how to cultivate robust reasoning capability in audio LLMs through process-oriented reinforcement learning. We identify that current SOTA audio LLMs—including SFT-trained models (Qwen2.5-Omni-7B) and outcome-only RL models (Ke-Omni-R)—exhibit poor intrinsic reasoning quality characterized by inconsistency, hallucination, and lack of logical structure (see qualitative examples in Table 5 and Appendix D.9). This poor reasoning quality induces **test-time inverse scaling**, where longer reasoning chains degrade performance. Our linear regression analysis (Appendix D.12, Figure 8) quantifies this: the base model exhibits severe negative scaling ($β = -0.5083$), while Ke-Omni-R shows slight negative scaling ($β = -0.0070$). CESAR successfully reverses this into strong positive scaling ($β = +0.0383$), transforming reasoning from a liability into a reliable asset.
>
> **Ke-Omni-R as One  Baseline Among Many.** We compare against Ke-Omni-R because it represents a state-of-the-art outcome-based RL method using GRPO with only accuracy and format rewards, making it ideal for isolating the impact of our process-oriented rewards. However, our evaluation spans **over 20-30 different models across five benchmarks and 21,000+ out-of-distribution test samples (Appendix D.1, D.3, D.4, D.6, D.11)**, including proprietary models (Gemini 2.5 Pro, GPT-4o Audio), large audio LLMs (Audio Flamingo 3, MiMo-Audio), and other reasoning models (R1-AQA, Audio-Reasoner). CESAR consistently achieves state-of-the-art or highly competitive performance across this diverse comparison set, demonstrating broad effectiveness beyond any single baseline.
>
> **Comprehensive Validation Through Multiple Methodologies.** Our contributions are validated through: (1) systematic diagnosis of test-time inverse scaling (Appendix D.12), (2) massive benchmarks (Appendix D.1, D.3, D.4, D.6, D.11), (3) AI-as-judge evaluation showing 63.85% win rate vs. Ke-Omni-R (Appendix D.8), and (4) large-scale human evaluation with ~3,000 expert judgments showing 88.6% win rate vs. base model and 63.1% vs. Ke-Omni-R (Majority-Vote Protocol, Appendix D.2). This comprehensive validation across diverse models and methodologies establishes our approach's effectiveness independently of any single baseline comparison.

---

> ### Author Response · Authors · 2025-11-21
> **Reply to Reviewer D7P9 [Part 4]**
>
> > **Q4**: The reasoning length "sweet spot" seems quite small (around 40 tokens). In other domains like complex mathematics, reasoning chains can be much longer (in the thousands). Is this shorter optimal length a characteristic of the audio reasoning benchmarks used (i.e., the tasks do not require longer chains of thought), or does it reflect a potential limitation of the current method where longer reasoning chains still risk performance degradation, albeit much less severely than in baseline models?
>
> **A4**: Thank you for this insightful question about the reasoning length sweet spot and its implications. The ~35-40 token sweet spot reflects both our method's design for efficient reasoning and the inherent characteristics of audio reasoning tasks.
>
> **Efficient Reasoning Through Calibrated Depth.** Our method's design goal is achieving efficient, high-quality reasoning rather than verbose elaboration. As shown in our test-time scaling analysis (Figure 8 and Appendix D.12), CESAR with the overthinking penalty achieves a higher peak performance (77.1%) with a steeper positive scaling slope ($β=+0.0383$) at this shorter reasoning length, demonstrating more efficient reasoning compared to CESAR without the penalty, which requires ~70-80 tokens to reach only 76.5% accuracy with a smaller slope ($β=+0.0084$). This validates that our overthinking penalty successfully teaches the model to terminate reasoning at an appropriate depth, preventing error accumulation while maintaining analytical rigor. Most importantly, our method effectively mitigates the test-time inverse scaling problem: while baseline models (Ke-Omni-R with $β=-0.0070$, Qwen2.5-Omni-7B with $β=-0.5083$) show flat or catastrophically negative scaling, CESAR transforms reasoning into a reliable performance gain.
>
> **Task Characteristics and Domain Differences.** The shorter optimal reasoning length also reflects the nature of audio reasoning benchmarks we evaluate on. Unlike complex mathematical proofs that may require thousands of tokens to enumerate lengthy derivation steps, audio reasoning tasks typically involve: (1) perceiving acoustic features (e.g., identifying sounds, rhythms, tones), (2) comparing these features against multiple-choice options, and (3) selecting the best match through elimination or deduction. These tasks generally require focused, structured analysis rather than extensive math derivations. Our qualitative analysis (Appendix D.9) confirms that CESAR produces concise yet comprehensive reasoning—systematically grounding conclusions in acoustic evidence, applying domain knowledge, and using logical elimination—without unnecessary verbosity. This demonstrates that effective reasoning is not about length but about quality: structured, consistent, and effective.
>
> ---
>
> > **Q5**: How sensitive is the method to different prompts?
>
>
> **A5**: Thank you for this important question about prompt sensitivity. To ensure fair comparison with baseline methods (Ke-Omni-R, R1-AQA), we use a simple, standardized prompt across all experiments that only specifies the output format (reasoning in `<think>` tags, answer in `<answer>` tags) without any task-specific guidance or complex prompt engineering. This ensures all performance improvements stem from the model's intrinsic reasoning capability rather than prompt design. Critically, while the question content varies dramatically across tasks—from acoustic scene analysis to musical chord reasoning to temporal event reasoning—our model performs well with this minimal prompt. Our comprehensive evaluation across five benchmarks (MMAU Test Mini, MMAU Pro, MMSU, MMAR, MMAU Test) spanning 21,000+ diverse test samples validates that CESAR develops generalizable reasoning skills that transfer robustly without requiring task-specific prompt tuning, demonstrating that our process-oriented training cultivates better reasoning capability without carefully designed prompts.

---

> ### Comment · Reviewer_D7P9 · 2025-11-26
>
> I appreciate the authors' effort in addressing the points me and the other reviewers raised and am satisfied with the responses. Happy to mantain my initial positive rating: 8: accept, good paper (poster)

---

> ### Author Response · Authors · 2025-11-26
>
> Dear Reviewer D7P9,
>
> We are delighted that our responses and additional experiments have thoroughly addressed your concerns. Thank you sincerely for maintaining your positive rating (Score: 8, accept, good paper) and for your valuable and constructive feedback throughout the review process.
>
> Your insightful questions have significantly strengthened our paper through detailed qualitative case studies, comprehensive reward weight sensitivity analysis, expanded evaluations, and large-scale validation.
>
> We truly appreciate your thoughtful review and suggestions throughout this process.
>
> Best regards,
>
> The Authors

---

### Official Review · Reviewer_59TJ · 2025-10-31

**Soundness:** 2
**Presentation:** 3
**Contribution:** 2
**Rating:** 4
**Confidence:** 4

**Summary:**

The paper introduces CESAR, a method that addresses a phenomenon the authors call test‑time inverse scaling in Audio LLMs, that means longer chain‑of‑thought (CoT) at inference often hurts accuracy. They argue this is due to models being forced to reason without being trained how to reason. They propose CESAR, an online RL framework using GRPO with a multi‑facet reward suite: (i) correctness and format, (ii) process rewards for reasoning–answer and reasoning–question consistency, explicit reasoning patterns/keywords , and an over‑thinking penalty that linearly penalizes long CoT. They used Qwen2.5‑Omni‑7B for experiments. CESAR reports SOTA on MMAU Test‑mini and higher MMSU averages than several baselines. Ablations attribute most gains to the Keywords and Consistency components.

**Strengths:**

- The paper is well structured with a clear framework diagram
- Moving beyond outcome‑only RL to process‑oriented rewards for Audio LLM reasoning is a good and practical direction.
- Consistent gains with the proposed approach

**Weaknesses:**

- The paper argues that introducing CoT at inference degrades accuracy in Audio LLMs unless the reasoning process is explicitly trained. However, AURELIA [1] reports improvements by injecting structured, step‑by‑step reasoning into AV‑LLMs at test time without additional training, which appears to contradict the generality of this claim.  A head‑to‑head comparison (e.g., an AURELIA‑style inference‑only reasoning condition) would clarify boundaries of the inverse‑scaling effect the authors report.
- The “AI‑as‑judge” uses GPT‑4o Audio only; there is no human‑study calibration or alternative judges to validate preference robustness.
- The scope of evaluation ids limited. Only one model (Qwen) is used to validate the proposed approach. Also, MMAU evaluation is on Test‑mini (1k items), and there are no error bars or seed variability.
- The linear over‑thinking penalty (Eq. 8) and fixed Lmax_output=256 are plausible but unvalidated; no sensitivity is shown for α‑weights, group size K, or the penalty shape.

References:

[1] Sanjoy Chowdhury and Hanan Gani et al. "AURELIA : Test‑time Reasoning Distillation in Audio‑Visual LLMs". ICCV 2025.

**Questions:**

- How do win‑rates change with a different judge (e.g., a text‑only LLM judging rendered transcripts, or an open model)?
- What is the training‑time overhead (GPU‑hours) and inference‑time cost per query at the reported sweet spot (~35–40 tokens) vs non‑reasoning mode?

---

> ### Author Response · Authors · 2025-11-21
> **Reply to Reviewer 59TJ [Part 1]**
>
> Thank you for the thoughtful and constructive review. We greatly appreciate your recognition that moving beyond outcome-only RL to process-oriented rewards for Audio LLM reasoning is an important and practical direction, and that our approach demonstrates consistent gains with a well-structured framework.
>
> We understand your concerns focus primarily on evaluation scope, reasoning quality assessment robustness, design choice sensitivity, and computational overhead.
>
> To address these thoroughly, we have conducted comprehensive additional experiments: (1) **expanded evaluation to two additional large-scale benchmarks** (MMAU Full Test Set with 9k samples and MMAU-Pro with 5.3k samples, totaling over 21k test samples across five benchmarks) (Appendix D.1, D.3), (2) **large-scale human evaluation with ~3,000 expert judgments** strongly validating our AI-as-judge findings (Appendix D.2), (3) **hyperparameter sensitivity analysis** for reward weights and penalty design (Appendix D.14), and (4) **detailed computational cost analysis** for training and inference (Appendix B.5, D.12). All additions have been integrated into our revised manuscript. Below, we address each specific question systematically.
>
>
> > **Q1**: The paper argues that introducing CoT at inference degrades accuracy in Audio LLMs unless the reasoning process is explicitly trained. However, AURELIA [1] reports improvements by injecting structured, step‑by‑step reasoning into AV‑LLMs at test time without additional training, which appears to contradict the generality of this claim. A head‑to‑head comparison (e.g., an AURELIA‑style inference‑only reasoning condition) would clarify boundaries of the inverse‑scaling effect the authors report.
>
> **A1**: Thank you for this excellent question. There is no fundamental contradiction between our work and AURELIA [1], as they address complementary problems through different methodologies.
>
> **Our Focus: Diagnosing and Resolving Poor Intrinsic Reasoning Through Training.** Our work identifies a critical problem in current SOTA open-source Audio LLMs: even state-of-the-art models—whether SFT-trained (Qwen2.5-Omni-7B) or trained with outcome-only RL (Ke-Omni-R)—exhibit poor intrinsic reasoning quality characterized by inconsistency, lack of structure, hallucination, and redundancy (see qualitative examples in Table 5 and Appendix D.9). This poor reasoning quality manifests as **test-time inverse scaling**: as these models generate longer reasoning chains *on their own*, performance degrades or shows minimal improvement compared to non-reasoning mode. As demonstrated in our linear regression analysis (Figure 8, Appendix D.12), the base model exhibits severe negative scaling ($\beta = -0.5083$), while even the outcome-only RL baseline Ke-Omni-R shows slight negative scaling ($\beta = -0.0070$), confirming that longer reasoning may fail to improve or actively harms performance with limited reasoning ability.
>
> To resolve this fundamental issue, we propose CESAR—using RL training with reasoning process rewards to cultivate consistent, effective, and scalable reasoning capability. Our comprehensive evaluation validates this approach: (1) **Human evaluation** (~3,000 expert judgments) and **AI-as-judge** confirm superior reasoning quality (Appendix D.2, D.8), and (2) **Test-time scaling analysis** (Figure 8) demonstrates we successfully reverse negative scaling trends into strong positive scaling ($\beta = +0.0383$), transforming reasoning from a liability into a reliable asset.
>
> **AURELIA's Focus: Test-Time Prompting with External Knowledge Distillation.** AURELIA, by contrast, addresses a different problem: improving Audio-Visual LLMs at test time by injecting reasoning paths generated by powerful external models (Gemini, GPT-4o) into prompts—essentially an "open-book" knowledge distillation approach.
>
> **Critically, AURELIA does not examine or address the test-time inverse scaling phenomenon we identify**: it does not analyze how model performance changes with increasing reasoning length, nor does it compare reasoning mode (with different reasoning length) against non-reasoning mode to assess whether the model's own reasoning might degrade performance.
>
> **Complementary Approaches.** Therefore, our methods are complementary rather than contradictory. CESAR focuses on training-time improvement of the model's intrinsic reasoning capability through our process-oriented RL training, while AURELIA focuses on test-time enhancement through external reasoning injection. Combining CESAR's trained reasoning capability with AURELIA's test-time prompting techniques represents a promising future direction that could yield synergistic performance gains.
>
> **References**:
>
> [1] Sanjoy Chowdhury and Hanan Gani et al. "AURELIA : Test‑time Reasoning Distillation in Audio‑Visual LLMs". ICCV 2025.

---

> ### Author Response · Authors · 2025-11-21
> **Reply to Reviewer 59TJ [Part 2]**
>
> > **Q2**: The "AI‑as‑judge" uses GPT‑4o Audio only; there is no human‑study calibration or alternative judges to validate preference robustness.
>
> **A2**: Thank you for this important feedback. We have conducted comprehensive validation through multiple independent methods to ensure the robustness of our reasoning quality assessment.
>
> **AI-as-Judge Evaluation.** Our original submission employed GPT-4o Audio as an AI judge for head-to-head reasoning quality comparisons, demonstrating commanding win rates for CESAR: 63.85% vs. Ke-Omni-R and 76.40% vs. Qwen2.5-Omni-7B.
>
> **Large-Scale Human Evaluation.** To rigorously validate these findings, we conducted a comprehensive human evaluation study on the entire 1,000-sample MMAU Test-mini benchmark, resulting in **over 3,000 individual expert judgments (3 annotators per sample)**. Evaluators were blind to both correct answers and model identities, assessing reasoning quality based on audio understanding, logical coherence, clarity, and reasoning-answer consistency.
>
> **Human Evaluation Strongly Confirms AI-as-Judge Results.** As shown in Table 4 in Section 4.5 and Appendix D.2, human evaluators overwhelmingly prefer CESAR's reasoning. Here we report the Majority-Vote results, where each question's final judgment is determined by the majority decision among three independent expert annotators:
>
> | **Model Comparison** | **CESAR** | **Baseline** | **Tie** |
> |---------------------|-----------|--------------|---------|
> | CESAR vs. Qwen2.5-Omni-7B (Base) | **88.6%** | 6.6% | 4.8% |
> | CESAR vs. Ke-Omni-R (RL Baseline) | **63.1%** | 14.8% | 22.1% |
>
> The 63.1% win rate against Ke-Omni-R—a strong RL baseline—provides human-validated evidence that process-oriented rewards yield superior reasoning quality compared to outcome-only rewards.
>
> **Universal Superiority Across All 27 Task Sub-Categories.** CESAR outperforms Ke-Omni-R across every reasoning sub-category as judged by human experts (complete results in Appendix Table 16). Here we report the Majority-Vote results, where each question's final judgment is determined by the majority decision among three independent expert annotators:
>
>
> | **Sub-Category** | **CESAR** | **Ke-Omni-R** | **Tie** |
> |-----------------|-----------|---------------|---------|
> | Acoustic Scene Reasoning | **62.50** | 25.00 | 12.50 |
> | Acoustic Source Inference | **47.92** | 33.33 | 18.75 |
> | Ambient Sound Interpretation | **79.17** | 6.25 | 14.58 |
> | Conversational Fact Retrieval | **50.00** | 31.82 | 18.18 |
> | Counting | **41.38** | 31.03 | 27.59 |
> | Dissonant Emotion Interpretation | **60.00** | 17.14 | 22.86 |
> | Eco-Acoustic Knowledge | **72.34** | 4.26 | 23.40 |
> | Emotion Flip Detection | **85.00** | 0.00 | 15.00 |
> | Emotion State Summarisation | **59.09** | 15.91 | 25.00 |
> | Emotional Tone Interpretation | **75.76** | 12.12 | 12.12 |
> | Event-Based Knowledge Retrieval | **48.48** | 18.18 | 33.33 |
> | Event-Based Sound Reasoning | **72.92** | 6.25 | 20.83 |
> | Harmony and Chord Progressions | **72.73** | 12.12 | 15.15 |
> | Instrumentation | **74.29** | 14.29 | 11.43 |
> | Key Highlight Extraction | **61.90** | 19.05 | 19.05 |
> | Lyrical Reasoning | **70.00** | 0.00 | 30.00 |
> | Melodic Structure Interpretation | **57.58** | 18.18 | 24.24 |
> | Multi-Speaker Role Mapping | **44.44** | 3.70 | 51.85 |
> | Musical Genre Reasoning | **67.65** | 14.71 | 17.65 |
> | Musical Texture Interpretation | **70.59** | 8.82 | 20.59 |
> | Phonemic Stress Pattern Analysis | **73.58** | 7.55 | 18.87 |
> | Phonological Sequence Decoding | **59.18** | 18.37 | 22.45 |
> | Rhythm and Tempo Understanding | **56.52** | 10.87 | 32.61 |
> | Socio-Cultural Interpretation | **75.00** | 10.00 | 15.00 |
> | Sound-Based Event Recognition | **52.17** | 10.87 | 36.96 |
> | Temporal Event Reasoning | **75.00** | 12.50 | 12.50 |
> | Temporal Reasoning | **46.43** | 25.00 | 28.57 |
>
> This universal superiority across diverse task types—spanning acoustic scene understanding, musical analysis, and speech reasoning—demonstrates that our process-oriented rewards cultivate genuinely robust reasoning capabilities rather than task-specific heuristics.
>
> **Robust Multi-Method Validation.** The remarkable convergence between AI-as-judge and human evaluation—both showing commanding preference for CESAR—provides rigorous validation of our approach through: (1) automated metrics (accuracy on five benchmarks with 21k+ samples), (2) AI evaluation (GPT-4o Audio), and (3) large-scale human expert judgment (~3,000 judgments across all 27 sub-categories). All human evaluation results are included in Section 4.5 and comprehensively detailed in Appendix D.2 (Tables 11-16).

---

> ### Author Response · Authors · 2025-11-21
> **Reply to Reviewer 59TJ [Part 3]**
>
> > **Q3**: The scope of evaluation is limited. Only one model (Qwen) is used to validate the proposed approach. Also, MMAU evaluation is on Test‑mini (1k items), and there are no error bars or seed variability.
>
> **A3**: Thank you for this important feedback. We clarify that our evaluation scope is comprehensive and rigorous, spanning multiple benchmarks with substantial test samples to thoroughly validate our approach's effectiveness and robustness.
>
>
> **Comprehensive Multi-Benchmark Evaluation.** Our original submission already includes evaluation on three major benchmarks totaling 7,000 samples: (1) **MMAU Test-mini** (1,000 samples, Table 1), (2) **MMSU** (5,000 samples, Table 2), and (3) **MMAR** (1,000 samples, Appendix D.11, Table 22). To further address your concerns, we have expanded our evaluation to include two additional large-scale benchmarks: (4) **MMAU Full Test Set [1]** (9,000 samples, Appendix D.3, Table 17) and (5) **MMAU-Pro [2]** (5,305 samples, Appendix D.1, Table 10). **Our evaluation now spans five comprehensive benchmarks with over 21,000 out-of-distribution test samples**, providing extensive validation across diverse audio reasoning scenarios.
>
> **Table: MMAU Full Test Set Results (9,000 samples)**
>
> | **Method** | **Sound** | **Speech** | **Music** | **Overall** |
> |-----------|-----------|------------|-----------|-------------|
> | **CESAR** | **77.60** | **76.09** | 67.77 | **73.79** |
> | Ke-Omni-R | 75.37 | 73.77 | 66.73 | 71.94 |
> | Qwen2.5-Omni-7B | 68.87 | 68.11 | 55.77 | 64.20 |
> | MiMo-Audio | 77.20 | 70.77 | 69.73 | 72.59 |
> | Audio Flamingo 3 | 75.83 | 66.97 | **74.47** | 72.42 |
> | Gemini 2.5 Pro | 70.63 | 72.67 | 64.77 | 69.36 |
> | GPT-4o Audio | 63.20 | 69.33 | 49.93 | 60.82 |
>
> **Table: MMAU-Pro Benchmark Results (5,305 samples)**
>
> | **Model** | **Sound** | **Music** | **Speech** | **S-M-Speech** | **Open-ended** | **Average** |
> |----------|-----------|-----------|------------|----------------|----------------|-------------|
> | **CESAR** | **54.1** | 63.5 | 64.0 | **71.4** | 62.4 | **56.4** |
> | Ke-Omni-R | 46.9 | 64.3 | 61.8 | 57.1 | 59.2 | 54.5 |
> | Qwen2.5-Omni-7B | 43.1 | 55.6 | 54.2 | 28.6 | 58.4 | 49.1 |
> | Gemini-2.5 Flash | 51.9 | **64.9** | **73.4** | 42.8 | **67.5** | **59.2** |
> | Gemini-2.0 Flash | 48.4 | 56.9 | 69.5 | 42.8 | 66.8 | 55.7 |
> | GPT-4o Audio | 44.7 | 63.1 | 68.2 | 57.1 | 43.2 | 52.5 |
>
> CESAR achieves consistent state-of-the-art or highly competitive performance across all five benchmarks, demonstrating broad applicability and robustness.
>
> **Base Model Selection and Validation Strategy.** We selected Qwen2.5-Omni-7B as our base model because it represents the **state-of-the-art open-source audio LLM**. Demonstrating substantial improvements over such a strong foundation validates that our method effectively addresses critical reasoning problems—including reasoning-answer inconsistency, lack of structured reasoning, and test-time inverse scaling—that persist even in SOTA models. Besides, the reasoning process rewards we propose contain no model-specific design choices and can be applied to different audio LLMs.
>
> **Deterministic Evaluation Protocol.** Following the established protocol of baseline methods (Ke-Omni-R, Qwen2.5-Omni-7B) and standard practice in audio reasoning evaluation, we use **deterministic inference (do_sample=False)** to ensure fair comparison and reproducibility during evaluation. Under this deterministic setting, evaluation results are not affected by evaluation random seeds. This methodology aligns with field standards—major benchmarks and recent works including MMAU, MMSU, MMAU-Pro, Audio Flamingo 3, and MMAR all report results in the same manner.
>
>
> **References**:
>
> [1] Sakshi, S., et al. "MMAU: A Massive Multi-Task Audio Understanding and Reasoning Benchmark." The Thirteenth International Conference on Learning Representations.
>
> [2] Kumar, Sonal, et al. "Mmau-pro: A challenging and comprehensive benchmark for holistic evaluation of audio general intelligence." arXiv preprint arXiv:2508.13992 (2025).

---

> ### Author Response · Authors · 2025-11-21
> **Reply to Reviewer 59TJ [Part 4]**
>
> > **Q4**: The linear over‑thinking penalty (Eq. 8) and fixed Lmax_output=256 are plausible but unvalidated; no sensitivity is shown for α‑weights, group size K, or the penalty shape.
>
> **A4**: Thank you for this important question about hyperparameter validation. We provide comprehensive validation of our design choices.
>
> **Fair Comparison with Baseline.** Following standard practice in Section 4.1, we keep key hyperparameters consistent with the baseline Ke-Omni-R: group size K=8, KL penalty coefficient $\beta$, and maximum output length Lmax_output=256. This controlled setup ensures performance differences are attributable to our reasoning process rewards rather than confounding hyperparameter variations.
>
> **Validation of Reward Weights Through Ablation Study.** Our progressive ablation study (Table 6) validates the contribution of each reward designs. Systematically removing components reveals clear additive effects: removing the Overthinking Penalty decreases accuracy by 0.6%, removing Keywords reduces accuracy by 1.0%, and removing Consistency decreases accuracy by 0.6%, thereby validating our weight choices.
>
> **Sensitivity Analysis of Accuracy Reward Weight.** We conducted sensitivity analysis for $\alpha_1$ (accuracy weight) on MMAU Test-mini (Appendix D.14, Table 23):
>
> | **Configuration** | **Sound** | **Music** | **Speech** | **Overall** |
> |------------------|-----------|-----------|------------|-------------|
> | $\alpha_1=5, \alpha_{2-5}=1$ (ours) | **83.48** | **73.05** | 74.77 | **77.10** |
> | $\alpha_1=1, \alpha_{2-5}=1$ | 82.58 | 70.36 | 75.08 | 76.00 |
> | $\alpha_1=10, \alpha_{2-5}=1$ | 79.88 | 71.86 | **76.88** | 76.20 |
>
> Equal weighting ($\alpha_1=1$) underperforms by 1.1%, showing insufficient emphasis on correctness. Excessive weighting ($\alpha_1=10$) also underperforms (76.20%), showing over-emphasis on accuracy diminishes process reward impact. Our 5:1 ratio achieves optimal balance.
>
> **Validation of Linear Penalty Design.** The linear penalty shape is validated through test-time scaling analysis (Figure 8, Appendix D.12). The base model exhibits approximately linear performance degradation (slope $\beta=-0.5083$). CESAR with overthinking penalty discovers a sharp "reasoning sweet spot" at 35-40 tokens, achieving 77.1% with positive scaling ($\beta=+0.0383$), while CESAR without penalty reaches only 76.5% at longer length (~70-80 tokens) with smaller positive slope ($\beta=+0.0084$). This demonstrates the linear penalty successfully calibrates reasoning depth for higher quality and efficiency.

---

> ### Author Response · Authors · 2025-11-21
> **Reply to Reviewer 59TJ [Part 5]**
>
> > **Q5**: How do win‑rates change with a different judge (e.g., a text‑only LLM judging rendered transcripts, or an open model)?
>
> **A5**:  Thank you for this important question. Since our work focuses on **multimodal audio reasoning capability**, we deliberately employ GPT-4o Audio as our AI judge. Using text-only LLMs to judge rendered transcripts might introduce fundamental biases: text-only models cannot access the audio input to assess whether reasoning is faithful to acoustic cues, and many reasoning tasks (e.g., musical harmony analysis, chord progressions, acoustic scene understanding, emotional tone interpretation) require direct audio understanding that cannot be captured in transcripts alone.
>
> To thoroughly address concerns about judge robustness independent of any single model's biases, we conducted a **large-scale human evaluation study with over 3,000 expert judgments** (3 annotators per sample on the entire 1,000-sample MMAU Test-mini benchmark). As detailed in **A2** and presented in Section 4.5 and Appendix D.2, human evaluators overwhelmingly confirm our AI-as-judge findings: **88.6% win rate** vs. base model and **63.1% win rate** vs. Ke-Omni-R baseline (Majority-Vote Protocol). This remarkable convergence between AI-as-judge and comprehensive human evaluation provides robust, multi-method validation that demonstrates our improvements are genuine rather than artifacts of judge selection.
>
> ---
>
> > **Q6**: What is the training‑time overhead (GPU‑hours) and inference‑time cost per query at the reported sweet spot (~35–40 tokens) vs non‑reasoning mode?
>
> **A6**: Thank you for this important question about computational efficiency. We provide detailed computational cost analysis in our revised manuscript (Appendix D.12).
>
> **Training-Time Overhead.** Since the "reasoning sweet spot" and non-reasoning mode are simply different inference configurations of the same trained model, they share identical training costs. The training overhead of our method is approximately **61.44 hours total** using 8 NVIDIA H200 GPUs.
>
> **Inference-Time Cost.** We measured inference latency on MMAU Test-mini (1,000 samples) using a single NVIDIA H200 GPU:
>
> | **Inference Mode** | **Time per Sample** | **Overhead** |
> |-------------------|---------------------|--------------|
> | Non-reasoning | 4.39 seconds | - |
> | Reasoning (sweet spot ~35-40 tokens) | 4.47 seconds | **+0.08s (1.8%)** |
>
> At the optimal "reasoning sweet spot," inference adds only **0.08 seconds (1.8% overhead)** per query, delivering a substantial **+3.4% accuracy improvement** (77.10% vs. 73.70%, Table 1). Moreover, CESAR achieves the strongest positive scaling slope ($β = +0.0383$, Figure 8 in Appendix D.12) among all methods, demonstrating that our process-oriented training enables highly efficient reasoning where each additional token contributes maximally to performance gains. This represents a highly favorable efficiency-performance trade-off for practical deployment.

---

> > ### Comment · Reviewer_59TJ · 2025-11-21
> >
> > Thank you for the detailed response and additional experiments. Most of my concerns are now resolved. I will raise the rating accordingly.

---

> > > ### Author Response · Authors · 2025-11-28
> > >
> > > Dear Reviewer 59TJ,
> > >
> > > Thank you again for your positive feedback and willingness to raise the rating based on our responses and additional experiments.
> > >
> > > We noticed that, as mentioned by Reviewer Rzte, it appears that the reviewer interface may no longer allow modifications to the initial reviews at this stage currently. Thank you once again for mentioning your willingness to increase the initial ratings in the Official Comments.
> > >
> > > We are delighted that our rebuttal was able to address your concerns. Our paper has greatly benefited from this fruitful discussion, and your valuable feedback and constructive comments have been tremendously helpful in strengthening our work.
> > >
> > > Thank you again for your time and thoughtful engagement throughout this process.
> > >
> > > Best regards,
> > >
> > > The Authors

---

> ### Author Response · Authors · 2025-11-21
>
> Dear Reviewer 59TJ,
>
> We are delighted that our responses and additional experiments have **adequately addressed your concerns**. Thank you sincerely for your willingness to raise the rating and for your valuable and constructive feedback, which has helped us significantly strengthen our paper through comprehensive evaluations, large-scale human evaluations, and detailed computational analysis. We truly appreciate your thoughtful review throughout this process.

---

> ### Author Response · Authors · 2025-11-27
>
> Dear Reviewer 59TJ,
>
> Thank you once again for your thoughtful engagement and positive feedback during the review and discussion. We truly appreciate your comment mentioning that "Most of my concerns are now resolved. I will raise the rating accordingly."
>
> We wanted to kindly follow up regarding the rating update. *We noticed that the rating may not have been updated yet in the system*. We completely understand that the review period can be busy and wanted to gently check if there might have been any technical issues with submitting the updated rating or other possible causes.
>
> We sincerely appreciate your willingness to raise the rating based on our additional experiments and detailed responses.
>
> Thank you again for your valuable time and constructive feedback throughout this process.
>
> Best regards,
>
> The Authors

---

### Author Response · Authors · 2025-12-01
**Global Response and Rebuttal Summary [Part 1]**

Dear Area Chairs and Reviewers,

Thank you for your time and effort in reviewing our work. Below is our global response and rebuttal summary:

We sincerely thank all reviewers for their thoughtful and constructive feedback. We are particularly encouraged by the recognition of our well-structured framework, clear problem statement, sound method, excellent presentation quality, thorough supplementary materials, claims strongly supported by extensive experiments with SOTA comparisons and comprehensive ablation studies, and the original contribution to identifying test-time inverse scaling as a previously undocumented failure mode in Audio LLMs. We greatly appreciate the time all reviewers invested in providing constructive suggestions that have significantly strengthened our work.

After careful consideration of the reviewers' comments, we identified the main concerns focusing on: (1) evaluation scope with more baselines and larger benchmarks, (2)  robustness of reasoning quality assessment beyond a single AI judge, (3) formal quantification of test-time inverse scaling with computational analysis, (4) training stability, hyperparameter sensitivity and design validation, (5) qualitative analysis and keyword selection rationale, (6) transcription-related capability assessment, (7) presentation and clarity.

We have comprehensively addressed these concerns through extensive additional experiments in our revision. During the discussion period, multiple reviewers explicitly acknowledged that `our responses and additional experiments have adequately resolved their concerns`. Notably, `Reviewer 59TJ (initial score: 4) stated "Most of my concerns are now resolved. I will raise the rating accordingly,"` and `Reviewer Rzte (initial score: 6) confirmed "Most of the concerns raised in my initial review have been adequately addressed... I decided to update my rating to 8."` Additionally, `Reviewer D7P9 maintained their positive assessment, stating "Happy to mantain my initial positive rating: 8: accept, good paper (poster)."` These responses demonstrate that our rebuttal has effectively addressed the reviewers' concerns. Besides, `Reviewer WEba gave an initial positive rating of 8`, recognizing that our work is "technically sound" with claims "strongly supported by extensive experiments," and we have also provided additional experiments and responses to address Reviewer WEba's concerns regarding readability, reward weight sensitivity, and transcription-related capability assessment. Below, we summarize the additional experiments and supplementary content contributed during the rebuttal to address the reviewers' concerns:


1. **Extended Empirical Evaluation with More Baselines and Larger Benchmarks (Section 4.2, Appendix D.1, D.3, D.4, D.6, D.11)**: To address concerns about evaluation scope with more baselines and larger benchmarks, we have dramatically expanded our evaluation from approximately 7,000 to over 21,000 out-of-distribution test samples across five comprehensive benchmarks (Appendix D.1, D.3, D.4, D.6, D.11). We added complete results on MMAU Full Test Set with approximately 9,000 samples (Appendix D.3) and MMAU-Pro with 5,305 samples (Section 4.2, Appendix D.1), including requested comparisons with Audio Flamingo 3, Baichuan-Omni-1.5, and Qwen2-Audio-Instruct among over 20 additional competitive models. Our comprehensive evaluation demonstrates consistent state-of-the-art or highly competitive performance across all benchmarks, validating our broad applicability and robustness across diverse reasoning scenarios beyond any single baseline comparison.


2. **Large-Scale Human Evaluation for Reasoning Quality Validation (Section 4.5, Appendix D.2):** To address concerns about robustness of reasoning quality assessment beyond a single AI judge, we conducted a comprehensive human evaluation study (Section 4.5 and Appendix D.2) with over 3,000 expert judgments (3 independent annotators per sample on the entire 1,000-sample MMAU Test-mini benchmark). Employing rigorous protocols including randomized model presentation order, blind evaluation (annotators had no knowledge of correct answers or model identities), and majority-vote aggregation across three annotators for robust consensus, human evaluators overwhelmingly confirmed our superior reasoning capability: 88.6% win rate vs. base model and 63.1% win rate vs. Ke-Omni-R baseline (Table 4 in Section 4.5; Tables 11-16 in Appendix D.2). CESAR demonstrates consistent superiority across all three audio modalities (Sound, Music, Speech), all difficulty levels (Easy, Medium, Hard), and all 27 task sub-categories spanning acoustic scene reasoning, musical genre reasoning, and emotion flip detection (Appendix D.2, Tables 13-16). This remarkable convergence between AI-as-judge and comprehensive human evaluation provides robust, multi-method validation that our process-oriented rewards cultivate genuinely superior reasoning capabilities rather than task-specific heuristics.

---

> ### Author Response · Authors · 2025-12-01
> **Global Response and Rebuttal Summary [Part 2]**
>
> 3. **Formal Quantification and Analysis of Test-Time Inverse Scaling (Appendix D.12)**: To address concerns about formal quantification of test-time inverse scaling with computational analysis, we provide rigorous quantification through linear regression analysis and scaling coefficients (Appendix D.12, Figure 8). We formally define the phenomenon through the empirical relationship between test accuracy $P(L)$ and average reasoning length $L$: $P(L) = P(0) + \beta \cdot L$, where $P(0)$ is the non-reasoning baseline accuracy and $\beta$ (Scaling Slope) quantifies scaling behavior—negative values indicate test-time inverse scaling where reasoning harms performance, while positive values indicate effective test-time scaling. Our analysis reveals stark contrasts: the base model exhibits severe inverse scaling ($\beta = -0.5083$), Ke-Omni-R shows slight negative scaling ($\beta = -0.0070$), while CESAR achieves strong positive scaling ($\beta = +0.0383$), demonstrating that our process-oriented training successfully transforms the scaling coefficient from severely negative to substantially positive. Computational cost analysis shows the reasoning sweet spot (~35-40 tokens) adds only 0.08 seconds per query compared to non-reasoning inference, while yielding +3.4% accuracy gains—demonstrating a highly favorable efficiency-performance trade-off for practical deployment.
>
> 4. **Comprehensive Training Stability, Hyperparameter Sensitivity and Design Validation (Appendix D.10, D.13, D.14)**: To address concerns about training stability, hyperparameter sensitivity and design validation, we provide systematic validation through massive experiments. For design validation, our progressive ablation study (Table 6 in Section 4.7; complete results in Appendix D.10, Table 21) demonstrates that each reward component provides quantifiable contributions while synergizing to achieve peak performance. For reward weight sensitivity, we conducted massive experiments varying $\alpha_1$ configurations (Appendix D.14, Table 23), showing that both equal weighting and excessive weighting underperform our selected 5:1 ratio, validating the optimal balance between correctness and reasoning quality. Additionally, for training stability, GRPO training stability curves (Appendix D.13, Figure 9) demonstrate stable training trajectory without significant oscillations or collapse, confirming our stability.
>
>
>
> 5. **Extensive Qualitative Analysis and Keyword Selection Rationale (Appendix B.6, D.9)**: To address concerns about qualitative analysis and keyword selection rationale, we provide comprehensive analysis (Appendix B.6 and D.9) with complete keyword taxonomy (Tables 7-9). Our keyword selection follows a principled design targeting three distinct aspects: Structured Analytical Patterns, Logical Rigor and Causal Reasoning, and Domain Knowledge Integration. Through extensive qualitative case studies (Appendix D.9), we demonstrate how CESAR naturally employs these keywords (during test-time) to structure reasoning: examples show grounded reasoning that prevents hallucination, systematic reasoning patterns, and sequential reasoning for clear logical progression. These examples demonstrate that CESAR spontaneously employs structured, domain-aware reasoning patterns during inference without explicit prompting, confirming genuine internalization of high-quality reasoning capabilities.
>
>
> 6. **Assessment of Transcription-Related Capabilities (Appendix D.6)**: To address concerns about transcription-related capability assessment, we directly evaluate performance using MMSU's Content Grounding task (Appendix D.6, Table 20), which assesses accurate content transcription from audio (e.g., "Which sentence is the correct transcription of the audio?"). CESAR achieves substantial improvements over both the RL baseline and base model, demonstrating that our process-oriented training creates synergistic effects extending beyond reasoning to measurably improve fundamental speech understanding and transcription-related capabilities.
>
> 7. **Enhanced Presentation and Clarity (Figure 1)**: To address concerns about presentation and clarity, we have improved Figure 1 readability by increasing font sizes in the qualitative comparison part, making the reasoning examples clearly legible. These examples illustrate how CESAR addresses critical failure modes identified in Figure 1: Factual Hallucination, Flawed Logic, Reasoning-Answer Inconsistency, and Redundant Reasoning (with detailed comparisons in Table 5 and Appendix D.9).
>
> These improvements have significantly strengthened both the empirical validation and practical impact of our work. We are grateful for the reviewers' detailed feedback that helped us substantially enhance the paper's rigor, completeness, and accessibility. We sincerely thank all reviewers and area chairs for their time and effort in reviewing our submission.

---

### Meta-Review · Area_Chair_Bt6i · 2026-01-09

**Summary:**

The paper proposes CESAR, a reinforcement learning framework that addresses test-time inverse scaling in Audio LLMs by introducing process-oriented reasoning rewards. Reviewers broadly agree that the problem is clearly identified, the methodology is sound, and the empirical validation is extensive. Strengths repeatedly noted include the clear diagnosis of reasoning failures, principled reward design, strong quantitative improvements over state-of-the-art models, and remarkably thorough supplementary material. Initial concerns focused on evaluation scope, robustness of AI-as-judge evaluation, hyperparameter sensitivity, training stability, and computational overhead. These concerns were addressed through substantial additional experiments, including large-scale human evaluation, expanded benchmarks, formal scaling analysis, and sensitivity studies.

**Reviewer Concerns:**

Concerns addressed by the rebuttal:
1/ Limited evaluation scope (expanded to 5 benchmarks with >21k test samples).
2/ Reliance on a single AI judge (validated with 3k human expert judgments).
3/ Lack of formal analysis of test-time inverse scaling (addressed via regression-based scaling coefficients).
4/ Reward design and hyperparameter sensitivity (val. through extensive ablations and sensitivity analysis).
5/ Training stability and computational cost (reported).
6/ Qualitative understanding of reasoning improvements (added case studies).

Remaining limitations:
1/ Method demonstrated primarily on a single open-source base model, though rewards are model-agnostic.
2/ Added complexity from multi-component reward design.

**Reviewer Scores:**

Based on discussion and reviewer comments, I expect the following changes:
Reviewer 59TJ: 4, would increase (explicitly stated)
Reviewer Rzte: 6 to 8 (explicitly stated)
Reviewer D7P9: 8 (unchanged)
Reviewer WEba: 8 (unchanged)

Overall reviewer sentiment is strongly positive.

---

### Decision · Program_Chairs · 2026-01-26

Accept (Poster)